# A sex-specific evolutionary interaction between *ADCY9* and *CETP*

**Isabel Gamache[1,2], Marc-André Legault[1,2,3], Jean-Christophe Grenier[2], Rocio Sanchez[2], Eric Rhéaume[1,2], Samira Asgari[4,5], Amina Barhdadi[2,3], Yassamin Feroz Zada[3], Holly Trochet[1,2], Yang Luo[4,5], Leonid Lecca[6,7], Megan Murray[4], Soumya Raychaudhuri[4,5,8,9,10], Jean-Claude Tardif[1,2], Marie-Pierre Dubé[1,2,3], Julie Hussin[1,2]***

[1]Université de Montréal, Montréal, Canada; [2]Montreal Heart Institute, Montréal, Canada; [3]Université de Montréal Beaulieu-Saucier Pharmacogenomics Centre, Montréal, Canada; [4]Center for Data Sciences, Brigham and Women's Hospital, Harvard Medical School, Boston, United States; [5]Program in Medical and Population Genetics, Broad Institute of MIT and Harvard, Cambridge, United States; [6]Socios En Salud, Lima, Peru; [7]Harvard Medical School, Boston, United States; [8]Centre for Genetics and Genomics Versus Arthritis, Manchester Academic Health Science Centre, University of Manchester, Manchester, United Kingdom; [9]Department of Biomedical Informatics, Harvard Medical School, Boston, United States; [10]Department of Medicine, Brigham and Women's Hospital and Harvard Medical School, Boston, United States

**Abstract** Pharmacogenomic studies have revealed associations between rs1967309 in the adenylyl cyclase type 9 (*ADCY9*) gene and clinical responses to the cholesteryl ester transfer protein (CETP) modulator dalcetrapib, however, the mechanism behind this interaction is still unknown. Here, we characterized selective signals at the locus associated with the pharmacogenomic response in human populations and we show that rs1967309 region exhibits signatures of positive selection in several human populations. Furthermore, we identified a variant in *CETP*, rs158477, which is in long-range linkage disequilibrium with rs1967309 in the Peruvian population. The signal is mainly seen in males, a sex-specific result that is replicated in the LIMAA cohort of over 3400 Peruvians. Analyses of RNA-seq data further suggest an epistatic interaction on *CETP* expression levels between the two SNPs in multiple tissues, which also differs between males and females. We also detected interaction effects of the two SNPs with sex on cardiovascular phenotypes in the UK Biobank, in line with the sex-specific genotype associations found in Peruvians at these loci. We propose that *ADCY9* and *CETP* coevolved during recent human evolution due to sex-specific selection, which points toward a biological link between dalcetrapib's pharmacogene *ADCY9* and its therapeutic target *CETP*.

**\*For correspondence:**
julie.hussin@umontreal.ca

## Introduction

Coronary artery disease (CAD) is the leading cause of mortality worldwide. It is a complex disease caused by the accumulation of cholesterol-loaded plaques that block blood flow in the coronary arteries. The cholesteryl ester transfer protein (CETP) mediates the exchange of cholesterol esters and triglycerides between high-density lipoproteins (HDL) and lower density lipoproteins (*Lagrost, 1994*; *Shinkai, 2012*). Dalcetrapib is a CETP modulator that did not reduce cardiovascular event rates in the overall dal-OUTCOMES trial of patients with recent acute coronary syndrome (*Schwartz et al., 2012*). However, pharmacogenomic analyses revealed that genotypes at rs1967309 in the *ADCY9* gene, coding for the ninth isoform of adenylate cyclase, modulated clinical responses to dalcetrapib

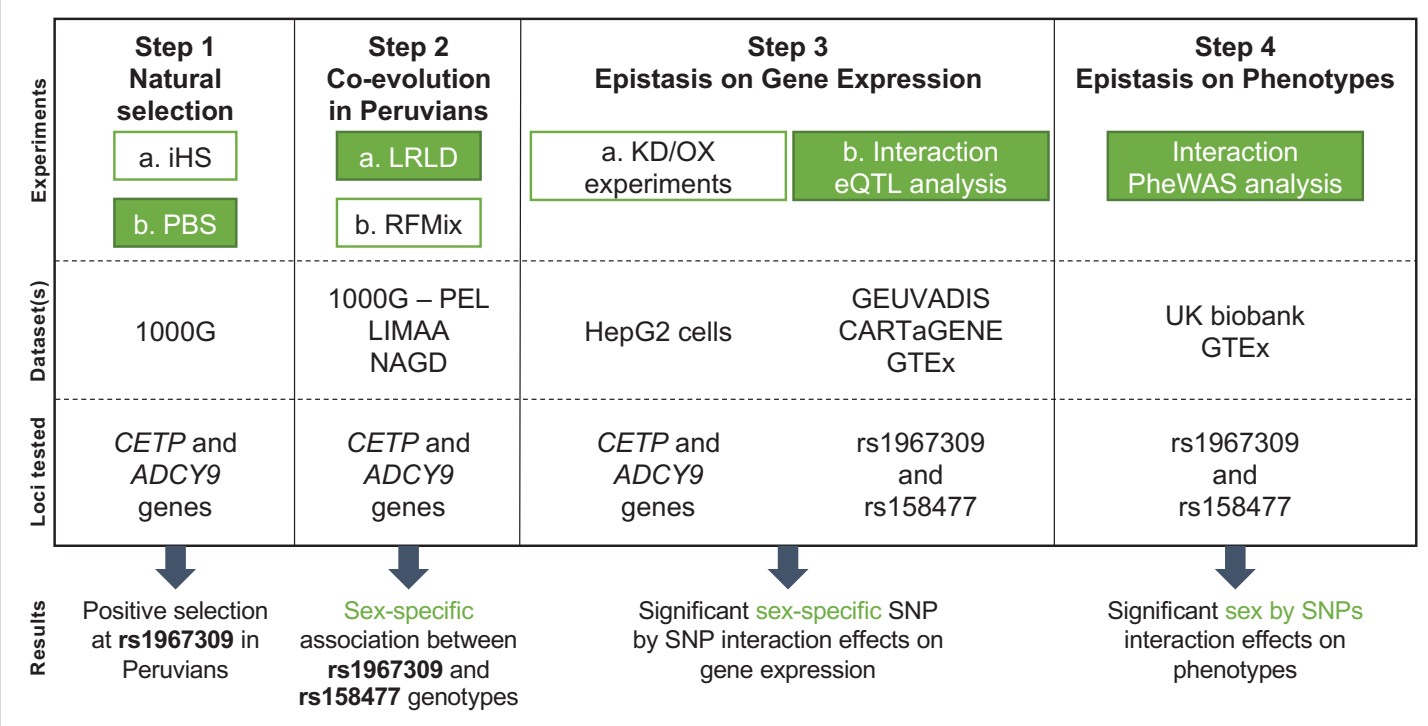

**Figure 1.** Flowchart of experimental design and main results. The four main steps of the analyses conducted in this study are reported along with the datasets used for each step and the genetic loci on which the analyses are performed. Green colored boxes represent analyses for which sex is considered. Abbreviations: KD = Knock down; OX = Overexpression.

(*Tardif et al., 2015*). Individuals who carried the AA genotype at rs1967309 in *ADCY9* had less cardiovascular events, reduced atherosclerosis progression, and enhanced cholesterol efflux from macrophages when treated with dalcetrapib compared to placebo (*Tardif et al., 2015*; *Tardif et al., 2016*). In contrast, those with the GG genotype had the opposite effects from dalcetrapib. Furthermore, a protective effect against the formation of atherosclerotic lesions was seen only in the absence of both *Adcy9* and *CETP* in mice (*Rautureau et al., 2018*), suggesting an interaction between the two genes. However, the underlying mechanisms linking *CETP* and *ADCY9*, located 50 Mb apart on chromosome 16, as well as the relevance of the rs1967309 non-coding genetic variant are still unclear.

Identification of selection pressure on a genetic variant can help shed light on its importance. Adaptation to different environments often leads to a rise in frequency of variants, by favoring survival and/or reproduction fitness. An example is the lactase gene (*LCT*) (*Bersaglieri et al., 2004*; *Enattah et al., 2007*; *Gamba et al., 2014*; *Itan et al., 2009*; *Poulter et al., 2003*), where a positively selected intronic variant in *MCM6* leads to an escape from epigenetic inactivation of *LCT* and facilitates lactase persistence after weaning (*Labrie et al., 2016*). Results of genomic studies for phenotypes such as adaptation to high-altitude hypoxia in Tibetans (*Yi et al., 2010*), fatty acid metabolism in Inuits (*Fumagalli et al., 2015*) or response to pathogens across populations (*Hollenbach et al., 2001*) have also been confirmed by functional studies (*Li et al., 2018*; *Reynolds et al., 2020*; *Tashi et al., 2017*; *Meyer et al., 2018*; *Blais et al., 2012*). Thus, population and regulatory genomics can be leveraged to unveil the effect of genetic mutations at a single non-coding locus and reveal the biological mechanisms of adaptation.

When two or more loci interact during adaptation, a genomic scan will likely be underpowered to pinpoint the genetic determinants. In this study, we took a multi-step approach on the *ADCY9* and *CETP* candidate genes to specifically study their interaction (*Figure 1*). We used a joint evolutionary analysis to evaluate the potential signatures of selection in these genes (Step 1), which revealed positive selection pressures acting on *ADCY9*. Sex-specific genetic associations between the two genes are discovered in Peruvians (Step 2), a population in which natural selection for high-altitude was previously found on genes related to cardiovascular health (*Crawford et al., 2017*). Furthermore,

our know-down experiments and analyses of large-scale transcriptomics (Step 3) as well as available phenome-wide resources (Step 4) bring further evidence of a sex-specific epistatic interaction between *ADCY9* and *CETP*.

## Results

### Signatures of selection at rs1967309 in *ADCY9* in human populations

The genetic variant rs1967309 is located in intron 2 of *ADCY9*, in a region of high linkage disequilibrium (LD), in all subpopulations in the 1000 Genomes Project (1000G), and harbors heterogeneous genotype frequencies across human populations (*Figure 2a*). Its intronic location makes it difficult to assess its functional relevance but exploring selective signals around intronic SNPs in human populations can shed light on their importance. In African populations (AFR), the major genotype is AA, which is the homozygous genotype for the ancestral allele, whereas in Europeans (EUR), AA is the minor genotype. The frequency of the AA genotype is slightly higher in Asia (EAS, SAS) and America (AMR) compared to that in Europe, becoming the most frequent genotype in the Peruvian population (PEL). Using the integrated haplotype score (iHS) (*Voight et al., 2006*) (Step 1 a, *Figure 1*), a statistic that enables the detection of evidence for recent strong positive selection (typically when |iHS| > 2), we observed that several SNPs in the LD block around rs1967309 exhibit selective signatures in non-African populations ($|iHS_{SAS}| = 2.66$, $|iHS_{EUR}| = 2.31$), whereas no signal is seen in this LD block in African populations (*Figure 2b*, *Appendix 1—figure 1*, Appendix 1). Our analyses suggest that this locus in *ADCY9* has been the target of recent positive selection in several human populations, with multiple, possibly independent, selective signals detectable around rs1967309. However, recent positive selection as measured by iHS does not seem to explain the notable increase in frequency for the A allele in the PEL population ($f_A = 0.77$), compared to the European ($f_A = 0.41$), Asian ($f_A = 0.44$), and other American populations ($f_A = 0.54$ in AMR without PEL).

To test whether the difference between PEL and other AMR allele frequencies at rs1967309 is significant, we used the population branch statistic (PBS) (Step 1b, *Figure 1*). This statistic has been developed to locate selection signals by summarizing differentiation between populations using a three-way comparison of allele frequencies between a specific group, a closely related population, and an outgroup (*Yi et al., 2010*). It has been shown to increase power to detect incomplete selective sweeps on standing variation. Applying this statistic to investigate rs1967309 allele frequency in PEL, we used Mexicans (MXL) as a closely related group and a Chinese population (CHB) as the outgroup (Methods). Over the entire genome, the CHB branches are greater than PEL and MXL branches ($mean_{CHB} = 0.020$, $mean_{MXL} = 0.008$, $mean_{PEL} = 0.009$), which reflects the expectation under genetic drift. However, the estimated PEL branch length at rs1967309 (*Figure 2c*), which reflects differentiation since the split from the MXL population ($PBS_{PEL,rs1967309}=0.051$, empirical p-value = 0.014), surpasses the CHB branch length ($PBS_{CHB,rs1967309}=0.049$, empirical p-value > 0.05), which reflects differentiation since the split between Asian and American populations, whereas no such effect is seen in MXL ($PBS_{MXL,rs1967309}=0.026$, empirical p-value > 0.05), or for any other AMR populations. Furthermore, the PEL branch lengths at several SNPs in this LD block (*Figure 2c*) are in the top 5 % of all PEL branch lengths across the whole genome ($PBS_{PEL,95th} = 0.031$), whereas these increased branch lengths are not observed outside of the LD block (*Figure 2c*). These results are robust to the choice of the outgroup and the closely related AMR population (Materials and methods).

The increase in frequency of the A allele at rs1967309 is also seen in genotype data from Native American populations (*Reich et al., 2012*), with Andeans showing genotype frequencies highly similar to PEL ($f_A = 0.77$, *Figure 2a*). The PEL population has a large Andean ancestry (Materials and methods, *Appendix 1—figure 2a and b*) and almost no African ancestry, strongly suggesting that the increase in AA genotype arose in the Andean population and not from admixture with Africans. The PEL individuals that harbor the AA genotype for rs1967309 do not exhibit a larger genome-wide Andean ancestry than non-AA individuals (p-value = 0.30, Mann-Whitney U test). Overall, these results suggest that the ancestral allele A at rs1967309, after dropping in frequency following the out-of-Africa event, has increased in frequency in the Andean population and has been preferentially retained in the Peruvian population's genetic makeup, potentially because of natural selection.

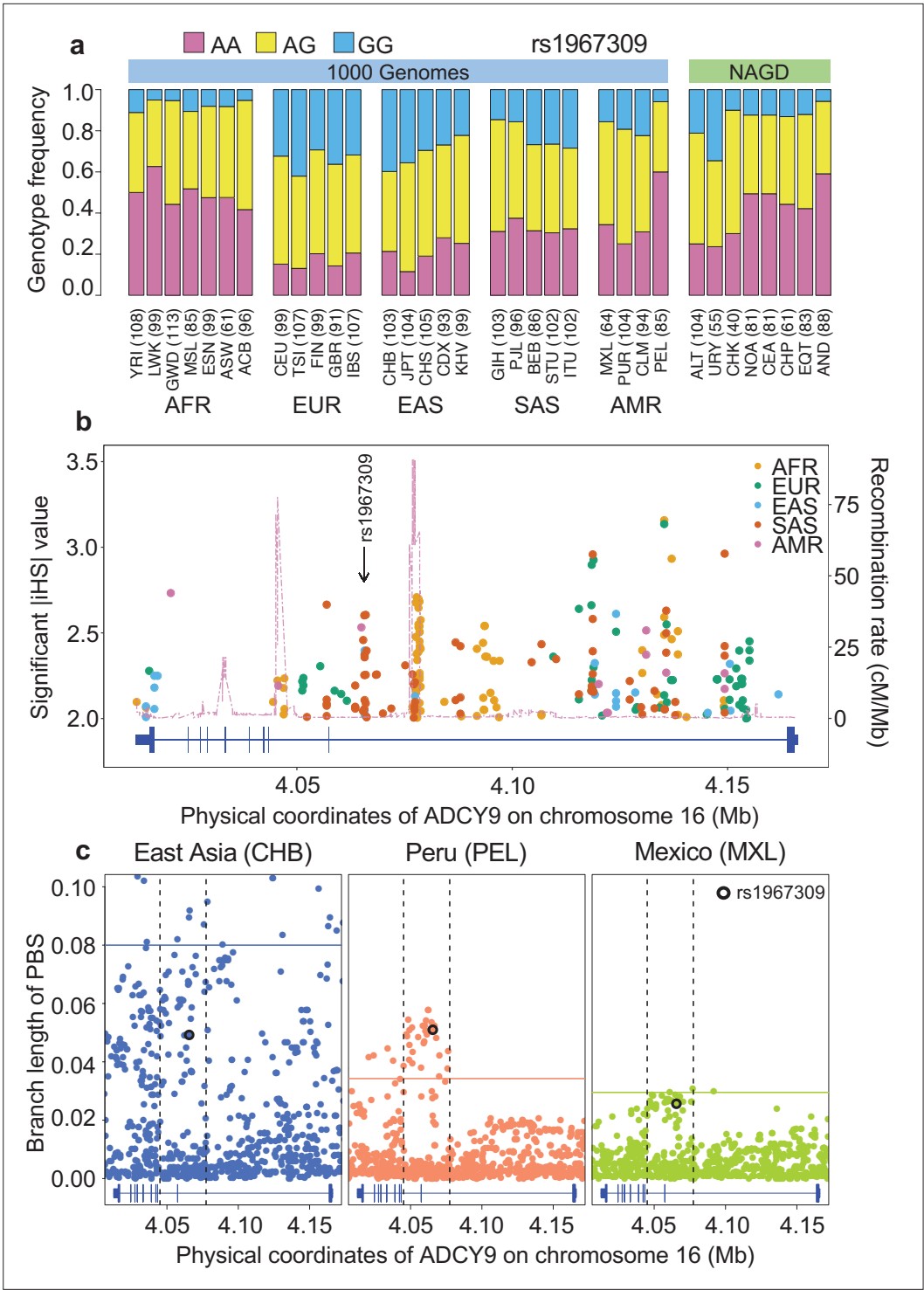

**Figure 2.** Natural selection signature at rs1967309 in *ADCY9*. (**a**) Genotype frequency distribution of rs1967309 in populations from the 1000 Genomes (1000G) Project and in Native Americans (NAGD). (**b**) Significant iHS values (absolute values above 2) for 1000G continental populations and recombination rates from AMR-1000G population-specific genetic maps, in the *ADCY9* gene. (**c**) PBS values in the *ADCY9* gene, in CHB (outgroup, left panel), PEL (middle panel), and MXL (right panel). Horizontal lines represent the 95th percentile PBS value genome-wide for each population. Vertical dotted black lines define the LD block around rs1967309 (black circle) from 1000G population-specific genetic maps. Gene plots for *ADCY9* showing location of its exons are presented in blue below each plot. Abbreviations: Altaic from Mongolia and Russia: ALT; Uralic Yukaghir from Russia: URY;

*Figure 2 continued on next page*

*Figure 2 continued*

Chukchi Kamchatkan from Russia: CHK; Northern American from Canada, Guatemala and Mexico: NOA; Central American from Costal Rica and Mexico: CEA; Chibchan Paezan from Argentina, Bolivia, Colombia, Costa Rica, and Mexico: CHP; Equatorial Tucanoan from Argentina, Brazil, Colombia, Gualana and Paraguay: EQT; Andean from Bolivia, Chile, Colombia and Peru: AND. For 1000G populations, abbreviations can be found here https://www.internationalgenome.org/category/population/.

The online version of this article includes the following figure supplement(s) for figure 2:

**Source data 1.** Source file for genotype frequency distribution of rs1967309.

**Source data 2.** Source file for iHS plot in the *ADCY9* gene.

**Source data 3.** Source file for PBS plots in the *ADCY9* gene.

## Evidence for co-evolution between *ADCY9* and CETP in Peru

The pharmacogenetic link between *ADCY9* and the CETP modulator dalcetrapib raises the question of whether there is a genetic relationship between rs1967309 in *ADCY9* and *CETP*, both located on chromosome 16. Such a relationship can be revealed by analyzing patterns of long-range linkage disequilibrium (LRLD) (*Lewontin and Kojima, 1960*; *Rohlfs et al., 2010*), in order to detect whether specific combinations of alleles (or genotypes) at two loci are particularly overrepresented. To do so, we calculated the genotyped-based linkage disequilibrium ($r^2$) (Step 2 a, *Figure 1*) between rs1967309 and each SNP in *CETP* with minor allele frequency (MAF) above 0.05. In the Peruvian population, there are four SNPs, (including two in perfect LD in PEL) that exhibit $r^2$ values with rs1967309 that are in the top 1 % of $r^2$ values (*Figure 3a*) computed for all 37,802 pairs of SNPs in *ADCY9* and *CETP* genes with MAF >0.05 (Materials and methods). Despite the $r^2$ values themselves being low ($r^2_{rs158477}$=0.080, $r^2_{rs158480;rs158617}$=0.089, $r^2_{rs12447620}$=0.090), these values are highly unexpected for these two genes situated 50 Mb apart (*ADCY9/CETP* empirical p-value < 0.006, *Appendix 1—table 1*) and thus correspond to a significant LRLD signal. This signal is not seen in other 1000G populations (*Appendix 1—table 1*). We also computed $r^2$ between the four identified SNPs' genotypes and all *ADCY9* SNPs with MAF above 0.05 (*Figure 3b*). The distribution of $r^2$ values for the rs158477 *CETP* SNP shows a clear bell-shaped pattern around rs1967309 in *ADCY9*, which strongly suggests the rs1967309-rs158477 genetic association detected is not simply a statistical fluke, while the signal in the region for the other SNPs is less conclusive. The SNP rs158477 in *CETP* is also the only one that has a PEL branch length value higher than the 95th percentile, also higher than the CHB branch length value (PBS$_{PEL,rs158477}$=0.062, *Appendix 1—figure 3a*), in line with the observation at rs1967309. Strikingly, this *CETP* SNP's genotype frequency distribution across the 1000G and Native American populations resembles that of rs1967309 in *ADCY9* (*Figure 3c*).

Given that the Peruvian population is admixed (*Harris et al., 2018*), particular enrichment of genome segments for a specific ancestry, if present, would lead to inflated LRLD between these segments (*Li and Nei, 1974*; *Nei and Li, 1973*; *Park, 2019*; *Slatkin, 2008*), we thus performed several admixture-related analyses (Step 2b, *Figure 1*). No significant enrichment is seen at either locus and significant LRLD is also seen in the Andean source population (*Figure 3—figure supplement 1a, b*, Appendix 1). Furthermore, we see no enrichment of Andean ancestry in individuals harboring the overrepresented combination of genotypes, AA at rs1967309+ GG at rs158477, compared to other combinations (p-value = 0.18, Mann-Whitney U test). These results show that admixture patterns in PEL cannot be solely responsible for the association found between rs1967309 and rs158477. Finally, using a genome-wide null distribution which allows to capture the LRLD distribution expected under the admixture levels present in this sample (Appendix 1), we show that the $r^2$ value between the two SNPs is higher than expected given their allele frequencies and the physical distance between them (genome-wide empirical p-value = 0.01, *Figure 3d*). Taken together, these findings strongly suggest that the AA/GG combination is being transmitted to the next generation more often (i.e. is likely selectively favored) which reveals a signature of co-evolution between *ADCY9* and *CETP* at these loci.

Still, such a LRLD signal can be due to a small sample size (*Park, 2019*). To confirm independently the association between genotypes at rs1967309 of *ADCY9* and rs158477 of *CETP*, we used the LIMAA cohort (*Asgari et al., 2020*; *Luo et al., 2019*), a large cohort of 3509 Peruvian individuals with genotype information, to replicate our finding. The ancestry distribution, as measured by RFMix (Methods) is similar between the two cohorts (*Appendix 1—figure 2a*), however, the LIMAA cohort

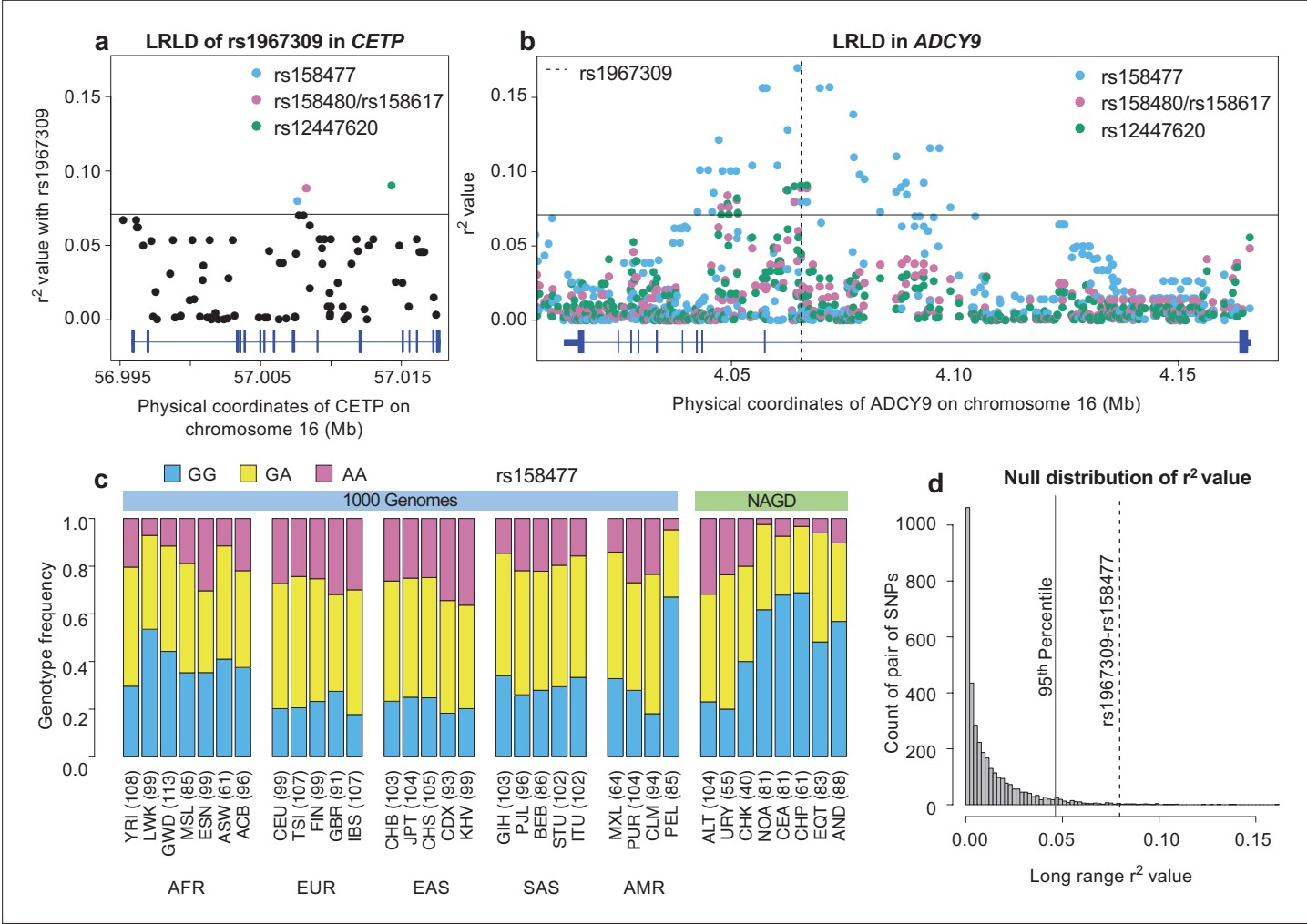

**Figure 3.** Long-range linkage disequilibrium between rs1967309 and rs158477 in Peruvians from Lima, Peru. (**a**) Genotype correlation ($r^2$) between rs1967309 and all SNPs with MAF >5% in *CETP*, for the PEL population. (**b**) Genotype correlation between the three loci identified in (**a**) to be in the 99th percentile and all SNPs with MAF >5% in *ADCY9*. The dotted line indicates the position of rs1967309. The horizontal lines in (**a,b**) represent the threshold for the 99th percentile of all comparisons of SNPs (MAF >5%) between *ADCY9* and *CETP*. *Figure 3—figure supplement 1* presents the same plots for Andeans and in the replication cohort (LIMAA) and *Figure 3—figure supplement 2* compares the $r^2$ values between PEL and LIMAA (**c**) Genotype frequency distribution of rs158477 in 1000G and Native American populations. (**d**) Genomic distribution of $r^2$ values from 3,513 pairs of SNPs separated by between 50 and 60 Mb and 61 ± 10 cM away across all Peruvian chromosomes from the PEL sample, compared to the rs1967309-rs158477 $r^2$ value (dotted gray line) (genome-wide empirical p-value = 0.01). The vertical black line shows the threshold for the 95th percentile threshold of all pairs. Gene plots showing location of exons for *CETP* (**a**) and *ADCY9* (**b**) are presented in blue below each plot. Abbreviations: Altaic from Mongolia and Russia: ALT; Uralic Yukaghir from Russia: URY; Chukchi Kamchatkan from Russia: CHK; Northern American from Canada, Guatemala and Mexico: NOA; Central American from Costal Rica and Mexico: CEA; Chibchan Paezan from Argentina, Bolivia, Colombia, Costa Rica and Mexico: CHP; Equatorial Tucanoan from Argentina, Brazil, Colombia, Gualana and Paraguay: EQT; Andean from Bolivia, Chile, Colombia and Peru: AND. For 1000G populations, abbreviations can be found here https://www.internationalgenome.org/category/population/.

The online version of this article includes the following figure supplement(s) for figure 3:

**Source data 1.** R$^2$ values of all SNPs between *ADCY9* and *CETP* genes in the PEL population from 1000G.

**Source data 2.** Source file for genotype frequency distribution of rs158477.

**Source data 3.** R$^2$ values used for the null distribution in the PEL population from 1000G.

**Figure supplement 1.** Long-range linkage disequilibrium in the Andean population from the Native Population (n = 88) (**a,b**) and in the LIMAA cohort (n = 3243) (**c,d**).

**Figure supplement 2.** Comparison of genotype correlation between Peruvian from 1000G and from the LIMAA cohort.

population structure shows additional subgroups compared to the 1000G PEL population sample (*Appendix 1—figure 2c-e*): to limit confounders, we excluded individuals coming from these subgroups (Appendix 1). In this dataset (N = 3,243), the pair of SNPs rs1967309-rs158477 is the only pairs identified in PEL who shows evidence for LRLD, with an $r^2$ value in the top 1 % of all pairs of SNPs in *ADCY9* and *CETP* (*ADCY9/CETP* empirical p-value = 0.003, *Figure 3—figure supplement 1c, d*, *Figure 3—figure supplement 2*, *Appendix 1—table 1*). The $r^2$ test used above is powerful to detect allelic associations, but the net association measured will be very small if selection acts on a specific genotype combination rather than on alleles. In that scenario, and when power allows it, the genotypic association is better assessed by with a $\chi^2$ distributed test statistic (with four degrees of freedom, $\chi_4^2$) comparing the observed and expected genotype combination counts (*Rohlfs et al., 2010*). The test confirmed the association in LIMAA ($\chi_4^2$ = 82.0, permutation p-value < 0.001, genome-wide empirical p-value = 0.0003, Appendix 1). The association discovered between rs1967309 and rs158477 is thus generalizable to the Peruvian population and not limited to the 1000 G PEL sample.

## Sex-specific long-range linkage disequilibrium signal

Because the allele frequencies at rs1967309 were suggestively different between males and females (*Figure 4—figure supplement 1*), we performed sex-stratified PBS analyses, which suggested that the LD block around rs1967309 is differentiated between sexes in the Peruvians (*Figure 4—figure supplement 2*, Appendix 1). We therefore explored further the effect of sex on the LRLD association found between rs1967309 and rs158477 and performed sex-stratified LRLD analyses. These analyses revealed that the correlation between rs1967309 and rs158477 is only seen in males in PEL (*Figure 4a and b*, *Appendix 1—figure 4a and b*, *Appendix 1—table 1*): the $r^2$ value rose to 0.348 in males (*ADCY9/CETP* empirical p-value = 8.23 × 10⁻⁵, genome-wide empirical p-value < 2.85 × 10⁻⁴, N = 41) and became non-significant in females (*ADCY9/CETP* empirical p-value = 0.78, genome-wide empirical p-value = 0.80, N = 44). In the Andean population, the association between rs1967309 and rs158477 is not significant when we stratified by sex (*Appendix 1—table 1*), but we still see significant association signals with rs158477 at other SNPs in *ADCY9* LD block in both sexes (*Figure 4—figure supplement 3*, *Appendix 1—figure 5a,b*). The LRLD result in PEL cannot be explained by differences of Andean ancestry proportion between males and females (p-value = 0.27, Mann-Whitney U test). A permutation analysis that shuffled the sex labels of samples established that the observed difference between the sexes is larger than what we expect by chance (p-value = 0.002, *Appendix 1—figure 4c*, Appendix 1). In the LIMAA cohort, we replicate this sex-specific result (*Figure 4c and d*, *Appendix 1—figure 5c,d,e,f*, *Appendix 1—table 1*) where the $r^2$ test is significant in males (*ADCY9/CETP* empirical p-value = 0.003, N = 1,941) but not in females (*ADCY9/CETP* empirical p-value = 0.52, N = 1302). The genotypic $\chi_4^2$ test confirms the association between *ADCY9* and *CETP* is present in males ($\chi_4^2$ = 56.6, permutation p-value = 0.001, genome-wide empirical p-value = 0.002, Appendix 1), revealing an excess of rs1967309-AA+ rs158477 GG. This is also the genotype combination driving the LRLD in PEL. In females, the test also shows a weaker but significant effect ($\chi_4^2$ = 37.0, permutation p-value = 0.017, genome-wide empirical p-value = 0.001) driven by an excess of a different genotype combination, rs1967309-AA+ rs158477 AA, which is, however, not replicated in PEL possibly because of lack of power (Appendix 1).

## Epistatic effects on CETP gene expression

LRLD between variants can suggest the existence of gene-gene interactions, especially if they are functional variants (*Park, 2019*). In order to be under selection, mutations typically need to modulate a phenotype or an endophenotype, such as gene expression. We have shown previously (*Rautureau et al., 2018*) that CETP and *Adcy9* interact in mice to modulate several phenotypes, including atherosclerotic lesion development. To test whether these genes interact in humans, we knocked down (KD) *ADCY9* in hepatocyte HepG2 cells (Step 3 a, *Figure 1*) and performed RNA sequencing on five KD biological replicates and five control replicates, to evaluate the impact of decreased *ADCY9* expression on the transcriptome. We confirmed the KD was successful as *ADCY9* expression is reduced in the KD replicates (*Figure 5a*), which represents a drastic drop in expression compared to the whole transcriptome changes (False Discovery Rate [FDR] = 4.07 x 10⁻¹⁴, Materials and methods). We also observed that *CETP* expression was increased in *ADCY9-KD* samples compared to controls (*Figure 5a*), an increase that is also transcriptome-wide significant (FDR = 1.97 × 10⁻⁷, ß = 1.257). This increased expression was validated by qPCR, and western blot also showed increased CETP protein product (Materials and methods, *Figure 5—figure supplement*

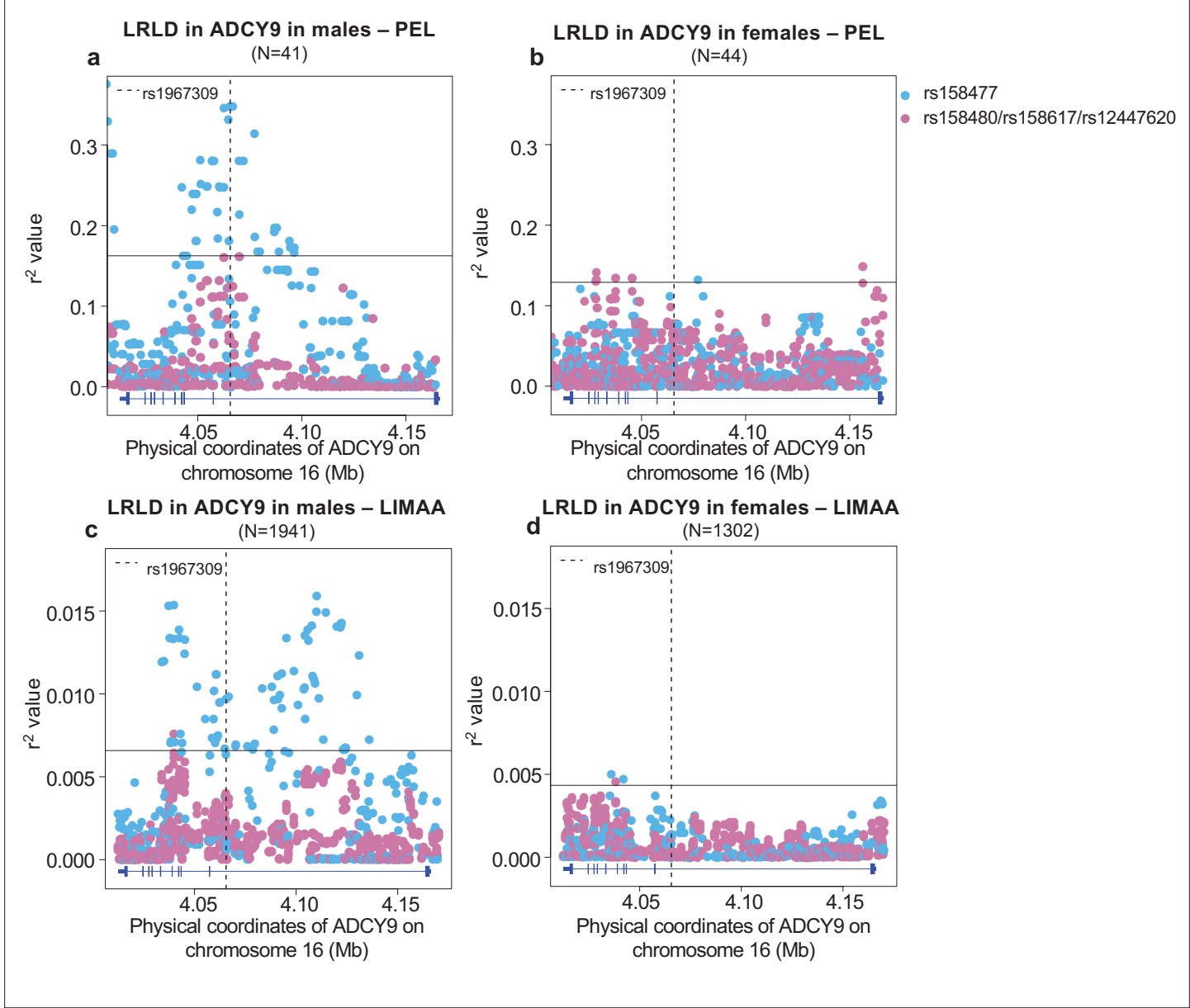

**Figure 4.** Sex-specific long-range linkage disequilibrium. Genotype correlation between the loci identified in *CETP* in *Figure 3a* and all SNPs with MAF >5% in *ADCY9* for (**a,b**) the PEL population and (**c,d**) LIMAA cohort in males (**a,c**) and in females (**b,d**). Genotype frequencies per sex are shown in *Figure 4—figure supplement 1* and sex-specific PBS values in *Figure 4—figure supplement 2*. The horizontal line shows the threshold for the 99th percentile of all comparisons of SNPs (MAF >5%) between *ADCY9* and *CETP*. The vertical dotted line represents the position of rs1967309. Blue dots represent the rs158477 SNPs and pink represents the other three SNPs identified in *Figure 3a* (rs158480, rs158617, and rs12447620), which are in near-perfect LD. *Figure 4—figure supplement 3* shows the same analysis in Andeans from NAGD. Gene plots for *ADCY9* showing location of its exons are presented in blue below each plot.

The online version of this article includes the following figure supplement(s) for figure 4:

**Source data 1.** R² values of all SNPs between *ADCY9* and *CETP* genes in the PEL population from 1000G and LIMAA cohort in male and female.

**Figure supplement 1.** Genotype frequency distribution per sex.

**Figure supplement 2.** PBS values in the *ADCY9* per sex, comparing the CHB (outgroup), MXL and PEL.

**Figure supplement 3.** Sex-specific long-range linkage disequilibrium in the Andean population (NAGD).

*1a, b*, Appendix 1), but its overexpression did not significantly modulate *CETP* expression (*Figure 5— figure supplement 1c*). Knocking down or overexpressing *CETP* did not impact *ADCY9* expression on qPCR (*Figure 5—figure supplement 1d, e*). These experiments demonstrate an interaction between

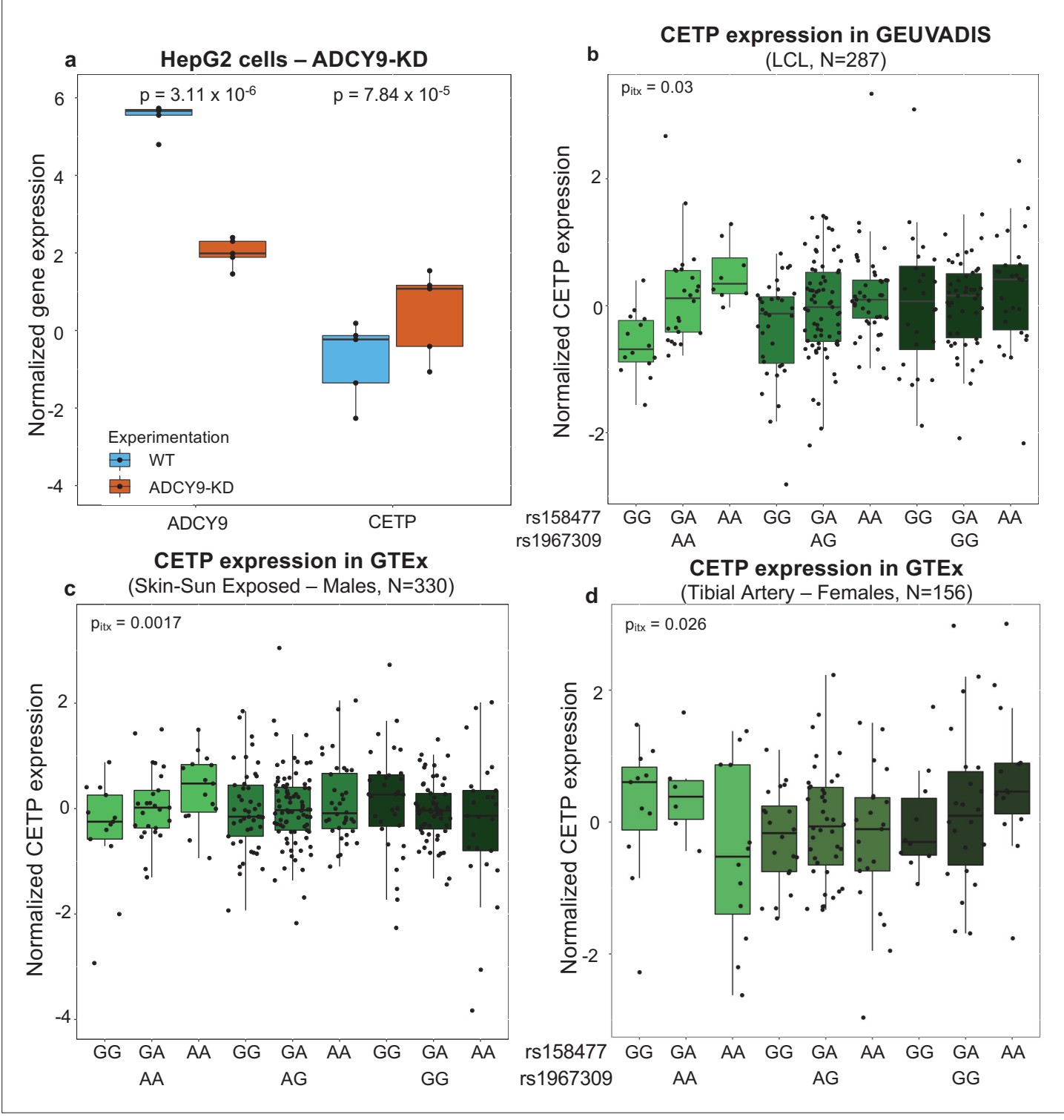

**Figure 5.** Effect of *ADCY9* on *CETP* expression. (**a**) Normalized expression of *ADCY9* or *CETP* genes depending on wild type (WT) and *ADCY9-KD* in HepG2 cells from RNA sequencing on five biological replicates in each group. p-Values were obtained from a two-sided Wilcoxon paired test. qPCR and western blot results in HepG2 are presented in *Figure 5—figure supplement 1*. (**b,c,d**) *CETP* expression depending on the combination of rs1967309 and rs158477 genotypes in (**b**) GEUVADIS (p-value = 0.03, ß = –0.22, N = 287), (**c**) GTEx-Skin Sun Exposed in males (p-value = 0.0017, ß = –0.32, N = 330) and in (**d**) GTEx-tibial artery in females (p-value = 0.026, ß = 0.38, N = 156), for individuals of European descent according to principal component analysis. p-Values reported were obtained from a two-way interaction of a linear regression model for the maximum number of PEER/sPEER factors considered. *Figure 5—figure supplement 2* show the interaction p-values depending on number of PEER/sPEER factors included in the linear models.

*Figure 5 continued on next page*

*Figure 5 continued*

The online version of this article includes the following figure supplement(s) for figure 5:

**Source data 1.** Normalized expression of ADCY9 and CETP genes HepG2 cells.

**Source data 2.** Residual of CETP expression by genotype.

**Figure supplement 1.** *ADCY9*/CETP interaction in HepG2 cells.

**Figure supplement 2.** Interaction effect p-values on *CETP* expression depending by the number of PEER factors in Skin-sun exposed (**a,b**) and Tibial artery (**c,d**) in GTEx.

*ADCY9* and *CETP* at the gene expression level and raised the hypothesis that *ADCY9* potentially modulates the expression of *CETP* through a genetic effect mediated by rs1967309.

To test for potential interaction effects between rs1967309 and *CETP*, we used RNA-seq data from diverse projects in humans: the GEUVADIS project (*Lappalainen et al., 2013*), the Genotype-tissue Expression (GTEx v8) project (*GTEx Consortium, 2013*) and CARTaGENE (CaG) (*Awadalla et al., 2013*) (Step 3b, *Figure 1*). When looking across tissues in GTEx, *ADCY9* and *CETP* expressions negatively correlate in almost all the tissues (*Appendix 1—figure 6*, Appendix 1), which is consistent with the effect observed during the *ADCY9-KD* experiment, showing increased expression of *CETP* expression when *ADCY9* is lowly expressed (*Figure 5a*, *Figure 5—figure supplement 1a, b*). We evaluated the effects of the SNPs on expression levels of *ADCY9* and *CETP* by modelling both SNPs as continuous variables (additive model) (Methods). The *CETP* SNP rs158477 was reported as an expression quantitative trait locus (eQTL) in GTEx v7 and, in our models, shows evidence of being a *cis* eQTL of *CETP* in several other tissues (Appendix 1), although not reaching genome-wide significance. To test specifically for an epistatic effect between rs1967309 and rs158477 on *CETP* expression, we included an interaction term in eQTL models (Materials and methods). We note here that we are testing for association for this specific pair of SNPs only, and that effects across tissues are not independent, such that we set our significance threshold at p-value = 0.05. This analysis revealed a significant interaction effect (p-value = 0.03, ß = −0.22) between the two SNPs on *CETP* expression in GEUVADIS lymphoblastoid cell lines (*Figure 5b*, *Appendix 1—figure 7a*). In rs1967309 AA individuals, copies of the rs158477 A allele increased *CETP* expression by 0.46 (95% CI 0.26–0.86) on average. In rs1967309 AG individuals, copies of the rs158477 A allele increased CETP expression by 0.24 (95% CI 0.06–0.43) on average and the effect was null in rs1967309 GG individuals (p-value$_{GG}$ = 0.58). This suggests that the effect of rs158477 on *CETP* expression changes depending on genotypes of rs1967309. The interaction is also significant in several GTEx tissues, most of which are brain tissues, like hippocampus, hypothalamus, and substantia nigra, but also in skin, although we note that the significance of the interaction depends on the number of PEER factors included in the model (*Appendix 1—figure 8*). These factors are needed to correct for unknown biases in the data, but also potentially lead to decreased power to detect interaction effects (*Brynedal et al., 2017*). In CaG whole blood samples, the interaction effect using additive genetic effect at rs1967309 was not significant, similarly to results from GTEx in whole blood samples. However, given the larger size of the dataset, we evaluated a genotypic encoding for the rs1967309 SNP in which the interaction effect is significant (*P*-value = 0.008, *Appendix 1—figure 7b*) in whole blood, suggesting that rs1967309 could be modulating rs158477 eQTL effect, in this tissue at least, with a genotype-specific effect. We highlight that the sample sizes of current transcriptomic resources do not allow to detect interaction effects at genome-wide significance, however the likelihood of finding interaction effects between our two SNPs on *CETP* expression in three independent datasets is unlikely to happen by chance alone, providing evidence for a functional genetic interaction.

Given the sex-specific results reported above, we stratified our interaction eQTL analyses by sex. We observed that the interaction effect on *CETP* expression in CaG whole blood samples (N$_{male}$ = 359) is restricted to male individuals, and, despite low power due to smaller sample size in GEUVADIS, the interaction is also only suggestive in males (*Appendix 1—figure 7c and d*). In GTEx, most well-powered tissues that showed a significant effect in the sex-combined analyses also harbor male-specific interactions (*Appendix 1—figure 9*). For instance, GTEx skin male samples (N$_{male}$ = 330) show the most significant male-specific interaction effects, with the directions of effects replicating the sex-combined result in GEUVADIS (an increase of *CETP* expression for each rs158477 A allele in rs1967309 AA individuals) albeit with an observable reversal of the direction in rs1967309 GG individuals (decrease of *CETP* expression with additional rs158477 A alleles) (*Figure 5c*, *Figure 5—figure supplement 2a*).

However, significant effects in females are detected in tissues not previously seen as significant for the interaction in the sex-combined analysis, in the tibial artery (*Figure 5d*, *Figure 5—figure supplement 2*) and the heart atrial appendage (*Appendix 1—figure 9*). For tissues with evidence of sex-specific effects in stratified analyses, we also tested the effect of an interaction between sex, rs158477 and rs1967309 (Materials and methods) on *CETP* expression: the three-way interaction is only significant for tibial artery (*Figure 5—figure supplement 2*).

## Epistatic effects on phenotypes

The interaction effect of rs1967309 and rs158477 on *CETP* expression in several tissues, found in multiple independent RNA-seq datasets, coupled with the detection of LRLD between these SNPs in the Peruvian population suggest that selection may act jointly on these loci, specifically in Peruvians or Andeans. These populations are well known for their adaptation to life in high altitude, where the oxygen pressure is lower and where the human body is subjected to hypoxia (*Beall, 2007*; *Brutsaert et al., 2005*; *Julian and Moore, 2019*; *Moore, 2017a*). High altitude hypoxia impacts individuals' health in many ways, such as increased ventilation, decreased arterial pressure, and alterations of the energy metabolism in cardiac and skeletal muscle (*Milledge et al., 2007*; *Murray, 2016*). To test which phenotype(s) may explain the putative coevolution signal discovered (Step 4, *Figure 1*), we investigated the impact of the interaction between rs1967309 and rs158477 on several physiological traits, energy metabolism and cardiovascular outcomes using the UK Biobank and GTEx cohort (*Figure 6—figure supplement 1*, *Appendix 1—table 2*). The UK Biobank has electronic medical records and GTEx has cause of death and variables from medical questionnaires (*GTEx Consortium, 2013*). The interaction term was found to be nominally significant (p-value < 0.05) for forced vital capacity (FVC), forced expiratory volume in 1 s (FEV1) and whole-body water mass, and suggestive (p-value < 0.10) for the basal metabolic rate, all driven by the effects in females (*Figure 6a*). For CAD, the interaction is suggestive (p-value < 0.10) and, in this case, driven by males (*Figure 6a*).

Given this sex-specific result on CAD, the condition targeted by dalcetrapib, we tested the effect of an interaction between sex, rs158477 and rs1967309 (genotypic encoding, see Materials and methods) on binary cardiovascular outcomes including myocardial infarction (MI) and CAD. For CAD, we see a significant three-way interaction effect, meaning that for individuals carrying the AA genotype at rs1967309, the association between rs158477 and CAD is in the opposite direction in males and females. In other words, in rs1967309-AA females, having an extra A allele at rs158477, which is associated with higher *CETP* expression (*Figure 5b*), has a protective effect on CAD. Conversely, in rs1967309-AA males, each A allele at rs158477 increases the probability of having an event (*Figure 6a*). Little effect is seen in either sex for AG or GG at rs1967309, although the heterozygotes AG behave differently in females (which further justifies the genotypic encoding of rs1967309). The beneficial effect of the interaction on CAD thus favors the rs1967309-AA+ rs153477 GG males and the rs1967309-AA+ rs153477 AA females, two genotype combinations which are respectively enriched in a sex-specific manner in the LIMAA cohort (Appendix 1). Again, observing such a result that concords with the direction of effects in the LRLD sex-specific finding is noteworthy. A significant interaction between the SNPs is also seen in the GTEx cohort (p-value = 0.004, *Figure 6—figure supplement 2*, Appendix 1), using questionnaire phenotypes reporting on MI, but the small number of individuals precludes formally investigating sex effects.

Among the biomarkers studied (*Appendix 1—table 2*), only lipoprotein(a) [Lp(a)] is suggestive in males (*P*-value = 0.08) for an interaction between rs1967309 and rs158477, with the same direction of effect as that for CAD (*Figure 6*). Again, given the differences observed between the sexes, we tested the effect of an interaction between sex, rs158477 and rs1967309 (genotypic coding, Materials and methods) on biomarkers, and only Lp(a) was nominally significant in a three-way interaction (p-value = 0.049). The pattern is similar to the results for CAD, ie. a change in the effect of rs158477 depending on the genotype of rs1967309 in males, with the effect for AA females in the opposite direction compared to males (*Figure 6b*). These concordant results between CAD and Lp(a) support that the putative interaction effect between the loci under study on phenotypes involves sex as a modifier.

## Discussion

In this study, we used population genetics, transcriptomics and interaction analyses in biobanks to study the link between *ADCY9* and *CETP*. Our study revealed selective signatures in *ADCY9* and a significant

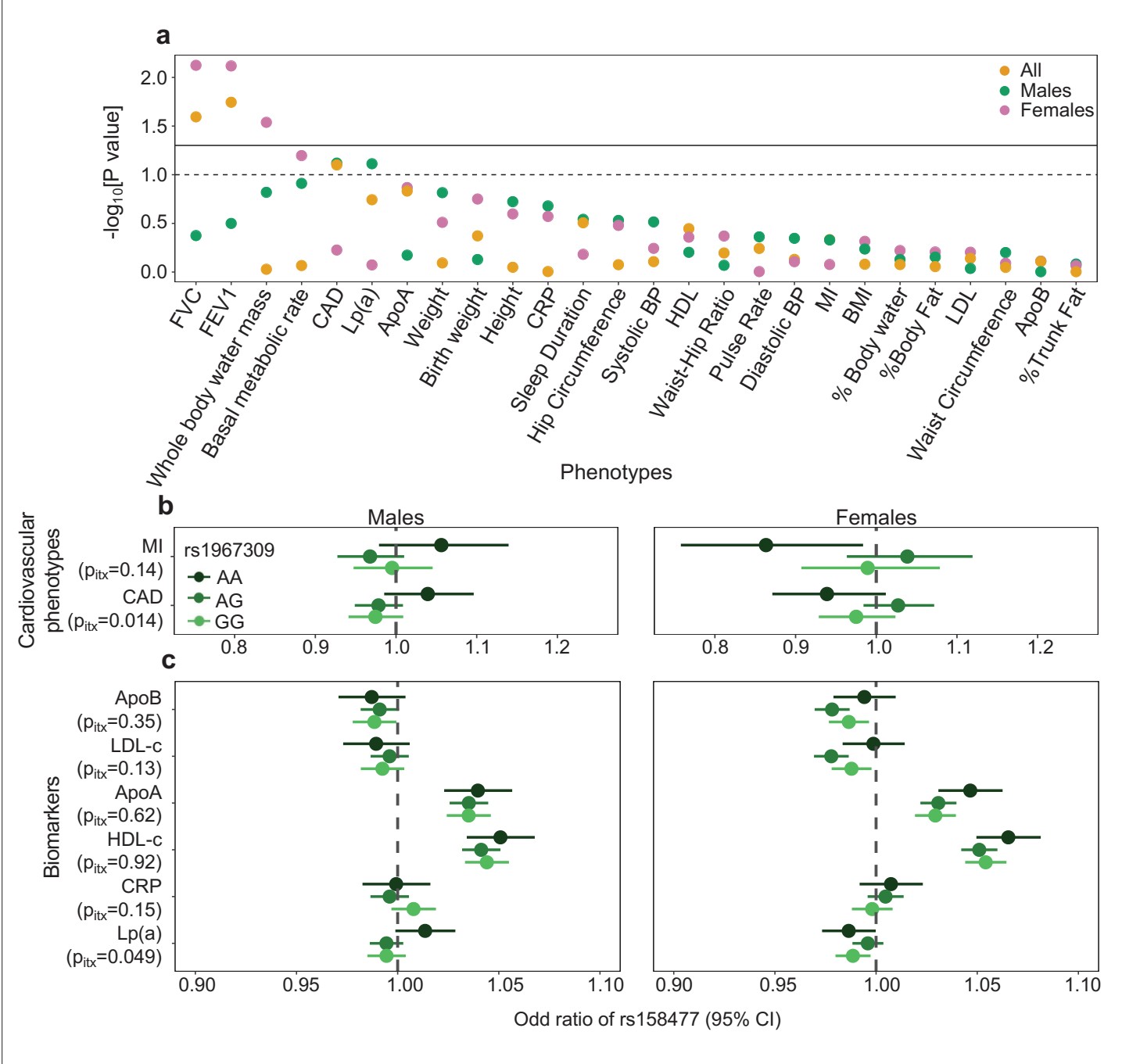

**Figure 6.** Epistatic association of rs1967309 and rs158477 on phenotypes in the UK biobank. (**a**) Significance of the interaction effect between rs1967309 and rs158477 on several physiological traits, energy metabolism and cardiovascular outcomes overall and stratified by sex in the UK biobank. Horizontal lines represent the p-value thresholds at 0.05 (plain) and 0.10 (dotted). Single-SNP p-values are shown in *Figure 6—figure supplement 1*. (**b,c**) Sex-stratified effects of rs158477 on (**b**) cardiovascular phenotypes and (**c**) biomarkers depending on the genotype of rs1967309 (genotypic encoding). The p-values $p_{itx}$ reported come from a likelihood ratio test comparing models with and without the three-way interaction term between the two SNPs and sex. Sex-combined results using GTEx cardiovascular phenotype data are shown in *Figure 6—figure supplement 2*. See Appendix 1—table 2 for the list of abbreviations.

The online version of this article includes the following figure supplement(s) for figure 6:

**Source data 1.** Results of the interaction between rs1967309 and rs158477 on phenotypes in the UK biobank.

**Source data 2.** Results for the cardiovascular phenotypes and biomarkers by sex and by rs1967309 genotypes in the UK biobank.

**Figure supplement 1.** Single SNP effects of rs1967309 and rs158477 on phenotypes in the UK biobank.

**Figure supplement 2.** Epistatic association of rs1967309 and rs158477 on cardiovascular disease in GTEx.

genotypic association between *ADCY9* and *CETP* in two Peruvian cohorts, specifically between rs1967309 and rs158477, which was also seen in the Native population of the Andes. The interaction between the two SNPs was found to be nominally significant for respiratory and cardiovascular disease outcomes (*Figure 6*, *Figure 6—figure supplement 2*). Additionally, a nominally significant epistatic interaction was seen on *CETP* expression in many tissues, including the hippocampus and hypothalamus in the brain. Despite brain tissues not displaying the highest *CETP* expression levels, CETP that is synthesized and secreted in the brain could play an important role in the transport and the redistribution of lipids within the central nervous system (*Albers et al., 1992*; *Yamada et al., 1995*) and has been associated with Alzheimer's disease risk (*Murphy et al., 2012*; *Oestereich et al., 2020*). These findings reinforce the fact that the SNPs are likely functionally interacting, but extrapolating on the specific phenotypes under selection from these results is not straight forward. Identifying the phenotype and environmental pressures that may have caused the selection signal is complicated by the fact that the UK Biobank participants, on which the marginally significant associations have been found, do not live in the same environment as Peruvians. In Andeans from Peru, selection in response to hypoxia at high altitude was proposed to have effects on the cardiovascular system (*Crawford et al., 2017*). The hippocampus functions are perturbed at high altitude (eg. deterioration of memory *Lieberman et al., 2005*; *Shukitt-Hale et al., 1994*), whereas the hypothalamus regulates the autonomic nervous system (ANS) and controls the heart and respiratory rates (*Horiuchi et al., 2009*; *Rahmouni, 2016*), phenotypes which are affected by hypoxia at high altitude (*Bärtsch and Gibbs, 2007*; *Hainsworth et al., 2007*). Furthermore, high altitude-induced hypoxia (*Bigham and Lee, 2014*; *Moore, 2017b*) and cardiovascular system disturbances (*Abe et al., 2017*; *Lee et al., 2019*) have been shown to be associated in several studies (*Faeh et al., 2009*; *Naeije, 2010*; *Ostadal and Kolar, 2007*; *Riley and Gavin, 2017*; *Savla et al., 2018*), thus potentially sharing common biological pathways. Therefore, our working hypothesis is that selective pressures on our genes of interest in Peru are linked to the physiological response to high-altitude, which might be the environmental driver of coevolution.

The significant interaction effects on *CETP* expression vary between sexes in amplitude and direction, with most signals driven by male samples, but significant interaction effects observed in females only, despite sample sizes being consistently lower than for males. Notably, in the tibial artery and heart atrial appendage, two tissues directly relevant to the cardiovascular system, the female-specific interaction effect on *CETP* expression is reversed between rs1967309 genotypes AA and GG, compared to the effects seen in males in skin and brain tissues. Given our *ADCY9-KD* were done in liver cell lines from male donors, future work to fully understand how rs1967309 and rs158477 interact will focus on additional experiments in cells from both male and female donors in these relevant tissues. In a previous study, we showed that inhibition of both *Adcy9* and *CETP* impacted many phenotypes linked to the ANS in male mice (*Rautureau et al., 2018*), but in the light of our results, these experiments should be repeated in female mice. The function of ANS is important in a number of pathophysiological states involving the cardiovascular system, like myocardial ischemia and cardiac arrhythmias, with significant sex differences reported (*Abhishekh et al., 2013*; *Dart et al., 2002*; *Nugent et al., 2011*).

The interaction effect between the *ADCY9* and *CETP* SNPs on both respiratory and cardiovascular phenotypes differs between the sexes, with effects on respiratory phenotypes limited to females (*Figure 6a*) and cardiovascular disease phenotype associations showing significant three-way sex-by-SNPs effects (*Figure 6*). Furthermore, the LRLD signal is present mainly in males (*Figure 4*), although the genotype association is also seen in female for a different genotype combination, suggesting the presence of sex-specific selection. This type of selection is very difficult to detect, especially on autosomes, with very few empirical examples found to date in the human genome despite strong theoretical support of their occurrence (*Morrow and Connallon, 2013*). However, sexual dimorphism in gene expression between males and females on autosomal genes has been linked to evolutionary pressures (*Connallon and Clark, 2010*; *Parsch and Ellegren, 2013*; *Williams and Carroll, 2009*), possibly with a contribution of epistasis. As the source of selection, we favor the hypothesis of differential survival over differential ability to reproduce, because the genetic combination between *ADCY9* and *CETP* has high chances to be broken up by recombination at each generation. Even in the case where recombination is suppressed in males between these loci, they would still have equal chances to pass the favored combination to both male and female offspring, which would not explain the sex-specific LRLD signal. We see an enrichment for the rs1967309-AA+ rs158477 GG in males and rs1967309-AA+ rs158477 AA in females, which are

the beneficial combination for CAD in the corresponding sex, possibly pointing to a sexually antagonistic selection pressure, where the fittest genotype combination depends on the sex.

Such two-gene selection signature, where only males show strong LRLD, can happen if a specific genotype combination is beneficial in creating males (through differential gamete fitness or in utero survival, for example) or if survival during adulthood is favored with a specific genotype combination compared to other genotypes. In the case of age-dependent differential survival, the genotypic association is expected to be weaker at younger ages, however the LRLD signal between rs1967309 and rs158477 in the LIMAA cohort did not depend on age neither in males nor in females (Appendix 1). Since very few individuals were younger than 20 years old, it is likely that the age range in this cohort is not appropriate to distinguish between the two possibilities. This age-dependent survival therefore remains to be tested in comparison with pediatric cohorts of Peruvians: if the LRLD signal is absent in newborns for example, it will suggest a strong selective pressure acts early in life on boys. To specifically test the in utero hypothesis, a cohort of stillborn babies with genetic information could allow to evaluate if the genotype combination is more frequent in these. Lastly, it may be that the evolutionary pressure is linked to the sex chromosomes (*Cox et al., 2017*; *McGlothlin et al., 2019*), and a three-way interaction between *ADCY9*, *CETP* and Y chromosome haplotypes or mitochondrial haplogroups remains to be explored.

Even though we observed the LRLD signal between rs1967309 and rs158477 in two independent Peruvian cohorts, reducing the likelihood that our result is a false positive, one limitation is that the individuals were recruited in the same city (Lima) in both cohorts. However, we show that both populations are heterogeneous with respect to ancestry (*Appendix 1—figure 2*), suggesting that they likely represent accurately the Peruvian population. As recent admixture and population structure can strongly influence LRLD, we performed several analyses to consider these confounders, in the full cohorts and in the sex-stratified analyses. All analyses were robust to genome-wide and local ancestry patterns, such that our results are unlikely to be explained by these effects alone (Appendix 1). Unfortunately, we could not use expression and phenotypic data from Peruvian individuals, which makes all the links between the selection pressures and the phenotype associations somewhat indirect. Future studies should focus on evaluating the phenotypic impact of the interaction specifically in Peruvians individuals, in cohorts such as the Population Architecture using Genomics and Epidemiology (PAGE) (*Wojcik et al., 2019*), in order to confirm the marginally significant associations found in European cohorts. Indeed, the Peruvian/Andean genomic background could be of importance for the interaction effect observed in this population, which reduces the power of discovery in individuals of unmatched ancestry. Furthermore, not much is known about the strength of this type of selection, and simulations would help evaluate how strong selection would need to be in a single generation to produce this level of LRLD. Another limitation is the low number of samples per tissue in GTEx and the cell composition heterogeneity per tissue and per sample (*Battle et al., 2017*), which can be partially captured by PEER factors and can modulate the eQTL effects. Therefore, our power to detect tissue-specific interaction effects is reduced in this dataset, making it quite remarkable that we were able to observe multiple nominally significant interaction effects between the loci.

Despite these limitations, our results support a functional role for the *ADCY9* intronic SNP rs1967309, likely involved in a molecular mechanism related to *CETP* expression, but this mechanism seems to implicate sex as a modulator in a tissue-specific way, which complicates greatly its understanding. In the dal-OUTCOMES clinical trial, the partial inhibitor of CETP, dalcetrapib, did not decrease the risk of cardiovascular outcomes in the overall population, but rs1967309 in the *ADCY9* gene was associated to the response to the drug, which benefitted AA individuals (*Tardif et al., 2015*). Interestingly, rs1967309 AA is found in both the male and female beneficial combinations of genotypes for CAD, the same that are enriched in Peruvians, but without taking rs158477 and sex into account, this association was masked. The modulation of *CETP* expression by rs1967309 could impact CETP's functions that are essential for successfully reducing cardiovascular events. The rs158477 locus could be a key player for these functions, and dalcetrapib may be mimicking its impact, hence explaining the pharmacogenomics association. Furthermore, in the light of our results, some of these effects could differ between men and women (*Metzinger et al., 2020*), which may need to be taken into consideration in the future precision medicine interventions potentially implemented for dalcetrapib.

In conclusion, we discovered a putative epistatic interaction between the pharmacogene *ADCY9* and the drug target gene *CETP*, that appears to be under selection in the Peruvian population. Our approach exemplifies the potential of using evolutionary analyses to help find relationships between

pharmacogenes and their drug targets. We characterized the impact of the *ADCY9/CETP* interaction on a range of phenotypes and tissues. Our gene expression results in brain tissues suggest that the interaction could play a role in protection against challenges to the nervous system caused by stress such as hypoxia. The female-specific eQTL interaction results in arteries and heart tissues further suggest a link with the cardiovascular system, and the phenotype association results support further this hypothesis. In light of the associations between high altitude-induced hypoxia and cardiovascular system changes, the interaction identified in this study could be involved in both systems: for example, ADCY9 and CETP could act in pathways involved in adaptation to high altitude, which could influence cardiovascular risk via their interaction in a sex-specific manner. Finally, our findings of an evolutionary relationship between *ADCY9* and *CETP* during recent human evolution points towards a biological link between dalcetrapib's pharmacogene *ADCY9* and its therapeutic target *CETP*.

# Materials and methods

## Key resources table

| Reagent type (species) or resource | Designation | Source or reference | Identifiers | Additional information |
|---|---|---|---|---|
| Gene (*Homo sapiens*) | CETP | GenBank | HGNC:1,869 | |
| Gene (*Homo sapiens*) | ADCY9 | GenBank | HGNC:240 | |
| Cell line (*Homo sapiens*) | HepG2 | ATCC | RRID:CVCL_0027 | Hepatoblastoma |
| Recombinant DNA reagent | pEZ-M46-AC9 plasmid | GeneCopoeia | EX-H0609-M46 | Methods section |
| Recombinant DNA reagent | pEZ-M50-CETP plasmid | GeneCopoeia | EX-C0070-M50 | Methods section |
| Antibody | Anti-CETP (rabbit monoclonal) | Abcam | #ab157183 | (1:1000) in 3 % BSA, TBS, tween 20 0.5%, O/N 4 °C |
| Antibody | Goat anti-rabbit antibody (goat polyclonal) | Abcam | RRID:AB_955447 | (1:10 000) in 3 % BSA 1 h at room temperature |
| Sequence-based reagent | Human CETP_F | IDT Technologies | PCR primers | CTACCTGT CTTTCCATAA |
| Sequence-based reagent | Human CETP_R | IDT Technologies | PCR primers | CATGATGT TAGAGATGAC |
| Sequence-based reagent | Human ADCY9_F | IDT Technologies | PCR primers | CTGAGGTT CAAGAACATCC |
| Sequence-based reagent | Human ADCY9_R | IDT Technologies | PCR primers | TGATTAATG GGCGGCTTA |
| Sequence-based reagent | Silencer Select siRNA against human ADCY9 | Ambion | Cat. #4390826 ID 1039 | CCUGAUGA AAGAUUACUU Utt |
| Sequence-based reagent | Silencer Select siRNA against human CETP | Ambion | Cat. #4392420 ID 2933 | GGACAGAUC UGCAAAGAGAtt |
| Sequence-based reagent | Negative Control siRNA | Ambion | Cat. #4390844 | |
| Commercial assay or kit | Lipofectamine RNAiMAX reagent | Invitrogen | Cat. #13,778 | |
| Commercial assay or kit | Lipofectamine 2000 reagent | Invitrogen | Cat. #11668–019 | |
| Commercial assay or kit | RNeasy Plus Mini Kit | Qiagen | Cat. #74,136 | |
| Commercial assay or kit | High-Capacity cDNA Reverse Transcription Kit | Applied Biosystems | Cat. #4368814 | |
| Commercial assay or kit | Agilent RNA 6000 Nano Kit for Bioanalyzer 2,100 System | Agilent Technologies | Cat. #5067–1511 | |
| Commercial assay or kit | SYBR-Green reaction mix | BioRad | Cat. #1725274 | |
| Commercial assay or kit | Amicon Ultra 0.5 ml 10 kDa cutoff units | Millipore Sigma | Cat. #UFC501096 | |

*Continued on next page*

*Continued*

| Reagent type (species) or resource | Designation | Source or reference | Identifiers | Additional information |
|---|---|---|---|---|
| Commercial assay or kit | Western Lightning ECL Pro | Perkin Elmer | Cat. #NEL122001EA | |
| Commercial assay or kit | TGX Stain-Free FastCast Acrylamide 10% | BioRad | Cat# 1610183 | |
| Software, algorithm | TrimGalore! | DOI:10.14806/ej.17.1.200 | RRID:SCR_011847 | |
| Software, algorithm | STAR (v.2.6.1a) | DOI:10.1093/bioinformatics/bts635 | RRID:SCR_019993 | |
| Software, algorithm | RSEM (v.1.3.1) | DOI:10.1186/1471-2105-12-323 | RRID:SCR_013027 | |
| Software, algorithm | R statistical software (v.3.6.0/v.3.6.1) | https://www.r-project.org/ | RRID:SCR_001905 | |
| Software, algorithm | FlashPCA2 | DOI:10.1093/bioinformatics/btx299 | RRID:SCR_021680 | |
| Software, algorithm | Vcftools (v.0.1.17) | DOI:10.1093/bioinformatics/btr330 | RRID:SCR_001235 | |
| Software, algorithm | RFMix (v.2.03) | DOI:10.1016 /j.ajhg.2013.06.020 | | |
| Software, algorithm | PEER | DOI:10.1038/nprot.2011.457 | RRID:SCR_009326 | |
| Software, algorithm | pyGenClean (v.1.8.3) | DOI:10.1093/bioinformatics/btt261 | | |
| Software, algorithm | SAS (v.9.4) | https://www.sas.com/en_us/software/stat.html | RRID:SCR_008567 | |
| Software, algorithm | EPO pipeline (version e59) | DOI:10.1093/database/bav096 | | |
| Software, algorithm | Bcftools (v.1.9) | DOI:10.1093/bioinformatics/btr509 | RRID:SCR_005227 | |
| Software, algorithm | Genotype Harmonizer (v.1.4.20) | DOI:10.1186/1756-0500-7-901 | | |
| Software, algorithm | Hapbin (v.1.3.0) | DOI:10.1093/molbev/msv172 | | |
| Software, algorithm | SHAPEIT2 (r.837) | DOI:10.1038/nmeth.1785 | | |
| Software, algorithm | PBWT | DOI:10.1093/bioinformatics/btu014 | | |
| Software, algorithm | Beacon designer software (v.8) (Premier Biosoft) | http://www.premierbiosoft.com/qOligo/Oligo.jsp?PID=1 | | |
| Other | 1000 Genomes project | DOI:10.1038/nature15393 | RRID:SCR_006828 | |
| Other | LIMAA | DOI:10.1038 /s41467-019-11664-1 | dbGAP:phs002025.v1.p1 | dbgap project #26,882 |
| Other | Native American genetic dataset | DOI:10.1038/nature11258 | | |
| Other | GEUVADIS | DOI:10.1038/nature12531 | RRID:SCR_000684 | |
| Other | GTEx (v8) | DOI:10.1038 /ng.2653 | RRID:SCR_013042 | dbgap project #19,088 |
| Other | CARTaGENE biobank | DOI:10.1093/ije/dys160 | RRID:SCR_010614 | CAG project number 406,713 |
| Other | UK biobank | DOI:10.1371/journal.pmed.1001779 | RRID:SCR_012815 | UKB project #15,357 and #20,168 |
| Other | Sanger Imputation Server | DOI:10.3389/fgene.2019.00034 | | |

## Population genetics datasets

The whole-genome sequencing data from the 1000 Genomes project (1000G) Phase III dataset (http://ftp.1000genomes.ebi.ac.uk/vol1/ftp/release/20130502/) was filtered to exclude INDELs and CNVs so that we kept only biallelic SNPs. This database has genomic variants of 2504 individuals across five ancestral populations: Africans (AFR, n = 661), Europeans (EUR, n = 503), East Asians (EAS, n = 504), South Asians (SAS, n = 489), and Americans (AMR, n = 347) (*Auton et al., 2015*). The replication dataset,

LIMAA, has been previously published (*Asgari et al., 2020*; *Luo et al., 2019*) and was accessed through dbGaP [phs002025.v1.p1, dbgap project #26,882]. This cohort was genotyped with a customized Affymetric LIMAAray containing markers optimized for Peruvian-specific rare and coding variants. We excluded related individuals as reported previously (*Asgari et al., 2020*), resulting in a final dataset of 3,509 Peruvians. We also identified fine-scale population structure in this cohort and a more homogeneous subsample of 3243 individuals (1302 females and 1941 males) in this cohort was kept for analysis (*Table 1*, Appendix 1). The Native American genetic dataset (NAGD) contains 2351 individuals from Native descendants from the data from a previously published study (*Reich et al., 2012*). Individuals were separated by their linguistic families identified by Reich and colleagues (*Reich et al., 2012*). NAGD came under the Hg18 coordinates, so a lift over was performed to transfer to the Hg19 genome coordinates. Pre-processing details for these datasets are described in Appendix 1.

## eQTL datasets

We used several datasets (*Table 1*) for which we had both RNA-seq data and genotyping. First, the GEUVADIS dataset (*Lappalainen et al., 2013*) for 1000 G individuals was used (available at https://www.internationalgenome.org/data-portal/data-collection/geuvadis). A total of 287 non-duplicated European samples (CEU, GBR, FIN, TSI) were kept for analysis. Second, the Genotype-tissue Expression v8 (GTEx) (*GTEx Consortium, 2013*) was accessed through dbGaP (phs000424.v8.p2, dbgap project #19088) and contains gene expression across 54 tissues and 948 donors, genetic and phenotypic information. Phenotype analyses are described in Appendix 1. The cohort contains mainly of European descent (84.6%), aged between 20 and 79 years old. Analyses were done on 699 individuals, 66 % of males and 34 % of females (*Appendix 1—figure 10a*). Third, we used the data from the CARTaGENE biobank (*Awadalla et al., 2013*) (CAG project number 406713) which includes 728 RNA-seq whole-blood samples with genotype data, from individuals from Quebec (Canada) aged between 36 and 72 years old (*Appendix 1—figure 10b*). Genotyping and RNA-seq data processing pipelines for these datasets are detailed in Appendix 1. To quantify *ADCY9* gene expression, we removed the isoform transcript ENST00000574721.1 (ADCY9-205 from the Hg38) from the Gene Transfer Format (GTF) file because it is a "retained intron" and accumulates genomic noise (Appendix 1), masking true signals for *ADCY9*. To take into account hidden factors, we calculated PEER factors (*Stegle et al., 2012*) on the normalized expressions, on all samples and stratified by sex (sPEER factors). To detect eQTL effects, we performed a two-sided linear regression on *ADCY9* and *CETP* expressions using R (v.3.6.0) (https://www.r-project.org/) with the formula $lm\,(p \sim rs1967309 * rs158477 + Covariates)$ for evaluating the interaction effect, $lm\,(p \sim rs1967309 + rs158477 + Covariates)$ for the main effect of the SNPs and $lm\,(p \sim rs1967309 * rs158477 * sex + Covariates)$ for evaluating the three-way interaction effect. Under the additive model, each SNP is coded by the number of non-reference alleles (G for rs1967309 and A for rs158477), under the genotypic model, dummy coding is used with homozygous reference genotype set as reference. The covariates include the first 5 Principal Components (PCs), age (except for GEUVADIS, information not available), sex, as well as PEER factors. We tested the

**Table 1.** Cohort information.
Sample sizes are reported after quality control steps.

| Cohort/Subpopulation | Abbreviation | Ethnicity | Sample size (% female) | Age | Reference |
|---|---|---|---|---|---|
| 1000 G – Peruvian | PEL* | Peruvian | 85 (52%) | NA | *Auton et al., 2015* |
| LIMAA/Peruvian | LIMAA | Peruvian | 3,243 (40%) | 29.6 ± 13.8 | *Asgari et al., 2020*; *Luo et al., 2019* |
| Native Amerind/Andean | NAGD/AND | Amerind/Peruvian | 88 (40%) | NA | *Reich et al., 2012* |
| GEUVADIS | GEUVADIS* | European descent | 287 (54%) | NA | *Lappalainen et al., 2013* |
| CARTaGENE | CaG | European descent | 728 (51%) | 53.6 ± 8.7 | *Awadalla et al., 2013* |
| GTEx | GTEx | European descent | 699 (34%) | 52.6 ± 13.1 | *GTEx Consortium, 2013* |
| UK biobank | Ukb* | European descent | 413,138 (54%) | 56.8 ± 8.0 | *Sudlow et al., 2015* |

*indicates a discovery cohort.
NA: not available.

robustness of our results to the inclusion of different numbers of PEER factors in the models and we report them all for GEUVADIS, CARTaGENE and GTEx (*Appendix 1—figures 7–9*). Reported values in the text are for 5 PEER factors in GEUVADIS, 10 PEER factors in CARTaGENE, 25 sPEER for skin sun exposed in male and 10 sPEER for artery tibial in female in GTEx. Covariates specific to each cohort are reported in Appendix 1.

## UK biobank processing and selected phenotypes

The UK biobank (*Sudlow et al., 2015*) contains 487,392 genotyped individuals from the UK still enrolled as of August 20th 2020, imputed using the Haplotype Reference Consortium as the main reference panel, and accessed through project #15,357 and UKB project #20,168. Additional genetic quality control was done using pyGenClean (v.1.8.3) (*Lemieux Perreault et al., 2013*). Variants or individuals with more than 2 % missing genotypes were filtered out. Individuals with discrepancies between the self-reported and genetic sex or with aneuploidies were removed from the analysis. We considered only individuals of European ancestry based on PCs, as it is the largest population in the UK Biobank, and because ancestry can be a confounder of the genetic effect on phenotypes. We used the PCs from UK Biobank to define a region in PC space using individuals identified as 'white British ancestry' as a reference population. Using the kinship estimates from the UK Biobank, we randomly removed individuals from kinship pairs where the coefficient was higher than 0.0884 (corresponding to a third-degree relationship). The resulting post QC dataset included 413,138 individuals. For the reported phenotypes, the date of baseline visit was between 2006 and 2010. The latest available hospitalization records discharge date was June 30th 2020 and the latest date in the death registries was February 14th 2018. We used algorithmically defined cardiovascular outcomes based on combinations of operation procedure codes (OPCS) and hospitalization or death record codes (ICD9/ICD10). A description of the tested continuous variables can be found in *Appendix 1—table 2*. We used age at recruitment defined in variable #21,022 and sex in variable #31. We ignored self-reported events for cardiovascular outcomes as preliminary analyses suggested they were less precise than hospitalization and death records.

In association models, each SNP analyzed is coded by the number of non-reference alleles, G for rs1967309 and A for rs158477. SNP rs1967309 was also coded as a genotypic variable, to allow for non-additive effects. For continuous traits (*Appendix 1—table 2*) in the UK Biobank, general two-sided linear models (GLM) were performed using SAS software (v.9.4). A GLM model was first performed using the covariates age, sex and PCs 1–10. The externally studentized residuals were used to determine the outliers, which were removed. The normality assumption was confirmed by visual inspection of residuals for most of the outcomes, except *birthwt* and *sleep*. For biomarkers and cardiovascular endpoints, regression analyses were done in R (v.3.6.1). Linear regression analyses were conducted on standardized outcomes and logistic regression was used for cardiovascular outcomes. Marginal effects were calculated using margins package in R. In both cases, models were adjusted for age at baseline and top 10 PCs, as well as sex when not stratified. In models assessing two-way (rs1967309 by rs158477) or three-way (rs1967309 by rs158477 by sex) interactions, we used a 2 d.f. likelihood ratio test for the genotypic dummy variables' interaction terms (genotypic model) (Appendix 1).

## RNA-sequencing of ADCY9-knocked-down Hepg2 cell line

The human liver hepatocellular HepG2 cell line was obtained from ATCC, a cell line derived from the liver tissue of a 15-year-old male donor (*López-Terrada et al., 2009*). Our cells tested negatively for mycoplasma contamination and have a morphology and expression profile concordant with this cell type. Cells were cultured in EMEM Minimum essential Medium Eagle's, supplemented with 10 % fetal bovine serum (Wisent Inc). A total of 250,000 cells in 2 ml of medium in a six-well plate were transfected using 12.5 pmol of Silencer Select siRNA against human ADCY9 (Ambion cat # 4390826 ID 1039), Silencer Select siRNA against CETP (Ambion cat 4392420 ID 2933) or Negative Control siRNA (Ambion cat #4390844) with 5 µl of Lipofectamine RNAiMAX reagent (Invitrogen cat #13778) in 500 µl Opti-MEM I reduced serum medium (Invitrogen cat # 31985) for 72 hr (*Appendix 1—table 3*, Appendix 1). The experiment was repeated five times at different cell culture passages. Total RNA was extracted from transfected HepG2 cells using RNeasy Plus Mini Kit (Qiagen cat #74136) in accordance with the manufacturer's recommendation. Preparation of sequencing library and sequencing was performed at the McGill University Innovation Center. Briefly, ribosomal RNA was depleted using

NEBNext rRNA depletion kit. Sequencing was performed using Illumina NovaSeq 6000 S2 paired end 100 bp sequencing lanes. Basic QC analysis of the 10 samples was performed by the Canadian Centre for Computational Genomics (C3G). To process the RNA-seq samples, we first performed read trimming and quality clipping using TrimGalore! (*Martin, 2011*; *Krueger et al., 2021*), we aligned the trimmed reads on the Hg38 reference genome using STAR (v.2.6.1a) and we ran RSEM (v.1.3.1) on the transcriptome aligned libraries. Prior to normalization with limma and voom, we filtered out genes which had less than six reads in more than 5 samples. For *ADCY9* and *CETP* gene-level differential expression analyses, we compared the mean of each group of replicates with a t-test for paired samples. The transcriptome-wide differential expression analysis was done using limma, on all genes having an average of at least 10 reads across samples from a condition. Samples were paired in the experiment design. The multiple testing was taken into account by correcting the p-values with the qvalue R package (v.4.0.0) (*Storey, 2002*), to obtain transcriptome-wide FDR values.

## Overexpression of *ADCY9* and *CETP* genes in HepG2 cell line

For *ADCY9* and *CETP* overexpression experiments, 500,000 cells in 2 ml of medium in a six-well plate were transfected using 1 μg of pEZ-M46-AC9 or pEZ-M50-CETP plasmids (GeneCopoeia) with 5 μl of Lipofectamine 2000 reagent (Invitrogen cat # 11668–019) for 72 hr. Total RNA was extracted from transfected HepG2 cells using RNeasy Plus Mini Kit (Qiagen cat #74136) in accordance with the manufacturer's recommendation (*Appendix 1—table 3*, Appendix 1).

## Natural selection analyses

We used the integrated Haplotype Statistic (iHS) (*Voight et al., 2006*) and the population branch statistic (PBS) (*Auton et al., 2015*) to look for selective signatures. The iHS values were computed for the each 1000G population. An absolute value of iHS above two is considered to be a genome wide significant signal (*Voight et al., 2006*). Prior to iHS computation, ancestral alleles were retrieved from six primates using the EPO pipeline (version e59) (*Herrero et al., 2016*) and the filtered 1000 Genomes vcf files were converted to change the reference allele as ancestral allele using bcftools (*Li et al., 2009*) with the fixref plugin. The hapbin program (v.1.3.0) (*Maclean et al., 2015*) was then used to compute iHS using per population-specific genetic maps computed by Adam Auton on the 1000G OMNI dataset (http://ftp.1000genomes.ebi.ac.uk/vol1/ftp/technical/working/20130507_omni_recombination_rates/). When the genetic map was not available for a subpopulation, the genetic map from the closest sub-population was selected according to their global FST value computed on the phase three dataset.

We scanned the *ADCY9* and *CETP* genes using the population branch statistic (PBS), using 1000G sub-populations data. PBS summarizes a three-way comparison of allele frequencies between two closely related populations, and an outgroup. The grouping we focused on was PEL/MXL/CHB, with PEL being the focal population to test if allele frequencies are especially differentiated from those in the other populations. The CHB population was chosen as an outgroup to represent a Eurasian population that share common ancestors in the past with the American populations, after the out-of-Africa event. Using PJL (South Asia) and CEU (Europe) as an outgroup, or CLM as a closely related population (instead of MXL) yield highly similar results. To calculate $F_{ST}$ for each pair of population in our tree, we used vcftools (*Danecek et al., 2011*) by subpopulation. We calculated normalized PBS values as in *Crawford et al., 2017*, which adjusts values for positions with large branches in all populations, for the whole genome. We use this distribution to define an empirical threshold for significance based on the 95th percentile of all PBS values genome-wide for each of the three populations.

## Long-range linkage disequilibrium

Long-range linkage disequilibrium (LRLD) was calculated using the function geno-r2 of vcftools (v.0.1.17) which uses the genotype frequencies. LRLD was evaluated in all subpopulations from 1000 Genomes Project Phase III, in LIMAA and NAGD, for all biallelic SNPs in *ADCY9* (chr16:4,012,650–4,166,186 in Hg19 genome reference) and *CETP* (chr16:56,995,835–57,017,756 in Hg19 genome reference). We analyzed loci from the phased VCF files that had a MAF of at least 5 % and a missing genotype of at most 10%, in order to retain a maximum of SNPs in NAGD which has higher missing rates than the others. We extracted the 99th percentile of all pairs of comparisons between *ADCY9* and *CETP* genes to use as a threshold for empirical significance and we refer to these as *ADCY9/CETP*

empirical p-values. In LIMAA, we also evaluated the genotypic association using a $\chi^2$ test with four degrees of freedom ($\chi_4^2$) using a permutation test, as reported in *Rohlfs et al., 2010* (Appendix 1).

Furthermore, for both cohorts, we created a distribution of LRLD values for random pairs of SNPs across the genome to obtain a genome-wide null distribution of LRLD to evaluate how unusual the genotypic association between our candidate SNPs (rs1967309-rs158477) is while taking into account the cohort-specific background genomic noise/admixture and allele frequencies. We extracted 3513 pairs of SNPs that match rs1967309 and rs158477 in terms of MAF, physical distance (in base pairs) and genetic distance (in centiMorgans (cM), based on the PEL genetic map) between them in both cohorts (Appendix 1), and report genome-wide empirical p-values based on this distribution. For the analyses of LRLD between *ADCY9* and *CETP* stratified by sex, we considered the same set of SNP pairs that we used for the full cohorts, but separated the dataset by sex before calculating the LRLD values. To evaluate how likely the differences observed in LRLD between sex are, we also performed permutations of the sex labels across individuals to create a null distribution of sex-specific effects (Appendix 1).

### Local ancestry inference

To evaluate local ancestry in the PEL subpopulation and in the LIMAA cohort, we constructed a reference panel using the phased haplotypes from 1000 Genomes (YRI, CEU, CHB) and the phased haplotypes of NAGD (Northern American, Central American and Andean) (Appendix 1). We kept overlapping positions between all datasets, and when SNPs had the exact same genetic position, we kept the SNP with the highest variance in allele frequencies across all reference populations (Appendix 1). We ran RFMix (v.2.03) (*Maples et al., 2013*) (with the option 'reanalyze-reference' and for 25 iterations) on all phased chromosomes. We estimated the whole genome average proportion of each ancestry using a weighted mean of the chromosome specific proportions given by RFMix based on the chromosome size in cM. For comparing the overall Andean enrichment inferred by RFMix between rs1967309/rs158477 genotype categories, we used a two-sided Wilcoxon -t-test. To evaluate the Andean local ancestry enrichment specifically at *ADCY9* and *CETP*, we computed the genome-wide 95th percentile for proportion of Andean attribution for all intervals given by RFMix.

### Code and source data

Numerical summary data represented as a graph in main figures, as well as the code to reproduce figures and analyses, can be found here: *Gamache, 2021*. Raw RNA sequencing data for knocked down experiments in hepatocyte HepG2 cells are deposited the data on NCBI Gene Expression Omnibus, accession number GSE174640.

## Acknowledgements

We thank all members of the Hussin lab for their constructive comments and feedback throughout this project, as well as the insightful input from three reviewers and reviewing and senior editors at eLife. This work was completed thanks to computational resources provided by Compute Canada clusters Graham and Beluga. This work was funded by the *Institut de Valorisation des Données* (IVADO), Health Collaboration Acceleration Fund from the Ministère de l'Économie et de l'Innovation of the Government of Quebec and the Montreal Heart Institute (MHI) Foundation. JGH is a *Fonds de la Recherche en Santé* (FRQS) Junior one fellow. IG receives a PhD scholarship from the MHI Foundation and MAL holds a PhD scholarship from Canadian Institutes of Health Research. MPD holds the Canada Research Chair in Precision Medicine Data Analysis. JCT holds the Canada Research Chair in Personalized Medicine and the Université de Montréal endowed research chair in atherosclerosis.

## Additional information

### Competing interests

Jean-Claude Tardif: reports grants from Government of Quebec, National Heart, Lung, and Blood Institute of the U.S. National Institutes of Health (NIH), the MHI Foundation, from Bill and Melinda Gates Foundation, Amarin, Esperion, Ionis, Servier, RegenXBio; personal fees from Astra Zeneca, Sanofi, Servier; and personal fees and minor equity interest from Dalcor. Has a patent (US20190070178A1)

Methods for Treating or Preventing Cardiovascular Disorders and Lowering Risk of Cardiovascular Events issued to Dalcor, no royalties received, a patent (US20170233812A1) Genetic Markers for Predicting Responsiveness to Therapy with HDL-Raising or HDL Mimicking Agent issued to Dalcor, no royalties received, and a patent (US Provisional Applications No. 62/935,751 and 62/935,865) Methods for using low dose colchicine after myocardial infarction with royalties paid to Invention assigned to the Montreal Heart Institute. Marie-Pierre Dubé: has a patent (US20190070178A1) Methods for Treating or Preventing Cardiovascular Disorders and Lowering Risk of Cardiovascular Events issued to Dalcor, no royalties received, a patent (US20170233812A1) Genetic Markers for Predicting Responsiveness to Therapy with HDL-Raising or HDL Mimicking Agent issued to Dalcor, no royalties received, and a patent (US Provisional Applications No. 62/935,751 and 62/935,865) Methods for using low dose colchicine after myocardial infarction with royalties paid to Invention assigned to the Montreal Heart Institute. M.P.D. reports personal fees and other from Dalcor and personal fees from GlaxoSmithKline, other from AstraZeneca, Pfizer, Servier, Sanofi.. Julie Hussin: has received speaker honoraria from Dalcor and District 3 Innovation Centre. The other authors declare that no competing interests exist.

## Funding

| Funder | Grant reference number | Author |
|---|---|---|
| Institut de Cardiologie de Montréal | | Isabel Gamache<br>Marc-André Legault<br>Jean-Christophe Grenier<br>Rocio Sanchez<br>Eric Rhéaume<br>Holly Trochet<br>Jean-Claude Tardif<br>Marie-Pierre Dubé<br>Julie Hussin |
| Université de Montréal | | Isabel Gamache |
| Canadian Institutes of Health Research | | Marc-André Legault |
| Canada Research Chairs | | Jean-Claude Tardif<br>Marie-Pierre Dubé |
| Fonds de Recherche du Québec - Santé | | Julie Hussin |
| Institut de Valorisation des Données IVADO | | Isabel Gamache<br>Jean-Christophe Grenier<br>Julie Hussin |
| Ministère de l'Économie, de la Science et de l'Innovation - Québec | Health collaboration acceleration fund | Jean-Claude Tardif<br>Marie-Pierre Dubé |

The funders had no role in study design, data collection and interpretation, or the decision to submit the work for publication.

## Author contributions

Isabel Gamache, Conceptualization, Data curation, Formal analysis, Investigation, Methodology, Software, Validation, Writing – original draft, Writing – review and editing; Marc-André Legault, Samira Asgari, Amina Barhdadi, Yassamin Feroz Zada, Data curation, Formal analysis; Jean-Christophe Grenier, Data curation, Methodology, Resources, Software, Writing – original draft; Rocio Sanchez, Formal analysis, Investigation, Resources, Validation; Eric Rhéaume, Investigation, Project administration, Resources; Holly Trochet, Software; Yang Luo, Leonid Lecca, Megan Murray, Soumya Raychaudhuri, Resources; Jean-Claude Tardif, Conceptualization, Funding acquisition, Resources; Marie-Pierre Dubé, Conceptualization, Funding acquisition, Project administration, Resources, Supervision; Julie Hussin, Conceptualization, Funding acquisition, Methodology, Project administration, Resources, Supervision, Writing – original draft, Writing – review and editing

## Author ORCIDs

Isabel Gamache (ID) http://orcid.org/0000-0002-0613-0979
Marie-Pierre Dubé (ID) http://orcid.org/0000-0001-8442-4393

Julie Hussin ⬛ http://orcid.org/0000-0003-4295-3339

### Ethics

Human subjects: The analyses done in this study were approved by the different cohorts used. Participants in these cohorts gave their general consent for their data to be used for research purposes. All individual-level data was anonymized and no efforts were made by the authors to deanonymize or recontact any of the participants from the cohorts, in keeping with our agreements with the UK Biobank, CARTaGENE, dbGAP, 1000Genomes and the Native American Genetics dataset (Universidad de Antioquia).

### Decision letter and Author response

Decision letter https://doi.org/10.7554/eLife.69198.sa1
Author response https://doi.org/10.7554/eLife.69198.sa2

---

## Additional files

### Supplementary files

• Transparent reporting form

### Data availability

The 1000 Genomes Project, GEUVADIS is freely available. The Native American genetic dataset was shared to us upon request to the authors of the initial paper and through a data access agreement with Universidad de Antioquia (Prof. Omar Triana Chavez). We contacted bedoya.g@gmail.com and a.ruizlin@ucl.ac.uk to get access to the dataset and we completed a data access application form and signed a data access approval once approved. Applications for access to these data can be submitted at any time. These are considered on a rolling basis and a decision was given within 1 month of receipt. PhD student applicants must include their supervisors as a co-applicant and provide their full contact details. A publication list must be provided for the applicant, co-applicants and PhD supervisors where PhD students have applied to provide proof of competence in handling datasets of this size and nature. The UK Biobank was accessed through data access approval under the project number #15357 and #20168. Information to apply for data access can be found here: https://www.ukbiobank.ac.uk/enable-your-research/apply-for-access. The CARTaGENE biobank was accessed through data access approval under the project number #406713. Information to apply for data access can be found here: https://www.cartagene.qc.ca/en/researchers/access-request. The GTEx v8 dataset was accessed through dbGaP under project number #19088. The LIMAA dataset was accessed through dbGaP under the project number #26882. Information to apply for data access through dbGAP can be found here: https://dbgap.ncbi.nlm.nih.gov. RNA-sequencing of ADCY9-knocked-down HepG2 cell line data has been deposited under GSE174640: https://www.ncbi.nlm.nih.gov/geo/query/acc.cgi?acc=GSE174640. Source data files and code for all main figures are available here: https://github.com/HussinLab/adcy9_cetp_Gamache_2021.

The following dataset was generated:

| Author(s) | Year | Dataset title | Dataset URL | Database and Identifier |
| --- | --- | --- | --- | --- |
| Gamache I | 2021 | RNA-sequencing of ADCY9-knocked-down HepG2 cell line (embargo) | https://www.ncbi.nlm.nih.gov/geo/query/acc.cgi?acc=GSE174640 | NCBI Gene Expression Omnibus, GSE174640 |

The following previously published datasets were used:

| Author(s) | Year | Dataset title | Dataset URL | Database and Identifier |
| --- | --- | --- | --- | --- |
| The 1000 Genomes Project Consortium | 2015 | 1000 Genomes Project | https://www.internationalgenome.org/data | IGSR: The International Genome Sample Resource, 1000genomes |

*Continued on next page*

*Continued*

| Author(s) | Year | Dataset title | Dataset URL | Database and Identifier |
|---|---|---|---|---|
| Hussin J, Trochet H, Gamache I, Grenier JC, Dubé MP | 2015 | Detection of pleiotropic effects among pharmacogenes in UK Biobank participants | https://www.ukbiobank.ac.uk/enable-your-research/approved-research/detection-of-pleiotropic-effects-among-pharmacogenes-in-uk-biobank-participants | UK Biobank, 15357 |
| Lonsdale J | 2013 | Common Fund (CF) Genotype-Tissue Expression Project (GTEx) | https://www.ncbi.nlm.nih.gov/projects/gap/cgi-bin/study.cgi?study_id=phs000424.v8.p2 | dbGaP, phs000424.v8.p2 |
| Lappalainen T | 2013 | GEUVADIS | https://www.internationalgenome.org/data-portal/data-collection/geuvadis | IGSR: The International Genome Sample Resource, geuvadis |
| Luo Y | 2019 | Early progression to active tuberculosis is a highly heritable trait driven by 3q23 in Peruvians | https://www.ncbi.nlm.nih.gov/projects/gap/cgi-bin/study.cgi?study_id=phs002025.v1.p1 | dbGaP, phs002025.v1.p1 |
| Awadalla P | 2013 | CAG project number #406713 | https://www.cartagene.qc.ca/fr/accueil | CARTaGENE biobank, 406713 |
| Dubé MP, Barhdadi A, Legault MA | 2017 | Pharmacogenomic study using the UK Biobank data | https://www.ukbiobank.ac.uk/enable-your-research/approved-research/pharmacogenomic-study-using-the-uk-biobank-data | UK Biobank, 20168 |

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

## Appendix 1

### Data pre-processing

#### Pre-processing of native american

The genetic data was obtained following correspondence with *Reich et al., 2012* co-authors. The Native American Genetic Dataset (NAGD) dataset being quite sparse and samples coming from many different populations, no missing data threshold nor minor allele frequency or Hardy-Weinberg equilibrium filters were applied prior to the imputation. Harmonization to the hg19 reference genome has been done using GenotypeHarmonizer v.1.4.20 (*Deelen et al., 2014*) and bcftools v.1.9 (*Li et al., 2009*) with the fixref plugin (-m flip option). Imputation was done using the Sanger Imputation Server (*McCarthy et al., 2016*) using Haplotype Reference Consortium (r1.1) reference panel, with a pre-phasing using SHAPEIT2 r.837 (*Delaneau et al., 2013*) and imputation using PBWT (*Durbin, 2014*). Post-imputation quality control was done by keeping sites with an INFO score over 0.8 and keeping genotypes having a posterior probability over 0.9. SHAPEIT2 was run to get phased genotypes (parameters: effective size of 10,000, burn of 10, prune of 10, main of 25, states of 400). The obtained VCF was used in the RFMix analysis (see below). SNPs with missing genotypes higher than 90 % after imputation were removed for LRLD analysis.

#### Pre-processing of the LIMAA cohort

A pre-imputation step was conducted keeping only positions passing minor allele frequency (MAF) of 1%, 1 % of missing data per site and HWE $P$-value > 1e-5 using PLINK v.1.9 (*Purcell et al., 2007*). Harmonization to the hg19 reference genome has then been done using GenotypeHarmonizer and bcftools with the fixref plugin (-m flip option). Imputation was done using the Sanger Imputation Server, using Haplotype Reference Consortium (r1.1) reference panel, with a pre-phasing using SHAPEIT2 and imputation using PBWT. Post-imputation quality control was done by keeping sites with an INFO score over 0.8 and keeping genotypes having a posterior probability over 0.9. Furthermore, positions having less than 5 % missing rate after the genotyping recoding step were kept and duplicated positions were removed. SHAPEIT2 was run to get phased genotypes (parameters: effective size of 10,000, burn of 10, prune of 10, main of 25, states of 400). Another dataset was built to recover one of our SNPs of interest (rs1967309), which was excluded from our previous pipeline because of their INFO score (0.79). In this new dataset, the INFO score threshold was put to 0.7 and the post-imputation position missing data threshold was set to 35%, being less stringent, but recovering our positions. To make sure imputation quality did not impact our results because of incorrectly imputed genotypes, we redid the imputation of LIMAA with the TOPMED reference panel (https://imputation.biodatacatalyst.nhlbi.nih.gov/#!). The imputation $r^2$ score with TOPMED is higher than 0.9 for both, and only very limited differences in imputed genotypes are seen (only 5% and 2% of individual allele mismatches in LIMAA for rs1967309 and rs158477, respectively for the 3,243 individuals).

#### Pre-processing of GTEx genetic data

Starting from the imputed genotyping dataset, we kept bi-allelic SNPs and removed positions with more than 5 % missing genotype, remaining 100,986 SNPs to calculate PCA using flashPCA2. To remove the Hispanic group, we reduced the dimensionality of the top 10 Principal Components (PCs) using the R package UMAP (*McInnes et al., 2020*) (default parameters) to obtain a two dimensional representation of the genetic information contained within those PCs. We identified the largest homogeneous group (self-reported 'white') and excluded outlier groups (*Appendix 1—figure 10a*), used only these individuals for the rest of the analyses. We did our all subsequent analyses with 699 individuals.

#### Pre-processing of CARTaGENE

CARTaGENE biobank (*Awadalla et al., 2013*) includes 40 K individuals from Quebec (Canada) having between 36 and 72 years old. 12,056 individuals were genotyped and among these 911 had RNAseq performed on whole blood (*Hussin et al., 2015*; *Favé et al., 2018*) The genotypes are coming from five different genotyping arrays on which imputation was processed independently. A pre-imputation step was conducted keeping only genotypes passing maf of 1%, 1 % of missing data per site and HWE $P$-value > 1e-5 using PLINK. Harmonization to the Hg19 reference genome has then been done using GenotypeHarmonizer and bcftools with the fixref plugin (-m flip option). Imputation was done using the Sanger Imputation Server, using Haplotype Reference Consortium

(r1.1) reference panel, with a pre-phasing using SHAPEIT2 and imputation using PBWT. Post-imputation quality control was done by keeping sites with an INFO score over 0.8 and keeping individual genotypes having a posterior probability over 0.9.

To extract only white European, we used the same filter as for GTEx, except that we removed SNPs having any missing genotypes which could create bias by different chips, then followed the recommendation from flashPCA2, remaining 8,869 SNPs to calculate PCA. We reduced the dimensionality of the top 10 PCs using the R package UMAP (default parameters) to obtain a two dimensional representation of the genetic information contained within those PCs. We identified the largest homogeneous group (*Appendix 1—figure 10b*), which contains a majority of individuals from European descent (self-reported 'white'), and used only those individuals for the rest of the analysis. We kept 11,362 individuals at the end and among these, 911 individuals for which we had RNAseq. For these individuals, we merged samples from different batches, we removed samples who had less than 10 millions of reads, remaining 790 individuals with expression. After filtering out individuals missing the genotype of either rs1967309 or rs158477 SNPs, we did our interaction analysis on 728 individuals.

## Population genetics iHS analyses

We computed the integrated haplotype score (iHS) (*Voight et al., 2006*) for each subpopulation in the 1000 Genomes project (Methods), a statistics that allows us to detect evidence for recent strong positive selection on derived alleles. The SNP rs1967309 is located in a region of high linkage disequilibrium (LD), delimited by recombination hotspots present in all populations. Several SNPs in this LD block exhibit absolute iHS values higher than two in non-African populations (*Figure 2b*, *Appendix 1—figure 1*), specifically in CEU and GBR (highest signal is a 15 Kb away from rs1967309), CHB, CHS, CDX, KHV, and in all SAS sub-populations, all of which showing signals in several SNPs in less than 200 base pairs from rs1967309. Of note, however, rs1967309 itself does not show value over two in any population. In African populations, no signal is seen in this LD block (*Appendix 1—figure 1*). Other SNPs in *ADCY9* are found to have absolute iHS values higher than 2, especially in the long intron one and around the last exon, but characterizing these signals is beyond the scope of this study.

## Sex-specific differentiation at rs1967309 in *ADCY9*

We first used $F_{ST}$ to evaluate differences in genotype frequencies between males and females. In the PEL from 1000G, we saw suggestive differences between males and females around rs1967309, but did not replicate in the LIMAA cohort, which suggests it was due to small sample size (*Kasimatis et al., 2019*). Another approach we took was to investigate the impact of sex on our PBS results, by splitting the sample between males and females, and recomputing all PBS values using PEL, MXL and CHB for SNPs on chromosome 16 in each subsample. We report result on chromosome 16 that account for chromosome specific population history, as in our analyses of the full cohort, tests on chromosome 16 were more conservative than on the whole genome (i.e. p-values were slightly larger with chromosome 16 alone). Although over the full chromosome, the distribution was not statistically different between males and females ($PBS_{95th-PEL,male} = 0.043$; $PBS_{95th-PEL,female} = 0.040$) as expected, curiously the PEL branch length for all SNPs around rs1967309 increases for males compared to the full-sample results: at rs1967309, the PBS value became 0.096 in males (chromosome 16 empirical p-value = 0.004). On the other hand, for females the value dropped to 0.017 (chromosome 16 empirical p-value = 0.20). No such male-female difference is seen in *CETP*, with the PEL PBS value for rs158477 remaining significantly elevated in both sexes (chromosome 16 empirical p-value$_{rs158477,male}$ = 0.04, chromosome 16 empirical p-value$_{rs158477,female}$ = 0.01, *Appendix 1—figure 3b and c*). This suggests that the LD block around rs1967309 is differentiated between males and females in the Peruvians from 1000G. However, we note that the null model for the $F_{ST}$ statistic underlying PBS assumes no difference in genotype frequencies between sex (i.e. may not be the appropriate tool to address this specific question), and we cannot exclude the possibility of random sampling noise.

## Admixture analyses

Recent admixture and migration events can influence LRLD. If segments of the genome are particularly enriched for a specific ancestry, this could lead to inflated LRLD between these segments. Given that the Peruvian is an admixed population between individuals of Native American ancestry (mainly Andean) as well as of European ancestry (*Appendix 1—figure 2*),

we ran several analyses to establish whether our results at *ADCY9/CETP* can be explained by admixture patterns.

## Local ancestry inference pre-processing

The reference populations used to run RFMix were YRI for the African ancestry, CEU for the European, CHB for the Asian from 1000 G, subpopulations in NAGD (Northern American, Central American and Andean) for the Native American ancestry. We estimated local ancestry with RFMix on PEL from 1000 G and LIMAA individuals.

For all 1000 G populations (YRI, CEU, CHB, PEL), NAGD (Northern American, Central American and Andean) and LIMAA cohort, from the pre-processed datasets (see above) we kept only biallelic SNPs positions, removed SNPs with a MAF under 1 % for each subpopulation, with more than 1 % of missing individuals, with Hardy-Weinberg equilibrium p-value < $10^{-4}$ with mid-adjustment using PLINK. We kept overlapping positions between all datasets and extracted the minor allele frequencies for each reference group. To avoid overlapping positions on the genetic maps, when SNPs had the exact same genetic position, we selected the SNP with the higher variance in allele frequencies (using var in R) between the reference groups (all subpopulations except PEL and LIMAA), keeping between 6742 and 57,238 SNPs per chromosome for RFMix analysis.

## Assessing proportions of global Andean ancestry

To see if there could be a potential enrichment or depletion of Andean ancestry at *CETP* and *ADCY9* loci compare to the rest of the genome, we looked at the proportion of attribution of Andean at those loci compared to the overall distribution of all chromosomes. From the 584,797 positions used for RFMix on all chromosomes, 4,476 position intervals were given, and we calculated the proportion of Andean attribution for each interval, then calculated the 95 % confidence interval (CI) for all chromosomes which is [0.43–0.75]. The proportion at *ADCY9* and *CETP* loci were 0.58 and 0.66 respectively, which suggests that the correlation between *ADCY9* and *CETP* loci is unlikely to be due to an enrichment or depletion of Andean ancestry at both loci. Results are similar when only considering chromosome 16 to calculate the 95% CI.

## LRLD in the Andean population from NAGD

Another question is to assess if the association was already present in the non-admixed ancestral Andean population. If this is the case, the association cannot be explained by the random distribution of Andean segments across the Peruvian genome. We computed LRLD as described in Peruvians in the Andean population from NAGD and we found that the association between rs1967309 and rs158477 is also significant (*ADCY9/CETP* empirical pvalue = 0.04, *Figure 3—figure supplement 1a, b*, *Appendix 1—table 1*). We note that, in this population, strong association signals with rs158477 are also seen at other SNPs in the *ADCY9* LD block region. This result provides convincing evidence that the results in PEL and LIMAA are not due to random distribution of admixed segments but rather might have been inherited from the Andean population, where it was already present, and is maintained since then by selection.

In the Andean population, the association between rs1967309 and rs158477 is not significant when we stratified by sex (*Appendix 1—table 1*), but we still see significant association signals with rs158477 at other SNPs in *ADCY9* LD block in both sexes (*Figure 4—figure supplement 3*)

## Comparison between Peruvian cohorts

To evaluate the genetic difference between Peruvian from 1000 G and LIMAA, we performed a PCA starting from the phased data files. We kept only biallelic SNPs with a MAF higher than 5 % in each cohort and kept only positions with no missing genotype. We followed the suggestion given by flashPCA2 (*Abraham and Inouye, 2014*), remaining 18,345 SNPs for the PCA. We then did a UMAP on 50 PCs given by flashPCA2 using the UMAP package on R (default parameters) (*Appendix 1—figure 2c,d,e*). As seen in the UMAP analysis, population structure exists in LIMAA, and PEL samples are mainly part of the largest subgroup observed in *Appendix 1—figure 2e*, which was kept for LIMAA analyses to remove any confounders linked to population subdivision (see below). Also, the LIMAA cohort was initially recruited as part of a tuberculosis study (*Luo et al., 2019*), but our PCA and UMAP analysis showed no separation according to disease state.

## Null distributions of LRLD

To evaluate how likely it is to observe, specifically in the admixed Peruvian population, a genotype correlation of $r^2 = 0.08$ between SNPs that are approximately 53 Mb apart on the same chromosome like between rs1967309 and rs158477, we have used two approaches. The first one was specific to the two genes under study, *ADCY9* and *CETP,* and therefore controls for all genomic factors specific to these regions. We selected all SNPs with MAF >0.05 in the two genes, and computed $r^2$ values for all 37,802 pairs (461 SNPs in *ADCY9* and 82 SNPs in *CETP*), yielding a null distribution for the expected genetic correlation between these genes. We then compared our $r^2$ value for rs1967309 and rs158477 to this distribution, with its rank being reported as an empirical p-value. This is referred to in the Results section as '*ADCY9/CETP* empirical p-value'.

 This approach is appropriate to correct for the genomic context specific to our genes of interest, but does not account neither for allele frequencies (most SNPs in the null will be at lower frequencies than our two SNPs) nor for overall admixture levels in the genome of this sample, thus we used a second empirical approach to account for these important confounders. For this genome-wide null distribution of the LRLD matching our SNPs, we generated one set of pairs of SNPs and evaluated LRLD between these random pairs in both LIMAA cohort and PEL from 1000G. Since frequencies in the LIMAA cohort are likely better estimates of allele frequencies in the Peruvian population because of the size of the sample, we started our selection based on SNPs' characteristic in this cohort: we extracted pairs of biallelic SNPs from chromosome 1–18, (the other chromosomes being too small) with a MAF between 15% and 30%, separated by between 50 and 60 Mb and 61 ± 10 cM based on the PEL genetic map from 1000G. If SNPs in a pair shared coordinates on the genetic map (in cM) with another SNP from another pair, we kept only one of these pairs. We ended up with 3,576 non-overlapping SNP pairs for calculating the LRLD null distribution matching our rs1967309-rs158477 pair obtained from the LIMAA cohort. For analysis in PEL from 1000G, we added an extra frequency filtering step to remove pairs for which one or both SNPs had a MAF below 5 % in PEL, leaving 3513 pairs for analysis for PEL of 1000G. To calculate an empirical p-value in PEL, we evaluated the number of pairs which had a LRLD value larger to the observed value for rs1967309-rs158477 and divided this number by the total number pairs (n = 3513). This is referred to in the Results section as 'genome-wide empirical p-value'.

 From the 3,513 pairs of SNPs sampled to create the genome-wide null distribution in both sexes in PEL, we stratified by sex and recomputed null distributions for males and females in the same way as for the full cohorts, also with a MAF filter at 5%, leaving 3,505 pairs in males and 3,512 in females in PEL. In males, the $r^2$ value between rs1967309 and rs158477 was the highest of the distribution (genome-wide empirical p-value < 2.85 x 10$^{-4}$), but for females, it was in the 20th percentile (genome-wide empirical p-value = 0.80).

## Permutation analysis of sex-specific LRLD at the positions rs1967309 and rs158477

A second null distribution was derived for evaluating if the LRLD difference between sex for the rs1967309-rs158477 pair was significant, given the significant LRLD observed at these loci. We permuted the sex labels within the cohort and split them into two random groups of 42 pseudo-males and pseudo-females, while making sure an equal number of real males and females (21 of each) are found in each random group, yielding a total of 919 unique random splits that respected these conditions for the 85 PEL individuals. For each iteration, we calculated LRLD between rs1967309-rs158477 for each group and computed the absolute difference between them. To calculate a p-value, we evaluated the number of iterations that had a LRLD difference of more than or equal to the observed difference for the rs1967309-rs158477 pair between true males and females. The true absolute difference in $r^2$ values between rs1967309 and rs158477 (0.346) is the third highest value in this null distribution (p-value = 0.002) (*Appendix 1—figure 4c*).

## Genotype association between rs1967309 and rs158477 in LIMAA

In the LIMAA cohort, we performed a genotype association test using a $\chi^2$ test with four degrees of freedom ($\chi^2_4$) with a permutation scheme to obtain the p-values, as reported in *Rohlfs et al., 2010*, to control for the marginal one-locus genotype counts. To avoid the potential effects of population subdivision on LRLD (*Nei and Li, 1973*; *Slatkin, 2008*), we only kept individuals in the largest, likely more homogeneous, group seen in the UMAP performed on the first 50 PCs with PEL from 1000G (*Appendix 1—figure 2e*). Two smaller distinct groups were identified in the

UMAP analysis and these individuals were excluded from our analysis (cross shaped individuals in *Appendix 1—figure 2e*), leaving 3243 individuals for analysis. The permutation scheme consists in permuting the rs1967309 values 1,000 times and computing the number $\chi_4^2$ values obtained by permutation that are higher than the observed value for the rs1967309/rs158477 pair. For LIMAA, the $\chi_4^2$ value is 82.0 (permutation p-value < 0.001). We then performed the same analysis by stratifying by sex, and obtained a $\chi_4^2$ value of 56.6 (permutation p-value = 0.001) in males and a $\chi_4^2$ value of 37.0 (permutation p-value = 0.017) in females. We note that performing this analysis in the full cohort of 3509 individuals (without excluding individuals from subpopulations shown in *Appendix 1—figure 2e*) yield very similar results (full cohort $\chi_4^2$ = 77.6, male $\chi_4^2$ = 56.5, female $\chi_4^2$ = 34.5). To assess which combination is driving the effect, we used an empirical combination-specific test: the p-value is obtained by breaking the real rs1967309-rs158477 genotype combinations by permuting rs1967309 genotypes and evaluating how many permutated samples show an enrichment of a specific genotype combination as large as in the real data. Interestingly, the combination driving the highly significant male effect is an excess of rs1967309-AA+ rs158477 GG (combination-specific permutation p-value < 0.001), whereas in female, the result seems to be driven by rs1967309-AA+ rs158477 AA (combination-specific permutation p-value = 0.014). These sex-specific genotypic effects could not be captured by a linear model and can explain why the $r^2$ value in LIMAA is smaller than in PEL. Additionally, we note that in both sexes (but mainly in males), the low-frequency rs1967309-GG+ rs158477 GA combination is enriched in LIMAA (observed counts is 112 whereas expected according to allele frequencies at both loci is 59.7).

Finally, to evaluate the effect genome-wide, we calculated the $\chi_4^2$ for all 3576 pairs from the above described genome-wide null distribution for LRLD, then compared these with the value obtained for the rs1967309/rs158477 pair, in all individuals, males and females of LIMAA. In all groups, the rs1967309/rs158477 $\chi_4^2$ values were in the top values (genome-wide empirical p-value$_{all}$ = 0.0003; genome-wide empirical p-value$_{males}$ = 0.002; genome-wide empirical p-value$_{females}$ = 0.001), meaning that the association is significant genome-wide, as found in PEL using $r^2$ (*Figure 3d*).

Despite lower power in 1000G PEL sample, we replicated the rs1967309-AA+ rs158477 GG enrichment in males in PEL using a 2 × 2 $\chi^2$ test, comparing specifically the rs1967309-AA+ rs158477 GG to the three other combinations (rs1967309-nonAA+ rs158477 GG; rs1967309-AA+ rs158477-nonGG; rs1967309-nonAA+ rs158477-nonGG, permutation p-value = 0.018). The rs1967309-AA+ rs158477 AA association seen in females does not replicate (permutation p-value = 0.51) possibly due to low sample size (observed counts is 1, expected counts is 0.66).

Age was available in the LIMAA cohort, enabling us to test whether the LRLD pattern is associated with age, which could suggest a survival benefit if the association is not seen at younger ages. No correlation was seen between genotype and age for rs1967309 and rs158477, and age distributions between males rs1967309-AA+ rs158477 GG and females rs1967309-AA+ rs158477 GG were not significantly different. Because sample size was large enough in this cohort to perform a stratified analysis, we further split the cohort into nearly balanced age categories in males (0–19 years old: 435; 20–25: 464, 26–35: 523; over 35: 519) to establish if the LRLD is present in a specific sub-group. To test if the enrichment of rs1967309-AA+ rs158477 GG in males varies between age group, we calculated the expected frequencies using the frequencies in all age combined in males only (using the whole sample allele frequencies did not change the results). First, we generated a 2 × 2 contingency table comparing rs1967309-AA+ rs158477 GG versus the three others (see above), then calculated a $\chi^2$, then we used a permutation test permuting rs1967309 genotypes 1,000 times to assess statistical significance. The empirical p-values suggest differences between age groups (permutation p-value$_{0-19}$ = 0.12; permutation p-value$_{20-25}$ = 0.21; permutation p-value$_{26-35}$ = 0.01; permutation p-value$_{>35}$ = 0.04), with significant p-values in older groups only. We thus more formally tested if the association differs between age groups between 0 and 25 years old (n = 899) and above 25 (n = 1042): we performed a $\chi^2$ test based on a 2 × 2 contingency table using the chisq.test() function in R, comparing the rs1967309-AA+ rs158477 GG versus all other genotypes for the two age groups, permutating age values 1,000 times to estimate an empirical p-value. There was no significant difference between age group ($\chi^2$ = 0.02, permutation p-value = 0.88). Similar results were obtained when the number of individuals were balanced across the two groups (cut off of 26 years old) or when the four initial age groups were used.

Finally, we also considered how the imputation quality in LIMAA could affect the main result of LRLD, because of imputation from non-representative reference panel populations is known to be problematic. We recomputed the genotype correlation ($r^2$) in LIMAA with our two SNPs imputed with the TOPMED panel, a more representative panel than the Haplotype Reference Consortium initially used. The value obtained is 0.0047 compared to 0.0046 before, showing that imputation quality is unlikely to have affected our results.

## Expression data

### *ADCY9* and *CETP* expression quantification from RNAseq data

By analysing more in depth the *ADCY9* gene and its isoforms, we noticed that a considerable proportion of reads were assigned to a specific isoform, *ADCY9-205* (ENST00000574721.1), a 2.4 Kb long retained intron that does not have a validated status. It was removed from the gene definition file (GTF) to remove any noise from spurious transcription. All GTEx data was therefore reprocessed for *ADCY9* and *CETP* by the same pipeline (see below) to obtain transcription levels per sample at the gene level, consistent for the two genes across cohorts.

For each eQTL dataset (GEUVADIS, GTEx, CARTaGENE), we recalculated the top 5 PCs using genotype data with flashPCA2. For duplicated samples, we kept the sample with the highest read count in the library and removed samples which had a total of less than 10 million of reads. We trimmed the sequencing reads from Illumina adaptors and bad quality ends (BQ >20) using TrimGalore!. We mapped the alignment files on Hg38 human genome reference using STAR v2.6.1a (*Dobin et al., 2013*) with the Ensembl 87 genome annotation, then estimated count for each gene using RSEM v1.3.1 (*Li and Dewey, 2011*). For GTEx, we separated each tissue at this step, then removed tissues with less than 50 samples, leaving samples from 49 different tissues to avoid over-interpretation due to low sample size while maximizing the number of tissues to be tested. We kept the genes which had more than six reads in at least 20 % of the sample. We then normalized expression data using limma (TMM normalization) (*Ritchie et al., 2015*) and voom (*Law et al., 2014*). We calculated PEER factors (*Stegle et al., 2012*) on the normalized expressions. For all sex-stratified analysis, we kept sex-stratified tissues that had at least 50 samples, and recomputed PEER factors with samples from only one sex (which we term sPEER factors).

To test if *ADCY9* and *CETP* expression is correlated across tissues in humans, we used data GTEx and performed a linear regression correcting for the first 5 PCs, age, sex, the collection site (SMCENTER), the sequencing platform (SMGEBTCHT) and total ischemic time (TRISCHD). We find that *ADCY9* and *CETP* gene expression levels are negatively correlated (and significantly($p < 0.05$) so for Adipose-Subcutaneous, Adrenal Gland, Artery Coronary, Artery Tibial, Brain-Cerebellar Hemisphere/Cerebellum/Cortex/Frontal Cortex/Putamen (basal ganglia), Breast-Mammary Tissue, Esophagus-Gastro esophageal Junction/Muscularis, Heart-Left Ventricle, Lung, Prostate, Small Intestine-Terminal, Spleen, Stomach, Uterus, Whole blood), except in skin tissues and cells cultured fibroblast, for which a significant positive correlation is found (*Appendix 1—figure 6*).

### Expression quantitative trait loci (eQTL) analysis for rs1967309 and rs158477

We first looked at the effects of the SNPs independently on their respective genes. The covariates include the first 5 PCs, age (except for GEUVADIS, information not available), sex, as well as PEER factors, calculated to take into account hidden factors. In GTEx, we added additional covariates: the collection site (SMCENTER), the sequencing platform (SMGEBTCHT) and total ischemic time (TRISCHD). One limitation is there is no standardized way of deciding how many PEER factors to include. We tested the robustness of results to the inclusion of different numbers of PEER factors in the models and we report them all for GEUVADIS, CARTaGENE (CaG) and GTEx for transparency (*Appendix 1—figures 7–9*). The maximum number of PEER factors considered follows recommendation by GTEx based on sample size for each tissue.

SNP rs1967309 is a *cis* eQTL of *ADCY9* in whole blood in CARTaGENE (p-value = $4.46 \times 10^{-13}$, ß = –0.10, N = 728, 10 PEER factors) with AA individuals having increased *ADCY9* expression compared to GG individuals. This effect is replicated in whole blood samples from GTEx (N = 559), and several other tissues in GTEx (with esophagus being the most significant), but some tissues show an inversion of the direction of the effect, such as lung and thyroid. These results may differ from GTEx reported eQTL results, because of the removal of ADCY9-205 isoform (see above), the different expression normalization method and filters by ethnicity applied here. SNP rs158477 is

found as a *cis* eQTL of *CETP* in GEUVADIS (p-value = $1.69 \times 10^{-4}$, ß = 0.26) lymphoblastoid cell lines, replicated in GTEx (EBV transformed lymphocytes) as well as in many other GTEx tissues. Tissues with p-value below 0.05 across most PEER factors values (if not all) are: adipose tissues, hippocampus, liver, lung, small intestine, muscle, stomach, thyroid, with GG genotype having consistently less *CETP* expression than AA.

We next tested whether the SNPs are *trans* eQTL for the genes. We found nominally significant associations between rs1967309 and *CETP* expression in the ovary (P-value = 0.0017, N = 138, max PEER factors = 15) and hippocampus (p-value = 0.049, N = 150, max PEER factors = 30), results that are stable across PEER factors values, two tissues for which rs1967309 is not significantly associated with *ADCY9* expression (P-value > 0.05). We found nominally significant associations between rs158477 and *ADCY9* expression in the brain-cerebellar hemisphere and liver.

We next evaluated the interaction effect between rs1967309 and rs158477 on gene expression levels, despite somewhat low statistical power, especially given the small number of samples by tissue. Because the appropriate value of the number of PEER factors to be included on the model is not obvious, we report all values of PEER until the maximum suggested by GTEx for the GTEx tissues (15 for N < 150, 30 for 150≤ N < 250, 45 for 250≤ N < 350, and 60 for N ≥ 350). We also required that the interaction term rs1967309*rs158477 for a tissue had p-values under 0.1 for a majority of values of the number of PEER factors included, to qualify a result to be suggestive of an interaction effect. In the GEUVADIS dataset, which has 287 samples, the interaction was significant on *CETP* expression and stable across PEER factors (*Appendix 1—figure 7a*), which could mean that the effect of a SNP could be modulated by the other SNP. To evaluate this effect further and make sure this is not due to outlier effects or other statistical flukes, we stratified by genotypes of each SNP and investigated the effect of the other SNP on *CETP* expression. We used a linear regression with the same covariates mentioned above. We first stratified by the genotype of rs1967309, then evaluated the effect of rs158477 on *CETP* expression. SNP rs158477 is significant in the AA of rs1967309 (p-value = 0.03, ß = 0.45, OR=[1.05–2.36], n = 46) and AG (p-value = 0.009, ß = 0.24, OR=[1.07–1.53], n = 143), but not for GG (p-value = 0.58, ß = 0.07, OR=[0.83–1.40], n = 96) (*Figure 5b*), potentially showing a mitigation of the eQTL effect of rs158477 for each alternative allele of rs1967309 on *CETP* expression. We also evaluated the effect of rs1967309 when we stratified by rs158477 on *CETP* expression. SNP rs1967309 is significant only for GG of rs158477 (p-value = 0.05, ß = 0.3, OR=[1.00–1.83], n = 72), and not for GA (p-value = 0.51, ß = −0.06, OR=[0.77–1.14], n = 139) nor AA (p-value = 0.68, ß = −0.07, OR=[0.66–1.30], n = 74). The second dataset that we used was the GTEx, in which we evaluated 49 tissues. Since the effects across tissues are likely not independent, we did not correct for multiple testing, keeping a suggestive threshold at 0.10 and a significant threshold at 0.05, but those values need to be reached for a majority values of the number of PEER factors included to be convincing. Among the 49 tissues (*Appendix 1—figure 8*), those with p-values under 0.10 for several numbers of PEER factors are hippocampus (N = 150), hypothalamus (N = 156), brain spinal cord (cervical c-1) (N = 114), substantia nigra (N = 100) and skin sun-exposed (N = 507). Among those tissues, rs1967309 is only a cis-eQTL of *ADCY9* in the substantia nigra.

Since the selective pressure differ between sexes, we stratified our expression analysis by sex. For *CETP* expression analysis, there are no consistent signals for GEUVADIS, possibly reflecting lack of power or that there is no sex-specific effect in lymphoblastoid cell lines (*Appendix 1—figure 7c*). However, in CaG, the significant interaction found is present only in male (again for the genotypic coding, *Appendix 1—figure 7d*). In GTEx, we see significant interaction effects in males in tissues that had signals with sex-combined, such as brain hippocampus ($N_{male}$ = 105), hypothalamus ($N_{male}$ = 112) and spinal cord cervical ($N_{male}$ = 72), skin sun-exposed for CETP ($N_{male}$ = 330) (*Figure 5d*, *Appendix 1—figure 9*). We note that for most brain tissues, the low sample size in females does not allow to conclude on the presence of an interaction effect in that sex. In these tissues, the direction of the effect in males is reversed compared to what is observed in GEUVADIS with sexes combined (*Figure 5b*), whereas the highly significant result in skin shows an effect consistent with the sex-combined GEUVADIS result (p-value = 0.0017, ß = −0.32). More specifically, in the sun-exposed skin samples, in rs1967309 AA males, copies of the rs158477 A allele increase CETP expression by 0.49 (95% CI 0.12–0.87) on average. In rs1967309 AG males,

the effect of rs158477 is null (p-value$_{AG}$ = 0.33) and the effect of the rs158477 A allele is suggestive in rs1967309 GG individual (p-value$_{GG}$ = 0.10) with a decrease of the CETP expression. Conversely, if we look at the effect of rs1967309 on *CETP* expression in skin sun-exposed when we stratified by rs158477 in males, SNP rs1967309 is neither a significant eQTL of *CETP* in GG of rs158477 in males (p = 0.11, n = 89) nor for GA (p = 0.65, n = 164), but is a significant eQTL for AA males (p = 0.026, ß = −0.46, OR=[-0.87,–0.06], n = 76).

We also identified new tissues where the interaction is either suggestive or significant in females, in artery tibial (N$_{female}$ = 156), heart atrial appendage (N$_{female}$ = 97), spleen (N$_{female}$ = 69), and stomach (N$_{female}$ = 105) (*Appendix 1—figure 9*), with an effect reversed compared to the initial GEUVADIS result (*Appendix 1—figure 7a*). For the pituitary tissue, it is significant in both sex (additive coding in female and genotypic coding in male, N$_{male}$ = 156, N$_{female}$ = 63), but the direction of the additive coding is reversed between sexes, possibly explaining why the sex-combined analysis did not show any signal. We note that the newly discovered signals are mainly for females, indicating that the signal was hidden by the male effects (or absence of effects), likely because of higher sample sizes.

## Experiments

### Real-time PCR quantification

Reverse transcription was performed from 500 ng total RNA in a 20 ml reaction using High-Capacity cDNA Reverse Transcription Kit (Applied Biosystems cat #4368814). RNA quantification was assessed using Agilent RNA 6000 Nano Kit for Bioanalyzer 2,100 System (Agilent Technologies). Primers were designed using the Beacon designer software v.8 (Premier Biosoft) (*Appendix 1—table 3*). The real-time PCR was carried out with SYBR-Green reaction mix (BioRad cat #1725274). The thermal cycling program was 3 min at 95 °C for initial denaturation followed by 40 cycles of denaturation for 10 s at 95 °C, 30 sec annealing at 60 °C and 30 sec extension at 72 °C. qPCR assay was normalized with PGK1 and HBS1L genes.

### Western blot analysis

A total of 200 ml of cell media from HepG2 transfected cells were concentrated using Amicon Ultra 0.5 ml 10 kDa cutoff units (cat #UFC501096) to 25 ml. Proteins were separated on 10 % TGX-acrylamide gel. After O/N electrotransfer at 10 volts to PVDF membranes, CETP protein was determined using a primary anti-CETP rabbit monoclonal antibody (Abcam cat #ab157183) 1:1000 in 3 % BSA, TBS, tween 20 0.5%, O/N 4 °C, followed by HRP-conjugated secondary antibody goat anti-rabbit 1:10,000 in 3 % BSA 1 h at room temperature. Detection was performed using Western Lightning ECL Pro (Perkin Elmer cat #NEL122001EA). Proteins levels were normalized with total proteins loaded.

## Phenotype associations

### Two-way and three-way interaction models in UK biobank

With rs1967309 coded under the genotypic model, allowing to capture non-additive effects, we tested if the effect of the interaction term was significant for a phenotype *Y* using a likelihood ratio test (LRT) by comparing the following models:

$$Y rs158477 + rs1967309 + sex + age + PC\left(1-5\right)\left(m1\right)$$

$$Y rs158477 * rs1967309 + sex + age + PC\left(1-5\right)\left(m2\right)$$

$$Y rs158477 * rs1967309 * sex + age + PC\left(1-5\right)\left(m3\right)$$

We used the R function glm with family = "binomial" and compared models using the following: anova(a, b, test = "LRT"), with a = *m1* and b = *m2* to test for two-way interaction effects, and a = *m2* and b = *m3* to test for three-way interaction effects.

Individually, SNP rs1967309 (f$_A$ = 39%) is nominally associated with heart rate, and rs158477 (f$_G$ = 47%) with the systolic blood pressure (*Figure 6—figure supplement 1*), both results being

mainly driven by association in females. Both SNPs are nominally associated with waist-hip ratio, rs1967309 in females, rs158477 in males. None of these effects are genome-wide significant.

## Phenotype associations in GTEx

In GTEx, we had the variable MHHRTATT (phv00169162.v8.p2) for cardiovascular disease. This variable is defined as *Heart attack, acute myocardial infarction, acute coronary syndrome*. GTEx also has phenotypes, including cardiovascular traits. The variable DTHFUCOD (First Underlying Cause Of Death) was used to identify individuals whose cause of death included *Heart Attack/ Stroke, Heart Disease, Acute Myocardial Infarction, Possible MI*, who were considered as cases. From the 699 samples kept, we excluded six for which the phenotype was missing or unknown for MHHRTATT variable and for which the cause of death was unrelated to heart disease, yielding a total of 130 cases and 563 controls. We added as covariates: sex, age and top 5 PCs. For this phenotype, neither rs1967309 nor rs158477 are associated with those phenotypes when taken alone. However, the interaction and both SNPs in the equation are significant (p-value ≤ 0.05) or close to be significant (p-value ≤ 0.10) for the phenotype (p-value$_{MHHRATT}$ = 0.01, Estimate$_{MHHRATT}$ = −0.54) (*Figure 6—figure supplement 1*). Like for *CETP* expression, this means that for each G allele for rs1967309, there is a decrease of the effect of rs158477 on cardiovascular outcome. A difference with *CETP*'s expression is that there is an inversion of the direction of effect. In other word, for AA of rs1967309, directions of the effect of rs158477 are positive, with GG having less probability to have an event than AA (Estimate$_{MHHRATT}$ = 0.47), but for GG of rs1967309, estimates of rs158477 are negative (Estimate$_{MHHRATT}$ = −0.79). Those results are consistent with the direction of dalcetrapib pharmacogenomic analysis. Considering that the GG genotype of rs158477, with less *CETP*'s expression, is a proxy for dalcetrapib, which is an inhibition of CETP, the same gradient is present for rs1967309. In AA of rs1967309, there is less heart disease with dalcetrapib (*Tardif et al., 2015*). In GG of rs1967309, there is more heart disease with dalcetrapib. More study of this interaction is needed to understand the mechanism. However, this could lead to new insights into the potential biological mechanism behind the pharmacogenomic association involving the gene *ADCY9* with cardiovascular outcome of dalcetrapib.

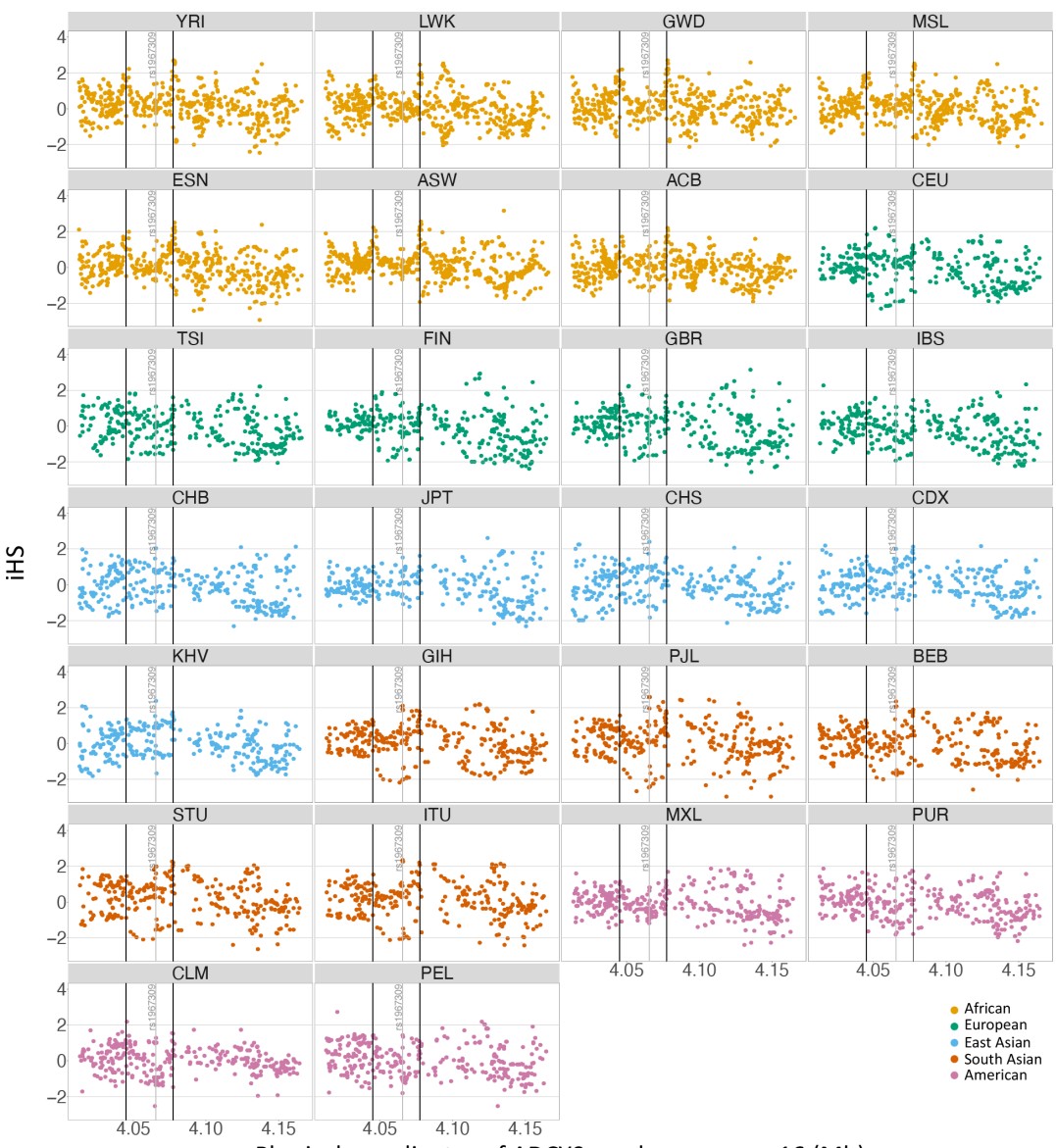

**Appendix 1—figure 1.** Selection signature in *ADCY9*. iHS values and recombination for all populations in the *ADCY9* gene. Vertical black lines represent the highest recombination rates around rs1967309 from 1000 G population-specific genetic maps. Horizontal line represents the value at 2 and –2. Different colors represent one super population. In order of color: African, European, East Asia, South Asia and America. Abbreviations for the subpopulation of 1000 G can be found here https://www.internationalgenome.org/category/population/.

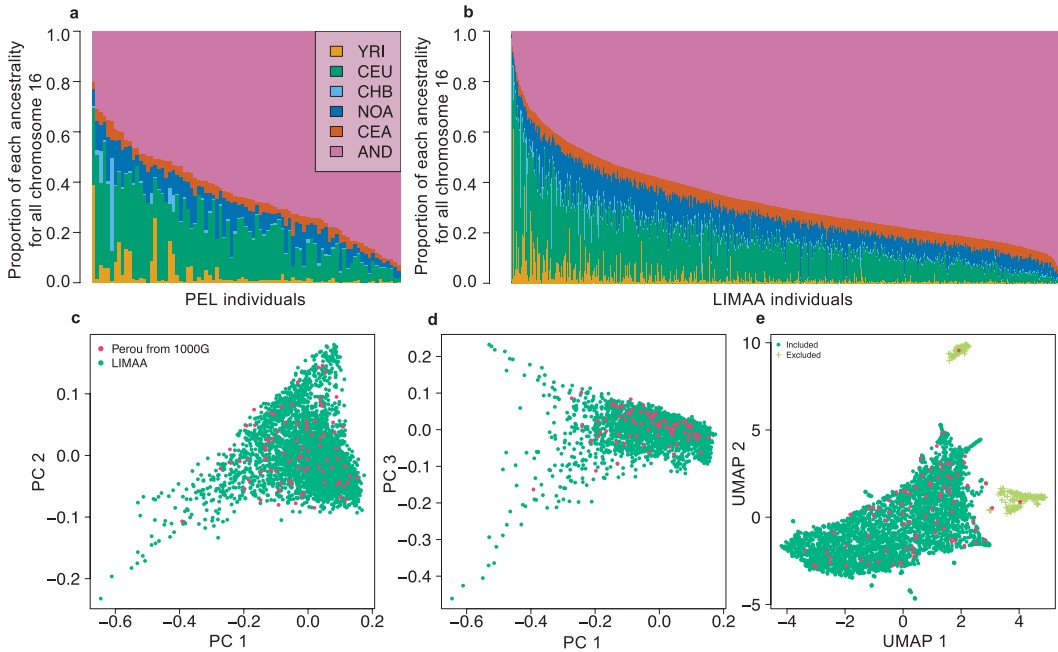

**Appendix 1—figure 2.** Population structure of Peruvian from LIMAA and Peruvian from 1000G. Ancestry distribution on all chromosomes in the Peruvian from 1000G (**a**) and LIMAA cohort (**b**). Overall weighted proportion given by RFMix using reference populations from 1000G and Native American Genetic Dataset (NAGD) for the Peruvian population from 1000G (**a**) and from LIMAA cohort (**b**) . 1000G populations YRI, CEU, and CHB were chosen to represent African, European, and Asian ancestry, respectively. (**c,d**) Principal Component Analysis using flashPCA on Peruvian from 1000G and LIMAA cohort. The top three PCs is shown. (**e**) UMAP analysis on the top 50 PCs. To limit confounders due to population structure, we excluded individuals in LIMAA coming from the two small groups identified by the UMAP (cross shaped light green symbols in (**e**)). Abbreviations for 1000G can be found here: https://www.internationalgenome.org/category/population/. Abbreviations for the Native American (NAGD): NOA: northern American; CEA: central American; AND: Andean.

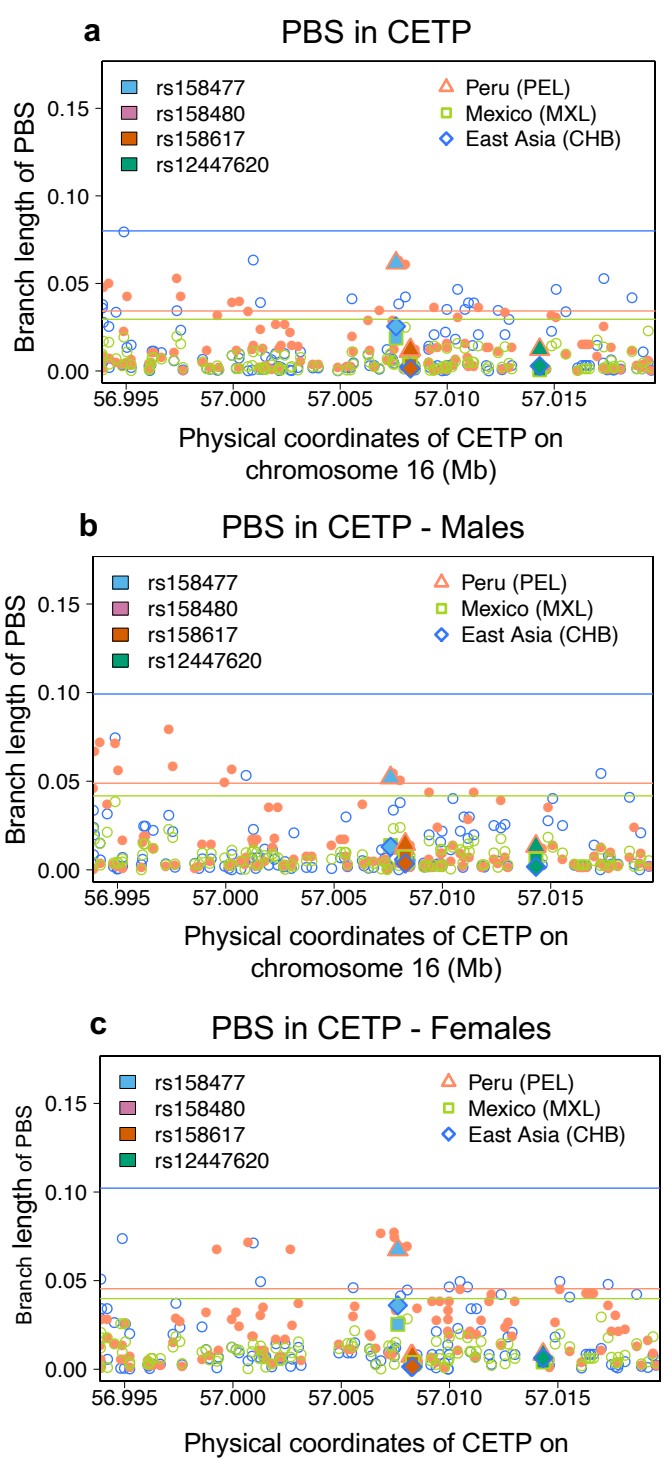

**Appendix 1—figure 3.** Populational differentiation of *CETP* gene using PBS statistic. PBS values in the *CETP* gene, comparing the CHB (outgroup), MXL and PEL identified by different colors, overall (**a**) , in males (**b**) and in females (**c**) . Horizontal lines represent the 95th percentile PBS value genome-wide (**a**) or the chromosome 16 (**b,c**) for each population. Position with $r^2$ higher than the 99th percentile in the Peruvian population from the 1000G are represented by colored shape.

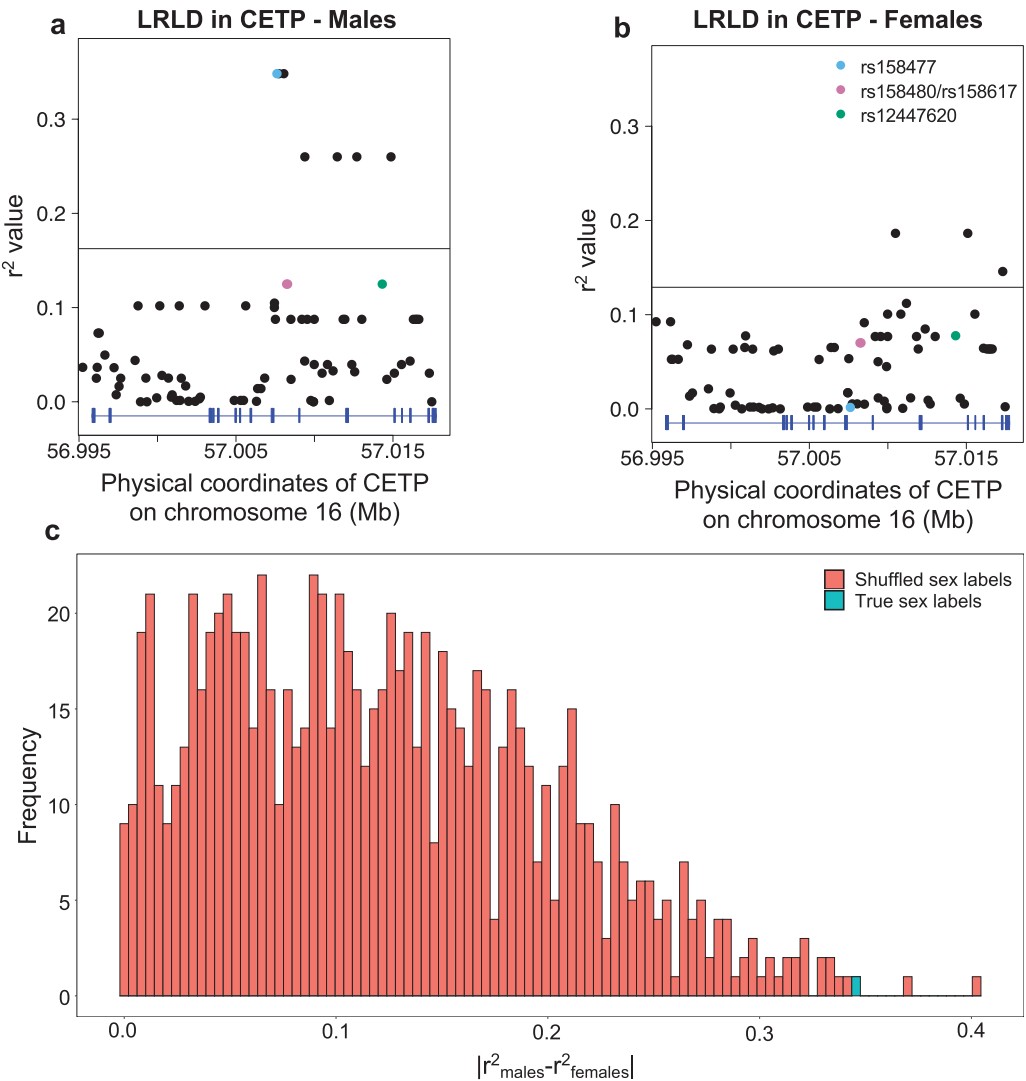

**Appendix 1—figure 4.** Long-range linkage disequilibrium shown in CETP for the PEL population from 1000G, stratified by sex. Genotype correlation ($r^2$) between the three loci identified in *CETP* (see *Figure 2a*) to be higher than the 99[th] percentile and all SNPs with MAF >5% in *ADCY9*, in males (**a**) and females (**b**). The horizontal black line is the 99th of all those comparisons between ADCY9 and CETP by sex. (**c**) Distribution of absolute difference of genotype correlation values obtained during the permutation analysis that shuffled the sex label for rs1967309 and rs158477 (red), compared to the value obtain with the real sex labels (blue).

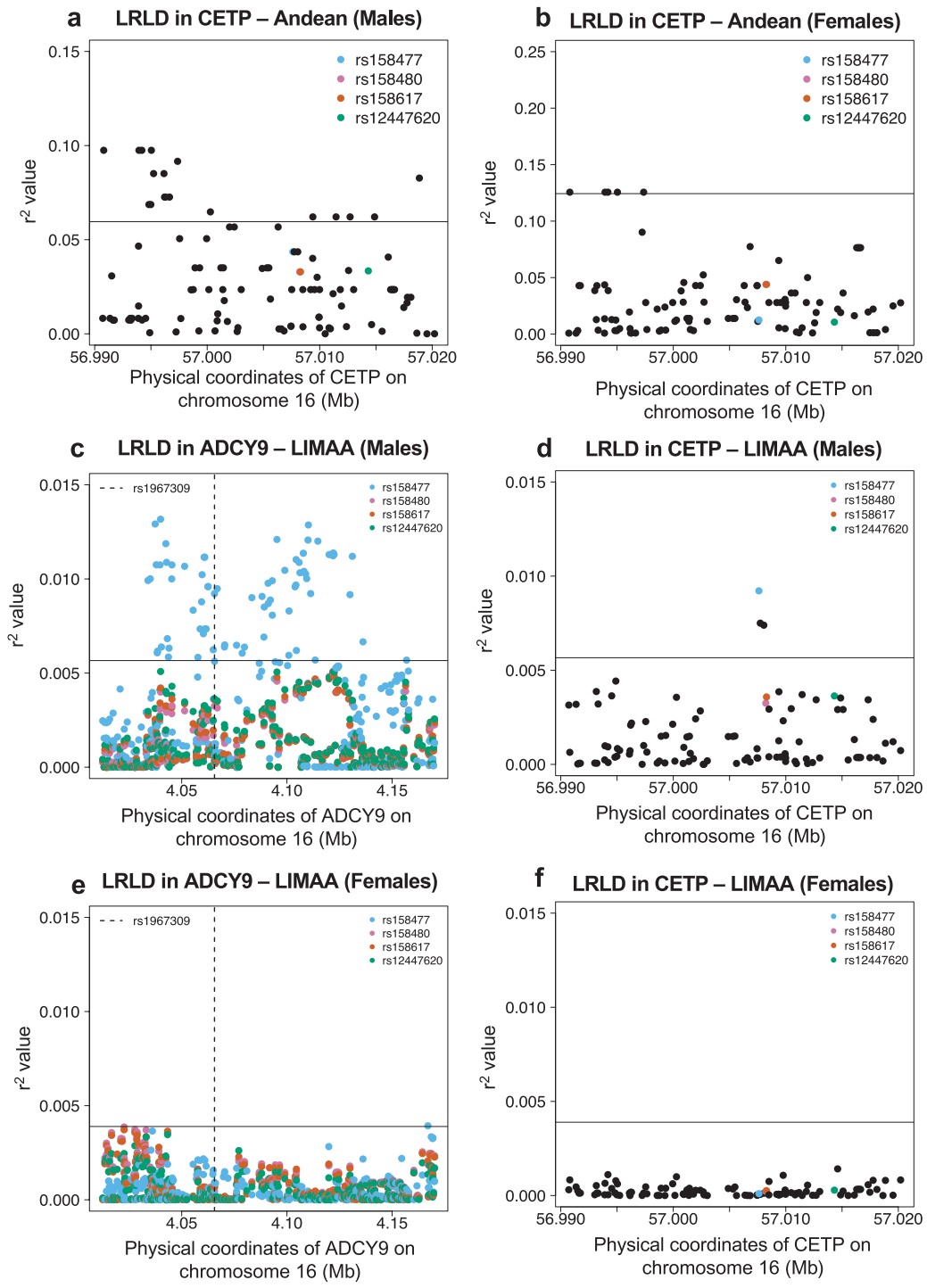

**Appendix 1—figure 5.** Long-range linkage disequilibrium in the Andean population from NAGD (**a,b**) and LIMAA cohort (**c–f**). (**a,b,d,f**) Genotype correlation ($r^2$) between rs1967309 and all SNPs with MAF >5% in *CETP*, for the Andean population from NAGD (**a,b**) and the LIMAA cohort (**d,f**). (**c,e**) Genotype correlation between the three loci identified in *Figure 3a* to be higher than the 99th percentile and all SNPs with MAF >5% in *ADCY9* in LIMAA. Males ($N_{Andean}$ = 54, $N_{LIMAA}$ = 1941) (**a,c,d**) and females ($N_{Andean}$ = 34, $N_{LIMAA}$ = 1302) (**b,e,f**) are shown separately. The horizontal line is the 95th (**a,b**) and 99th (**c–f**) percentile of all comparisons between *ADCY9* and *CETP* genes.

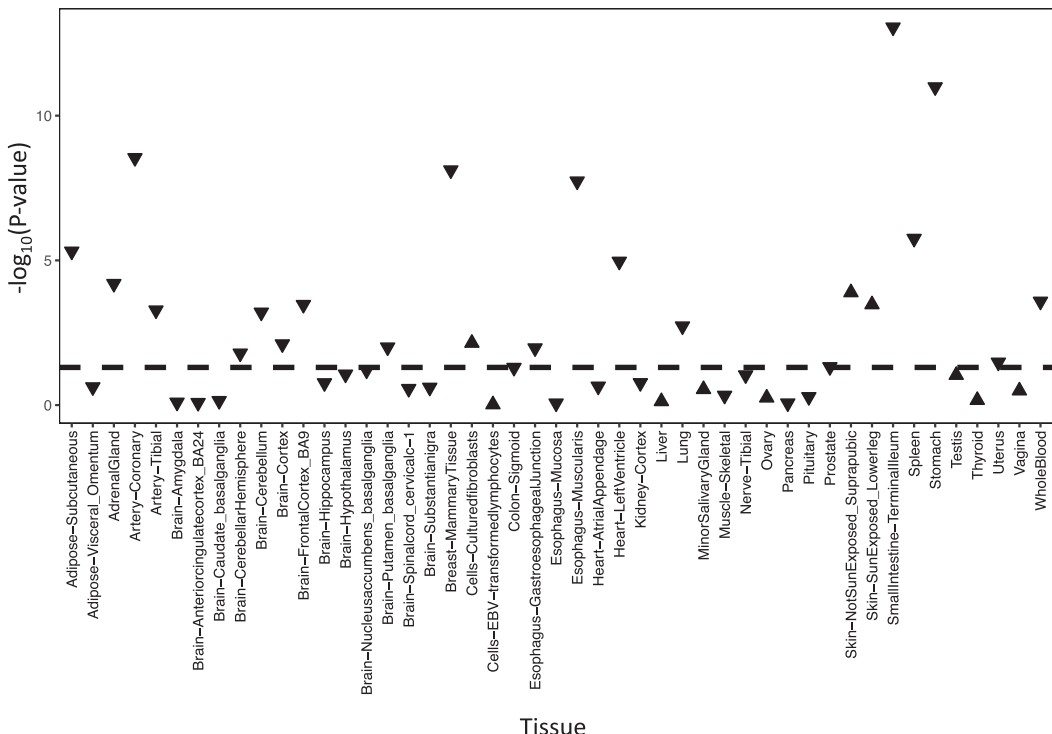

**Appendix 1—figure 6.** Significance of the correlation between *ADCY9* and *CETP* expression across GTEx tissues. P-values are presented on a -log$_{10}$ scale and are obtained from a linear regression on normalized expression with correction for age, sex, top 5 PCs, ischemic time death, sequencing platform, and sequencing center. Regular triangles mean that both gene expression levels are positively correlated, inverted triangles mean that both gene expression levels are inversely correlated. The dashed line represents the p-value at 0.05.

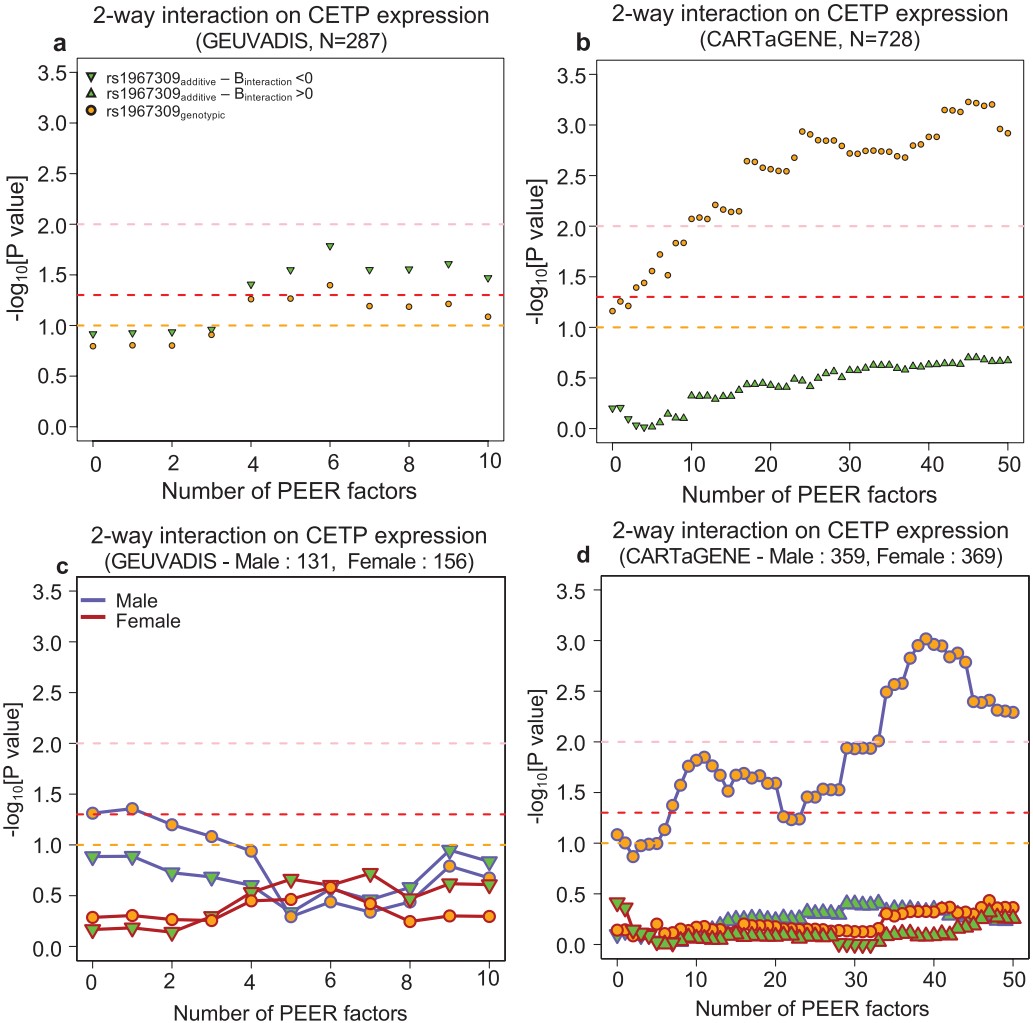

**Appendix 1—figure 7.** Epistatic effects between rs1967309 and rs158477 on CETP expression in GEUVADIS (LCL, N = 287) and CARTaGENE (Whole blood samples, N = 728). P-values are presented on a -$\log_{10}$ scale and are reported in function of the number of PEER/sPEER factors in GEUVADIS (LCL) (**a,c**) and CARTaGENE (**b,d**) in sex-combined (**a,b**) and sex-stratified (**c,d**) analyses. For all models, rs158477 is coded as additive (GG = 0, GA = 1, AA = 2). In the additive model (green triangle), rs1967309 is coded as additive (AA = 0, AG = 1, GG = 2), p-values are obtained using a linear regression in R. In the genotypic model (orange circle), rs1967309 is coded as a genotypic variable and p-values are obtained from a likelihood ratio test comparing models with and without the interaction term between the SNPs. The orange, red, and pink lines represent p-values of 0.1, 0.05, and 0.01 respectively. The sample sizes reported are the number of individuals left after removing participants with missing genotypes for rs1967309 and/or rs158477. In (**c,d**) the color of the lines represents the sex label.

rs1967309*rs158477 on CETP expression

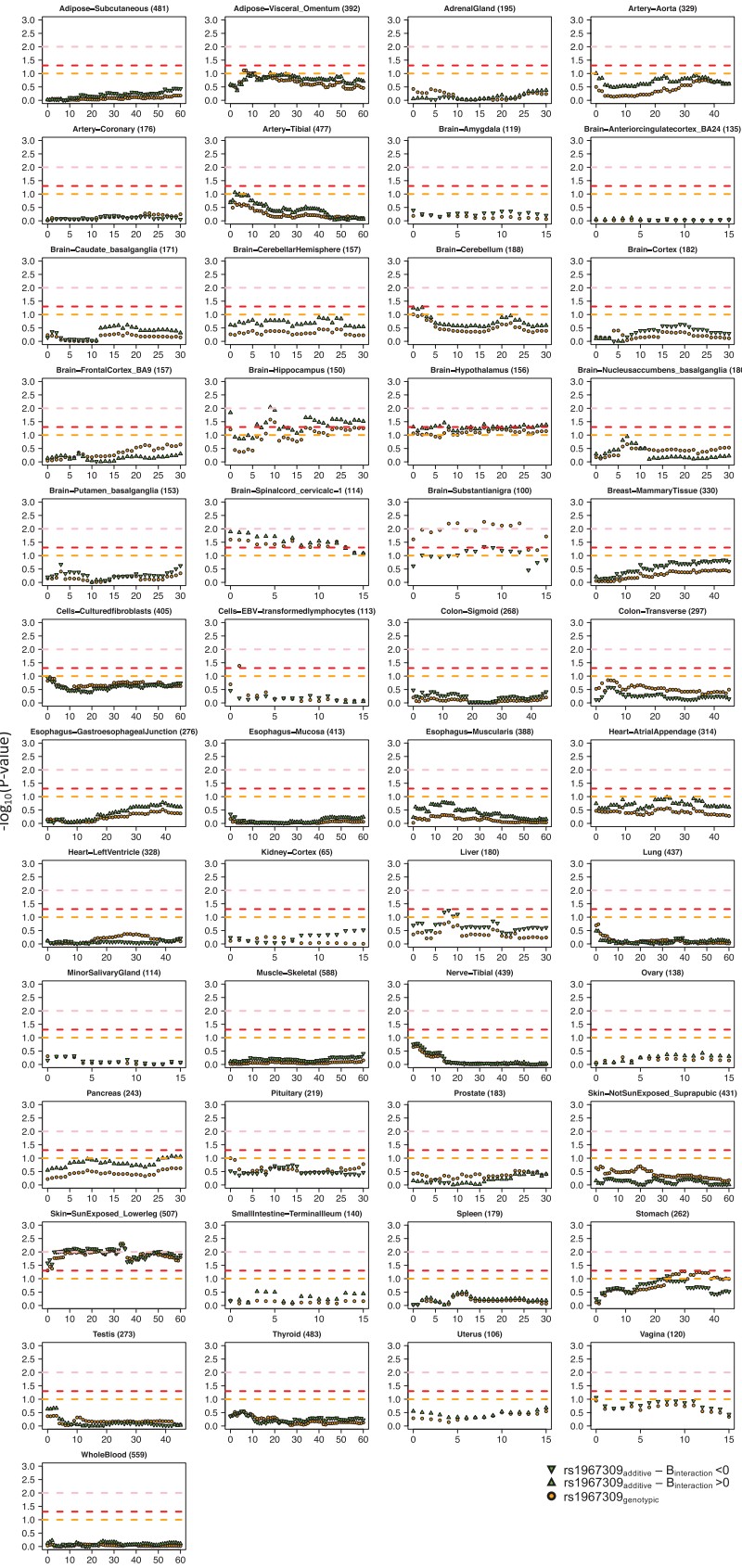

Number of PEER factors

**Appendix 1—figure 8.** Sex-combined epistatic effect p-values for the interaction between rs1967309 and rs158477 on *CETP* expression depending on the number of PEER factors in GTEx by tissue. P-values are presented on a -log$_{10}$ scale. For all models, rs158477 is coded as additive (GG = 0, GA = 1, AA = 2). In the additive model (green triangle), rs1967309 is coded as additive (AA = 0, AG = 1, GG = 2), p-values are obtained using a linear regression in R. In the genotypic model (orange circle), rs1967309 is coded as a genotypic variable and p-values are obtained from a likelihood ratio test comparing models with and without the interaction term between the SNPs. The orange, red and pink lines represent p-values of 0.1, 0.05 and 0.01 respectively. The tissue type and the number of samples for each, used in the analysis, are reported in the titles of the subgraphs.

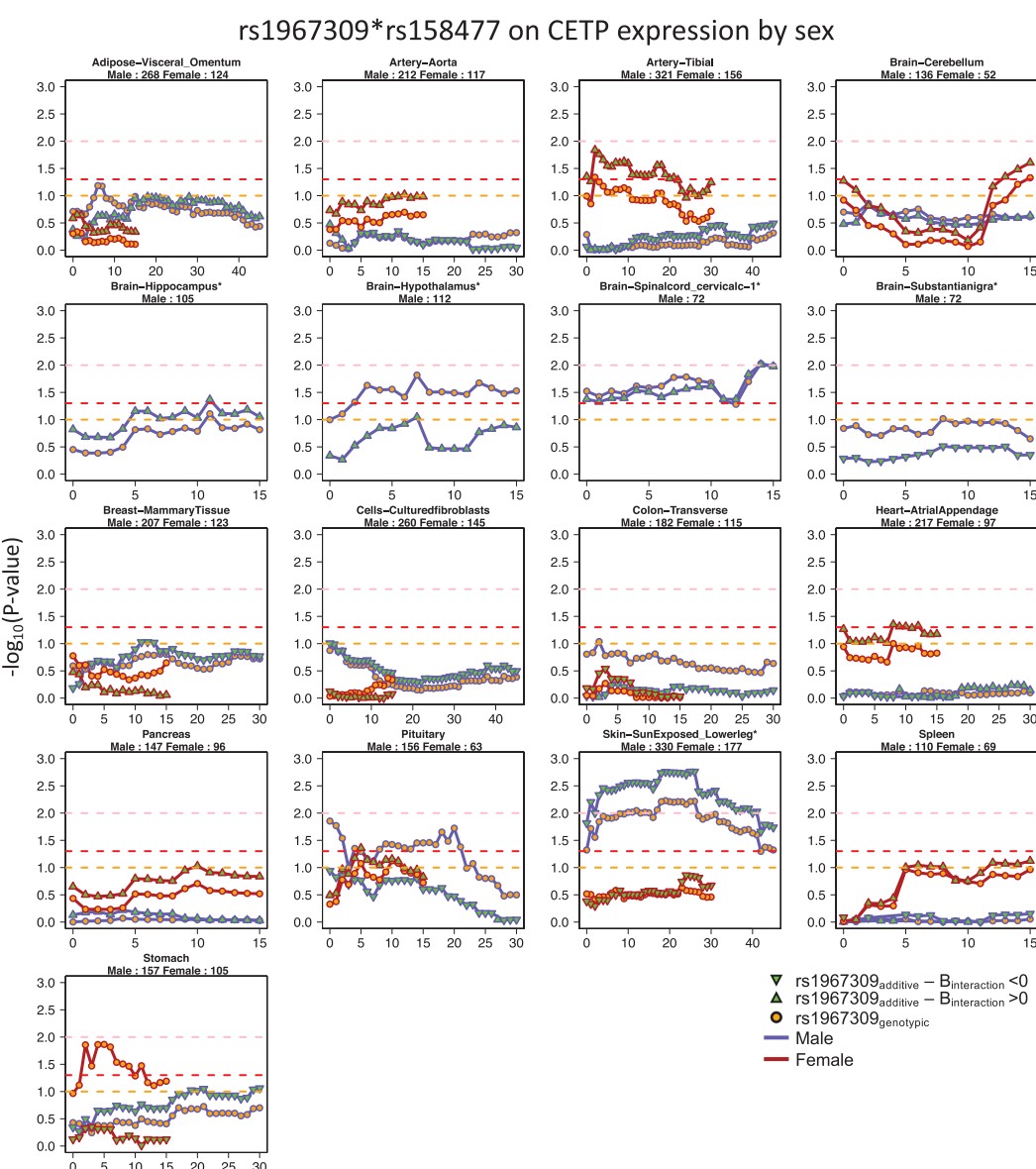

**Appendix 1—figure 9.** Sex-specific epistatic effects between rs1967309 and rs158477 on *CETP* expression depending on the number of sPEER factors in GTEx by tissue. P-values are presented on a -log$_{10}$ scale. For all models, rs158477 is coded as additive (GG = 0, GA = 1, AA = 2). In the additive model (green triangle), rs1967309 is coded as additive (AA = 0, AG = 1, GG = 2), p-values are obtained using a linear regression in R. In the genotypic model (orange circle), rs1967309 is coded as

*Appendix 1—figure 9 continued on next page*

*Appendix 1—figure 9 continued*

a genotypic variable and p-values are obtained from a likelihood ratio test comparing models with and without the interaction term between the SNPs. The orange, red and pink lines represent p-values of 0.1, 0.05 and 0.01 respectively. The tissue type and the number of samples for each, used in the analysis, are reported in the titles of the subgraphs. The color of lines represents the sex label. Only tissues with at least one value under 0.10 are showed. Tissues with an asterisk (*) next to their title are tissues showing a the suggestive/significant effect in the sex-combined analysis.

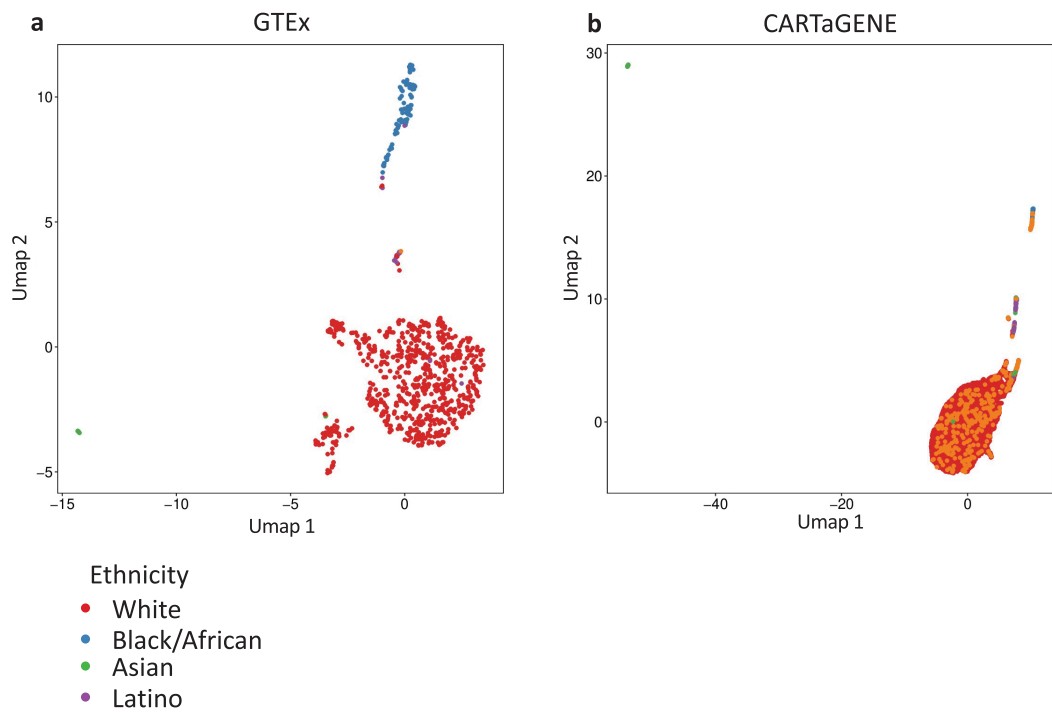

**Ethnicity**
- White
- Black/African
- Asian
- Latino
- Other/Unknown

**Appendix 1—figure 10.** Population structure in datasets analysed. We estimate population structure using UMAP on the top 10 PCs generated with flashPCA2 on (**a**) GTEx (N = 699) and (**b**) CARTaGENE (N = 12,056) biobanks. The self-reported white non-Latino individuals were selected for further analyses.

**Appendix 1—table 1.** Long-range linkage disequilibrium analysis in three datasets, and in subsets of the cohorts.

Number of individuals (N) in each subset is reported. P-values correspond to the *ADCY9/CETP* empirical p-values computed as described in Section *Long-range linkage disequilibrium* in Methods. $r^2$ were obtained from the geno-r2 option of vcftools software. For 1000 G populations, abbreviations can be found here https://www.internationalgenome.org/category/population/.

| Cohort | Population | Sex | Number | r² | p-value *ADCY9-CETP* |
|---|---|---|---|---|---|
| | YRI | All | 108 | 0.0236 | 0.11 |
| | CEU | All | 99 | 0.0003 | 0.86 |
| | GBR | All | 91 | 0.0117 | 0.28 |
| | CHB | All | 103 | 0.004 | 0.53 |
| | MXL | All | 64 | 0.0007 | 0.83 |
| | | All | 85 | 0.0796 | $5.42 \times 10^{-3}$ |
| | | Male | 41 | 0.3483 | $8.23 \times 10^{-5}$ |
| 1000 G | PEL* | Female | 44 | 0.0016 | 0.78 |
| | | All | 3,243 | 0.0046 | $3.24 \times 10^{-3}$ |
| | | Male | 1941 | 0.0097 | $3.71 \times 10^{-3}$ |
| LIMAA | LIMAA | Female | 1,302 | 0.0003 | 0.52 |
| NAGD | | All | 81 | 0.0084 | 0.44 |
| | Northern Amerind (NOA) | Male | 27 | 0.0634 | 0.16 |
| | | Female | 54 | 0.0699 | 0.07 |
| | | All | 81 | 0.0281 | 0.12 |
| | Central Amerind (CEA) | Male | 34 | 0.0316 | 0.28 |
| | | Female | 47 | 0.0257 | 0.24 |
| | Andean (AND) | All | 88 | 0.0293 | 0.04 |
| | | Male | 54 | 0.0436 | 0.09 |
| | | Female | 34 | 0.0125 | 0.55 |

*Discovery cohort.

**Appendix 1—table 2.** Details on metabolic and clinical variables extracted from the UK Biobank.

| Variable ID | UK biobank variable location | Number of samples used for interaction |
|---|---|---|
| *Category 100011 - Blood pressure - Physical measures - UK Biobank Assessment Centre* | | |
| Pulse rate at baseline (Pulse rate) Units: bpm | Data-Field 102 (automatic entry) or Data-Field 95 (manual entry), to be derived as follows: <br> • Pulse rate, automated reading (Data-Field 102) used mean of available measures for instance 0 (baseline) only. If a manual measure is available for an individual (Data Field 95 below) then do not use this automated reading (assumed to be abnormal). <br> • Pulse rate (during blood-pressure measurement) (Data-Field 95), use Instance 0 (baseline). Use mean when there are multiple measures for a same individual. | All = 395,319 <br> Male = 182,279 <br> Female = 213,040 |

*Appendix 1—table 2 Continued on next page*

*Appendix 1—table 2 Continued*

| Variable ID | UK biobank variable location | Number of samples used for interaction |
|---|---|---|
| Diastolic blood pressure at baseline (Diastolic BP)<br>Units: mmHg | Data-Field 4,079 (automatic entry) or Data-Field 94 (manual entry), as follow:<br>• Diastolic blood pressure, automated reading: Data-Field 4079,, use mean of available measures for instance 0 (baseline) only. If a manual measure is available for an individual (Data Field 94) then do not use this automated reading (assumed to be abnormal).<br>• Diastolic blood pressure, manual reading: Data-Field 94, use mean of available measures for instance 0 (baseline) only. | All = 395,384<br>Male = 182,326<br>Female = 213,058 |
| Systolic blood pressure at baseline (Systolic BP)<br>Units: mmHg | Data-Field 4,080 (automatic entry) or Data-Field 93 (manual entry), as follow:<br>1. Systolic blood pressure, automated reading: Data-Field 4080,, use mean of available measures for instance 0 (baseline) only. If a manual measure is available for an individual (Data Field 93) then do not use this automated reading (assumed to be abnormal).<br>2. Systolic blood pressure, manual reading: Data-Field 93, use mean of available measures for instance 0 (baseline) only. | All = 395,353<br>Male = 182,316<br>Female = 213,037 |
| *Category 100010 - Body size measures - Anthropometry - Physical measures - UK Biobank Assessment Centre* | | |
| Waist circumference at baseline (Waist circumference)<br>Units: cm | Data field 48, use mean of available measures for instance 0 (baseline) only. | All = 395,006<br>Male = 182,089<br>Female = 212,917 |
| Hip circumference at baseline (Hip circumference)<br>Units: cm | Data field 49, use mean of available measures for instance 0 (baseline) only. | All = 394,651<br>Male = 181,988<br>Female = 212,663 |
| Waist-hip ratio | Compute waist/hip | All = 394,944<br>Male = 182,056<br>Female = 212,888 |
| Weight<br>Units: Kg | Data-Field 21,002 (automatic entry) or Data-Field 3,160 (manual entry), as follow:<br>(3) Weight: Data-Field 21002,, use mean of available measures for instance 0 (baseline) only.<br>Only if unavailable, then use:<br>(4) Weight, manual reading: Data-Field 3160,, use mean of available measures for instance 0 (baseline) only. | All = 394,377<br>Male = 181,732<br>Female = 212,645 |
| Height<br>Units: cm | Data-Field 50 or 12,144.<br>(5) Standing height: Data Field 50, used mean of available measures for instance 0 (baseline) only.<br>Only if unavailable, then use:<br>(6) Height: Data-Field 12144,, used mean of available measures, as this is a singular instance field | All = 394,871<br>Male = 181,969<br>Female = 212,902 |

*Appendix 1—table 2 Continued*

| Variable ID | UK biobank variable location | Number of samples used for interaction |
|---|---|---|
| UK Biobank BMI (BMI) Units: Kg/m2 | Data field 21001,, used mean of available measures for instance 0 (baseline) only. | All = 394,173 Male = 181,705 Female = 212,468 |

*Category 100009 - Impedance measures - Anthropometry - Physical measures - UK Biobank Assessment Centre*

| Variable ID | UK biobank variable location | Number of samples used for interaction |
|---|---|---|
| Trunk fat percentage (% Trunk fat) Units: % | Data field 23127,, use mean of available measures for instance 0 (baseline) only. | All = 388,569 Male = 178,837 Female = 209,732 |
| Body fat percentage (% Body fat) Units: % | Data field 23099,, use mean of available measures for instance 0 (baseline) only. | All = 388,600 Male = 178,752 Female = 209,848 |
| Basal metabolic rate Units: KJ | Data field 23105,, use mean of available measures for instance 0 (baseline) only. | All = 388,585 Male = 178,758 Female = 209,827 |
| Whole body water mass Unites: Kg | Data field 23102,, use mean of available measures for instance 0 (baseline) only. | All = 388,719 Male = 178,881 Female = 209.838 |

*Category 100020 - Spirometry - Physical measures - UK Biobank Assessment Centre*

| Variable ID | UK biobank variable location | Number of samples used for interaction |
|---|---|---|
| Forced vital capacity (FVC) Units: L | Data field 20151,, use mean if more than one measure. | All = 297,461 Male = 138,909 Female = 158,552 |
| Forced expiratory volume in 1 second (FEV1) Units: L | Data field 20150,, use mean if more than one measure. | All = 297,499 Male = 138,937 Female = 158,562 |

*Category 100057 - Sleep - Lifestyle and environment - Touchscreen - UK Biobank Assessment Centre*

| Variable ID | UK biobank variable location | Number of samples used for interaction |
|---|---|---|
| Sleep duration Units: hours/day | Data field 1160,, use mean of available measures for instance 0 (baseline) only. | All = 393,133 Male = 181,452 Female = 211,681 |

*Category 100072 - Early life factors - Verbal interview - UK Biobank Assessment Centre*

| Variable ID | UK biobank variable location | Number of samples used for interaction |
|---|---|---|
| Birth weight Units: Kg | Data field 20022,, use mean if more than one measure. | All = 227,244 Male = 89,715 Female = 137,529 |

*Category 717 - Biomarkers*

| Variable ID | UK biobank variable location | Number of samples used for interaction |
|---|---|---|
| Apolipoprotein A1 (ApoA) Units: g/L | Data field 30630, use mean of available measures for instance 0 (baseline) only. Standardized using the mean: (x-mean)/sd | |
| High Density Lipoprotein (HDL-c) Units: mmol/L | Data field 30760, use mean of available measures for instance 0 (baseline) only. Standardized using the mean: (x-mean)/sd | |
| Lipoprotein (a) (Lp(a)) Units: nmol/L | Data field 30780, use mean of available measures for instance 0 (baseline) only. Standardized using the mean: (x-mean)/sd | |
| C-Reactive Protein (CRP) Units: mmol/L | Data field 30710, use mean of available measures for instance 0 (baseline) only. Ln transformation, then standardized using the mean: (x-mean)/sd | |
| Low Density Lipoprotein (LDL-c) Units: mmol/L | Data field 30790, use mean of available measures for instance 0 (baseline) only. Standardized using the mean: (x-mean)/sd | |
| Apolipoprotein B (ApoB) Units: g/L | Data field 30640, use mean of available measures for instance 0 (baseline) only. Standardized using the mean: (x-mean)/sd | All = 413,138 Male = 190,454 Female = 222,684 |

*Appendix 1—table 2 Continued on next page*

Appendix 1—table 2 Continued

| Variable ID | UK biobank variable location | Number of samples used for interaction |
|---|---|---|
| *Category of operation procedure codes (OPCS) and hospitalization or death record codes(ICD9/ICD10)* | | |
| Coronary artery disease (CAD) | Prevalent or incident | (cases/controls) All = 413,138 (44,713/368,425) Male = 190,454 (29,910/160,544) Female = 222,684 (14,803/207,881) |
| Myocardial Infarction (MI) | Prevalent or incident | (cases/controls) All = 413,138 (18,559/394,579) Male = 190,454 (13,812/176,642) Female = 222,684 (4,747/217,937) |

**Appendix 1—table 3.** Primers sequence for real-time PCR quantification in HepG2 cells for the *KD-ADCY9* and KD-CETP experimentations.

| Species | Gene | Strain | Sequence |
|---|---|---|---|
| Human | *ADCY9* | Forward | 5' CTGAGGTTCAAGAACATCC 3' |
| | | Reverse | 5' TGATTAATGGGCGGCTTA 3' |
| | *CETP* | Forward | 5' CTACCTGTCTTTCCATAA 3' |
| | | Reverse | 5' CATGATGTTAGAGATGAC 3' |
| | *HBS1L* | Forward | 5' ACAAGAATGAGGCAACAG 3' |
| | | Reverse | 5' AGATACTCCAGGCACTTC 3' |
| | *PGK1* | Forward | 5' GTGGAGGAAGAAGGGAAG 3' |
| | | Reverse | 5' AAGCATCATTGACATAGACAT 3' |

