## [Decision Letter]

**Acceptance summary:**

This study provides a range of evidence for population and sex-specific selection and an epistatic interaction in humans between variants in the genes ADCY9 and CETP of pharmacogenetic importance. Few such interactions have been identified making this a novel finding that motivates more research and methods to look for such effects. The relationship uncovered contributes new insight towards understanding why drug response to a CETP-modulator is affected by genotypes at ADCY9.

**Decision letter after peer review:**

Thank you for submitting your article "A sex-specific evolutionary interaction between ADCY9 and CETP" for consideration by *eLife*. Your article has been reviewed by 3 peer reviewers, and the evaluation has been overseen by a Reviewing Editor and George Perry as the Senior Editor. The following individual involved in review of your submission has agreed to reveal their identity: Eduardo Tarazona-Santos (Reviewer #1).

Essential revisions:

The reviewers identified a number of issues and questions that should be addressed in a revised manuscript. Here is a list of revisions considered essential, many of which come from the individual reviews. The individual reviews are included below, which have both a public review and recommendations for the authors from each reviewer. Please do also respond to all individual reviewers' points and suggestions in your response and revised manuscript.

1. Please include a figure or flowchart that illustrates the overall experimental design and analyses conducted, and which datasets each were performed with. A clear key for the population datasets with sample sizes and the abbreviations that refer to them would be helpful. Please also carefully define and use consistent terms in how datasets are referred to (e.g., First Nation, LIMAA, NAGD, Andean vs. Peruvian).

2. Please include a summary table of the male / female sample sizes, LD values and p-values for the datasets where LRLD was analyzed.

3. Please include the results showing the effect of CETP knockdowns on ADCY9 expression, in addition to the current results showing ADCY9 knockdowns on CETP expression.

4. Please include an additional analysis that considers the effect of sex on gene expression. Please also clarify which analyses did not consider the effect of sex (and why) and for which it was considered but no effect found.

5. Please consider the imputation quality across the different datasets, and especially how it differs across populations to address the concern that imputation from non-representative reference panel populations could be problematic.

6. Please discuss in more detail the proposed interpretation for sex-specific selection. First, the type of selection is not stated in the abstract (only "natural selection"). Second, in the discussion two possibilities are proposed (lines 353-358) but it is not clear if any are supported under the presented evidence. Does the lack of correlation with age rule out a mechanism affecting survival? How strong would selection need to be in a single generation to produce this level of LRLD?

7. Please improve figure visibility and naming as suggested in the specific recommendations below. Also, where possible indicate on the figures where the annotated genes are located, as is only currently shown in Figure 1B. Use consistent naming, e.g., for populations in Figure 1 and 3 (are 3a and b only PEL although the figure title says 1000G?).

*Reviewer #1 (Recommendations for the authors):*

Because the experimental design of the authors is complex, I suggest that they provide a Figure representing and the different steps of the rationales as well as their experiments. It may also help if they number the different steps of the experimental design both in the Figure and the Main Text.

Specific comments:

– the authors should explicitly state which array was used for the genotyping of the LIMAA replication cohort.

*Reviewer #2 (Recommendations for the authors):*

a) I think having a summary figure that goes through and connects the different analyses could potentially be very helpful for the flow of this manuscript. I think the collection of analyses are very impressive, but I find myself continually taking a step back to keep track of the different levels presented. You move from selection analyses, to LD analyses, to gene expression analyses, and finally to phenotype analyses. In addition, at multiple points, there is a further stratification by sex. It may not be immediately intuitive on what a summary figure would be for this, but having something to help readers keep track of how or why these different analyses are connected would be very useful.

b) In the introduction the authors highlight that one of the open questions to be addressed is: "However, the underlying mechanisms linking CETP and ADCY9, located 50 Mb apart on chromosome 16, as well as the *relevance of the rs1967309 non-coding genetic variant are still unclear*." However, the authors do not return to this specific point regarding rs1967309 later in the discussion. I think, since the authors do indeed identify and explore some of the relevance of rs1967309, it would benefit the manuscript to reconnect to this point later at the end of the paper.

c) At the start of the 'sex-specific long-range linkage disequilibrium' section, the authors state the following: "We explored the effect of sex on the LRLD association found between rs1967309 and rs158477. The allele frequencies at rs1967309 were suggestively different between males and females and sex-stratified PBS analyses suggest that the LD block around rs1967309 is differentiated between sexes in the Peruvians…" It is unclear to me if the second sentence is a result of exploring the effect of sex on LRLD associations, or if it is meant to be the justification for exploring the effect of sex. I think clarifying this would be important since it is not otherwise clear why you would move to doing sex-specific analyses, though it does produce an interesting result.

d) I'm wondering why the data regarding the lack of impact CETP knockdowns have on ADCY9 expression is not shown? I think this is an interesting and important result. It helps demonstrate a directionality to the interaction that continually shows up elsewhere. If possible, showing this data would be helpful.

e) Now that you have identified some more relevant tissue types and phenotypes from your epistasis analyses, I am wondering if there are any links that can be made back to your mouse knockouts or any other ADCY9 or CETP knockouts? It seems like with a new spectrum of traits to consider, previous knockout results may make more sense now. Are there any OMIM conditions that now may have some connection to these results?

f) In regards to the CAD outcome findings, I am wondering if you could make use of some of the CAD polygenic risk scores that exist now to further show a link. Specifically, I am wondering whether you could stratify individuals into high and low risk for CAD, and see whether the protective genotype combinations are overrepresented among the low risk individuals vs. high risk (and between the sexes as well). Some example CAD PRSs can be found in the PGS Catalogue: https://www.pgscatalog.org/trait/EFO_0001645/

*Reviewer #3 (Recommendations for the authors):*

I have a few comments and questions below on the approach and results.

1. Population/cohort specific effects. Is there any known ascertainment bias of the cohorts? Skimming the source papers, it looks like for one cohort, they all presented to a clinic for tuberculosis? I'm not sure if I missed it – but is the sex-effect replicated in the UK biobank data? And the linkage missing or apparent in the UK biobank data? One result that is glossed over (again apologies if I missed it in the supplement), but the effect looks reversed/stronger in females in the Andean population (sup Figure 4 d/f)? Is this true or just not significant/too noisy? Also, if this is testable with the other datasets available (since it is a much larger cohort), they could be used as negative controls of sorts. Ie the UK biobank and/or the other native cohorts.

2. Speculation/comments on why there would be a male/female difference in these population (or any). This circles back to the UK biobank question I had earlier. Are frequencies linked to Mt or Y chromosome haplotypes? Is there a way to measure or view this? I understand the RFMix analysis was meant to test for admixture, but would it also show something like this? As it is hard to say definitively or exclude all potential counterfactuals for these results, without including additional datasets from the same region, along with geographic and historical accounts. I wouldn't want to speculate on this of course, but perhaps characterizing the samples a little further in the supplement might help (like a table of sorts, with totals, split by male/female, and any other known demographics). I found it a little difficult to follow all the different populations tested; and which is the "discovery" and which is the "replication" cohort. There are also some discrepancies in the numbers in the text, supplement and captions.

PEL (1000G) LIMAA (?) Andean (NAGD) UK Biobank?

Males 41 2076 ? 2078 54 ?

Females 44 1433 ? 1434 34 ?

Total 85 3509 ? 88 413083

3. Data quality and artefacts. Although this is probably unlikely (but a potential concern/caveat) – wouldn't imputing data from non-represented reference genomes lead to false positives or incorrect phasing? Just curious if there was any check using a different reference panel? I understand this might be out of the scope of this particular analysis to delve into more details about it, but maybe as a sanity check? I'm coming to this from looking through this paper that evaluated potential influences/causes for LRLD:

https://journals.plos.org/plosone/article?id=10.1371/journal.pone.0080754

4. Sex-biased gene expression and eQTLs. Was there any sex-biased expression in the assessed data? Noted sample sizes are still limited here, so I agree this might not be powered. But for some tissues, if you repeat the PEER analysis not only by tissue, but by sex? Correct me if I am wrong, but including sex as a covariate would regress out the sex effects? I'd be interested in seeing the split analysis, if there was a modulating effect that was also sex specific.

Is the expectation that ADCY9 and CETP are negatively correlated with a particular genotype but only in males? Or is there no expectation for sex-specific functional differences too (ie it is either genetic or phenotypic/functional). One way to check could be to look to see if these genes correlated in the GTEx/GEUVADIS/CARTaGENE data.

5. Epistatic/phenotypic data associations. The difference in genotype and MI association is a very interesting effect, but because the data is from a different population, I believe the message/point is kind of lost in the narrative. What are the distributions of the genotypes in the UK biobank? And how do these differ by sex? And what are the distributions of the two genotype combinations? Furthermore, a lot of these traits are correlated, I'm curious to see what those correlations are for these selected traits? Those that aren't associated with the traits but are associated when split by genotype, is that also a numbers effect? I'm not sure I caught that in the tests.

Figure suggestions/comments:

Figure 1 and consistency with labels/IDs. Would it be possible to show the LIMAA cohort here? It is not the "First nation" set -which I believe is the NAGD? The AND subpopulation (Andean) is the one carried on in the analyses, correct? It would be good to highlight these/specify in the caption and/or the figure. Is there a way to also show these frequencies by sex in those subpopulations you later work with? Ie for PEL, AND and LIMAA?

In Figure 1b – Is the recombination rate the lines? Would it be possible to only plot the AMR? It is hard to see/interpret the results here. The others could be in a supplementary Figure

In Figure 1 c – the PBS is also hard to see clearly here. One option would be to smooth the lines out or again, only show the PEL population.

The sex specific results are also super interesting (sup Figure 3 b/d) -> I think moving these also to this main figure shows a clearer point on sex effects.

iHS and recombination rates. I'm not very aware of this measure, but I was curious as to whether the other SNPs that have highly significant iHS values are of relevance, and what the interpretation would be here? Ie the one near the start of the gene seems to have the highest iHS value for the AMR pop. Is that also in high LD with the main SNP of interest in the ADCY9 gene?

Figure 2:Figure 2c, also show the LIMAA cohort as suggested for figure 1?

Figure 4: Could you repeat panel b with the GTEx data at all in the well-powered tissues?

Figure 5:Points in panel a are a little small, perhaps increasing the point size as in panel b/c?

---

## [Author Response]

Essential revisions:The reviewers identified a number of issues and questions that should be addressed in a revised manuscript. Here is a list of revisions considered essential, many of which come from the individual reviews. The individual reviews are included below, which have both a public review and recommendations for the authors from each reviewer. Please do also respond to all individual reviewers' points and suggestions in your response and revised manuscript.1. Please include a figure or flowchart that illustrates the overall experimental design and analyses conducted, and which datasets each were performed with. A clear key for the population datasets with sample sizes and the abbreviations that refer to them would be helpful. Please also carefully define and use consistent terms in how datasets are referred to (e.g., First Nation, LIMAA, NAGD, Andean vs. Peruvian).

We agree that our overall experimental design is complex, and that it can be confusing to the reader, as exemplified by comments from all 3 reviewers.

To improve this aspect, we added main Figure 1 showing for each step the analyses performed, the datasets used, and the key results obtained (needed to move forward to next steps). We also identified the steps where we did consider sex in the analysis (green coloring). We also modified the main text to add references to the different steps throughout (see track changes). To increase clarity about the population datasets, we added main Table 1 containing information about each cohort used, defining clearly the abbreviations used, ethnicity, number of samples, female percentage, age (if information was available) and the citation. We carefully checked the terms and abbreviations in the main text and modified it where appropriate (see track changes).

Finally, in response to reviewer 3 (R3) comment R3.2, we have verified all numbers reported throughout the text, making sure it is clear if the sample sizes are reported before or after quality control (QC) steps (see track changes).

2. Please include a summary table of the male / female sample sizes, LD values and p-values for the datasets where LRLD was analyzed.

We have added Appendix – Table 1 to clarify the nature of cohorts used for the LRLD analyses, along with male/female sample sizes, LD values (r^2^) and empirical p-values. This is also in response to comment R3.2.

3. Please include the results showing the effect of CETP knockdowns on ADCY9 expression, in addition to the current results showing ADCY9 knockdowns on CETP expression.

Our initial knockdowns (KD) experiments included only one replicate, and in that first step, significant results were only detected for ADCY9 KD, which led us to validate this result further (by adding 5 replicates, performing western blot, and ultimately RNAseq). In the CETP-KD, we did not see any significant ADCY9 expression changes, but we did not want a figure with N=1, hence the “data not shown” mention in the previous version.

In response to your request and to comment R2.4, we have now increased the number of replicates to N=4 for CETP-KD and to N=4 for CETP overexpression, to report these results more appropriately. The results remain unchanged, the differences in ADCY9 expression between CETP-KD/overexpressed and controls are still not significant and are now reported in Figure 5—figure supplement 1. We also repeated with N=4 an experiment for *ADCY9* overexpression, which is also not significant (Figure 5—figure supplement 1). The ADCY9 KD experiment was also repeated in this batch and remained significant (p<0.0026).

The following changes to the main text were made:

In Results:

– “This increased expression was validated by qPCR, and western blot also showed increased CETP protein product (Methods, Figure 5—figure supplement 1a,b, Appendix 1), but its overexpression did not significatively modulate CETP expression (Figure 5—figure supplement 1c).”

– L. 307 P. 11, we replaced “data not shown” by Figure 5—figure supplement 1

– in Material and Methods:

– “Silencer Select siRNA against CETP (Ambion 4392420 ID 2933)”

– We added a section: “Overexpression of ADCY9 and CETP genes in HepG2 cell line”

“For ADCY9 and CETP overexpression experiments, 500 000 cells in 2 ml of medium in a six-well plate were transfected using 1 ug of pEZ-M46-AC9 or pEZ-M50-CETP plasmids (GeneCopoeiaTM) with 5 ul of Lipofectamine 2000 reagent (Invitrogen cat # 11668-019) for 72h. Total RNA was extracted from transfected HepG2 cells using RNeasy Plus Mini Kit (Qiagen cat #74136) in accordance with the manufacturer’s recommendation (Appendix – Table 3, Appendix 1).”

– We added a clarification about the sex of donor of the HepG2 cell line:

“a cell line derived from the liver tissue of a 15-year-old male donor (78)”

We also updated the legend of the Figure 5—figure supplement 1:

– “(a) Relative mRNA expression of CETP of HepG2 cells 72h post-transfection with siRNA against human ADCY9 (si1039). qPCR assay was normalized with PGK1 and HBS1L genes, n=5 independent experiments, (p-value=0.0026 from t-test). (b) Quantification of CETP protein by Western blot assay, 200 ml of cell media (concentrated with Amicon ultra 0.5 ml 10 kDA units) from cells transfected with siRNA against human ADCY9 (si1039), were separated on 10% TGX-acrylamide gel and transferred to PVDF membrane. CETP protein expression was determined using a primary antibody rabbit monoclonal anti-CETP (Abcam, ab157183) 1:1000 (3% BSA, TBS, Tween 20 0.5%) O/N 4oC, followed by HRP-conjugated secondary antibody goat anti-rabbit 1:10 000 (3% BSA) 1h RT. Figure b represents densitometry analysis of n=3 experiments, p-value=0.0029 from t-test. (c,e) Relative mRNA expression of (c) CETP and (e) ADCY9 genes in HepG2 cells post-transfection with pEZ-M50-CETP (overexpression of CETP) or pEZ-M46-ADCY9 (overexpression of ADCY9 ) plasmids. qPCR assay was normalized with PGK1 and HBS1L genes, n=4 independent experiments. (d) Relative mRNA expression of ADCY9 of HepG2 cells 72h post-transfection with siRNA against human CETP. qPCR assay was normalized with PGK1 and HBS1L genes, n=4 independent experiments.”

4. Please include an additional analysis that considers the effect of sex on gene expression. Please also clarify which analyses did not consider the effect of sex (and why) and for which it was considered but no effect found.

This is a very important point that we carefully considered in this new version of the manuscript. Initially, we performed a preliminary sex-stratified analysis in our RNAseq discovery cohort (GEUVADIS) which was not very conclusive likely due to low sample size. We did not pursue this in our replication cohorts (GTEx and CaG). Thanks to this request and comment R3.4, we calculated PEER factors by stratifying the datasets by sex (sPEER factors) and performed further analyses in GTEx and CaG that produced very interesting new results, in line with the sex-specific nature of our other results, that we are happy to report.

We see differences between sex in these datasets. In summary, most sex-combined results were driven by males (including the CaG result) but stratifying by sex in GTEx samples revealed additional tissues in which females only show a significant interaction, and very intriguingly, the sign of the interaction effect is reversed in this case. We tested for a three-way interaction in tissues harboring a convincing sex-specific pattern in stratified analyses. We note that some tissues in GTEx did not have enough samples in one sex (<50 samples), so we did not include them in the sex-stratified analyses. This number was selected to maximize the number of tissues for which we tested the interaction while having high chances to have samples in all genotype combinations. Also, with this N, there are still some tissues where the rarest combinations in Europeans (AA-GG or AA-GA) are missing, so going below is not informative. If the reviewers/editors think that a higher threshold would be better, we can always remove the plots for some of the tissues. See Author response table 1 for the list of tissues excluded from our sex-specific analyses.

**Author response table 1. sa2table1:** 

Organ	Sex	Number of samples
Brain-Amygdala	Female	34
Brain-Anterior Cingulate Cortex (BA24)	Female	39
Brain-Caudate basal ganglia	Female	45
Brain-Cerebellar Hemisphere	Female	46
Brain-Frontal Cortex (BA9)	Female	44
Brain-Hippocampus	Female	45
Brain-Hypothalamus	Female	44
Brain-Nucleus accumbens basal ganglia	Female	49
Brain-Putamen (basal ganglia)	Female	38
Brain-Spinal cord cervical c-1	Female	42
Brain-Substantia nigra	Female	28
Cells-EBV-transformed lymphocytes	Female	43
Kidney-Cortex	Male/Female	48/17
Minor Salivary Gland	Female	32

We added these additional results to the “Epistatic effects on *CETP* gene expression” section of the main text and “Expression Quantitative Trait Loci (eQTL) analysis for rs1967309 and rs158477” section of Appendix 1. We modified Figure 5 (previously Figure 4) to add two box-plot panels (c,d), similar to previous Figure 4b, to include the most interesting results (skin in males, tibial artery in females), as well as Figure 5—figure supplement 2 (child) reporting on these analyses. Sex-stratified p-value plots for all tissues based on number of sPEER factors are included in new Appendix 1-figure 9.

– We added new subfigures in the Figure 5

– The new subfigures of Figure 5 (c,d) represent the interaction between our SNPs for CETP expression in Skin-Sun exposed in male (c) and Artery Tibial in female (d) in GTEx. Both have the interaction and enough samples to create a plot like the one of GEUVADIS (b), answering R3 request to repeat panel b with the GTEx data (R3.6)

– The legend was modified: “Figure 5. Effect of ADCY9 on CETP expression. (a) Normalized expression of ADCY9 or CETP genes depending on wild type (WT) and ADCY9-KD in HepG2 cells from RNA sequencing on five biological replicates in each group. P-values were obtained from a two-sided Wilcoxon paired test. qPCR and western blot results in HepG2 are presented in Figure 5—figure supplement 1. (b,c,d) CETP expression depending on the combination of rs1967309 and rs158477 genotypes in (b) GEUVADIS (p-value=0.03, ß=-0.22, N=287), (c) GTEx-Skin Sun Exposed in males (p-value=0.0017, ß=-0.32, N=330) and in (d) GTEx-Tibial artery in females (p-value=0.026, ß=0.38, N=156), for individuals of European descent according to principal component analysis. P-values reported were obtained from a two-way interaction of a linear regression model for the maximum number of PEER/sPEER factors considered. Figure 5—figure supplement 2 show the interaction p-values depending on number of PEER/sPEER factors included in the linear models.”

– We added a sentence in the abstract:

L 32-33, P. 2 “Analyses of RNA-seq data further suggest an epistatic interaction on CETP expression levels between the two SNPs in multiple tissues, which also differs between males and females.”

– We added a section in the main text for the sex-analysis results and discussion:

“Given the sex-specific results reported above, we stratified our interaction eQTL analyses by sex. We observed that the interaction effect on CETP expression in CaG whole blood samples (N_male_=359) is restricted to male individuals, and, despite low power due to smaller sample size in GEUVADIS, the interaction is also only suggestive in males (Appendix 1-figure 7c,d). In GTEx, most well-powered tissues that showed a significant effect in the sex-combined analyses also harbor male-specific interactions (Appendix 1-figure 9). For instance, GTEx skin male samples (N_male_=330) show the most significant male-specific interaction effects, with the directions of effects replicating the sex-combined result in GEUVADIS (an increase of CETP expression for each rs158477 A allele in rs1967309 AA individuals) albeit with an observable reversal of the direction in rs1967309 GG individuals (decrease of CETP expression with additional rs158477 A alleles) (Figure 5c, Figure 5—figure supplement 2a). However, significant effects in females are detected in tissues not previously seen as significant for the interaction in the sex-combined analysis, in the tibial artery (Figure 5d, Figure 5—figure supplement 2) and the heart atrial appendage (Appendix 1-figure 9). For tissues with evidence of sex-specific effects in stratified analyses, we also tested the effect of an interaction between sex, rs158477 and rs1967309 (Methods) on CETP expression: the three-way interaction is only significant for tibial artery (Figure 5—figure supplement 2).”

“The significant interaction effects on CETP expression vary between sexes in amplitude and direction, with most signals driven by male samples, but significant interaction effects observed in females only, despite sample sizes being consistently lower than for males. Notably, in the tibial artery and heart atrial appendage, two tissues directly relevant to the cardiovascular system, the female-specific interaction effect on CETP expression is reversed between rs1967309 genotypes AA and GG, compared to the effects seen in males in skin and brain tissues. Given our ADCY9-KD were done in liver cell lines from male donors, future work to fully understand how rs1967309 and rs158477 interact will focus on additional experiments in cells from both male and female donors in these relevant tissues.”

“The female-specific eQTL interaction results in arteries and heart tissues further suggest a link with the cardiovascular system, and the phenotype association results support further this hypothesis.”

– We added in Methods:

“To take into account hidden factors, we calculated PEER factors (75) on the normalized expressions, on all samples and stratified by sex (sPEER factors). To detect eQTL effects, we performed a two-sided linear regression on ADCY9 and CETP expressions using R (v.3.6.0) (https://www.r-project.org/) with the formula lm(p ∼rs1967309*rs158477+Covariates) for evaluating the interaction effect, lm(p ∼rs1967309+rs158477+Covariates) for the main effect of the SNPs and lm(p ∼rs1967309*rs158477*sex+Covariates) for evaluating the three-way interaction effect.”

“Reported values in the text are for five PEER factors in GEUVADIS, ten PEER factors in CARTaGENE, 25 sPEER for skin sun exposed in male and 10 sPEER for artery tibial in female in GTEx.”

We also added several paragraphs in the Appendix 1 in the sections of “ADCY9 and CETP expression quantification from RNAseq data” and “Expression Quantitative Trait Loci (eQTL) analysis for rs1967309 and rs158477”:

Appendix 1: “For all sex-stratified analysis, we kept sex-stratified tissues that had at least 50 samples, and recomputed PEER factors with samples from only one sex (which we term sPEER factors).”

Appendix 1: “Since the selective pressure differ between sexes, we stratified our expression analysis by sex. For CETP expression analysis, there are no consistent signals for GEUVADIS, possibly reflecting lack of power or that there is no sex-specific effect in lymphoblastoid cell lines (Appendix 1-figure 7c). However, in CaG, the significant interaction found is present only in male (again for the genotypic coding, Appendix 1-figure 7d). In GTEx, we see significant interaction effects in males in tissues that had signals with sex-combined, such as brain hippocampus (Nmale=105), hypothalamus (Nmale=112) and spinal cord cervical (Nmale=72), skin sun-exposed for CETP (Nmale=330) (Figure 5d, Appendix 1-figure 9). We note that for most brain tissues, the low sample size in females does not allow to conclude on the presence of an interaction effect in that sex. In these tissues, the direction of the effect in males is reversed compared to what is observed in GEUVADIS with sexes combined (Figure 5b), whereas the highly significant result in skin shows an effect consistent with the sex-combined GEUVADIS result (p-value=0.0017, ß=-0.32). More specifically, in the sun-exposed skin samples, in rs1967309 AA males, copies of the rs158477 A allele increase CETP expression by 0.49 (95% CI 0.12-0.87) on average. In rs1967309 AG males, the effect of rs158477 is null (p-valueAG=0.33) and the effect of the rs158477 A allele is suggestive in rs1967309 GG individual (p-valueGG=0.10) with a decrease of the CETP expression. Conversely, if we look at the effect of rs1967309 on CETP expression in skin sun-exposed when we stratified by rs158477 in males, SNP rs1967309 is neither a significant eQTL of CETP in GG of rs158477 in males (p=0.11, n=89) nor for GA (p=0.65, n=164), but is a significant eQTL for AA males (p=0.026, ß=-0.46, OR=[-0.87, -0.06], n=76). We also identified new tissues where the interaction is either suggestive or significant in females, in artery tibial (Nfemale=156), heart atrial appendage (Nfemale=97), spleen (Nfemale=69) and stomach (Nfemale=105) (Appendix 1-figure 9), with an effect reversed compared to the initial GEUVADIS result (Appendix 1-figure 7a). For the pituitary tissue, it is significant in both sex (additive coding in female and genotypic coding in male, Nmale=156, Nfemale=63), but the direction of the additive coding is reversed between sexes, possibly explaining why the sex-combined analysis did not show any signal. We note that the newly discovered signals are mainly for females, indicating that the signal was hidden by the male effects (or absence of effects), likely because of higher sample sizes.”

As shown in our new Figure 1, most analyses consider the effect of sex (green boxes), except:

– iHS, this statistic is powerful to detect hard selective sweeps and is not designed to detect sex-specific selection (which will likely look like an incomplete soft sweep) and did not show significant results in the PEL cohort.

– RFMix inference analyses of local ancestry did not consider males and females separately, it would only result in loss of precision if we split the reference panel in this way. However, we did check whether there are significant differences between males and females in global ancestry and did not find significant results. This was reported in Results.

– KD/over-expression experiments, which were done in HepG2 cells from a male donor. We have added this information in the Methods and discussed future steps in cell lines from male and female donors in the Discussion.

“The human liver hepatocellular HepG2 cell line was obtained from ATCC, a cell line derived from the liver tissue of a 15-year-old male donor”

“Given our ADCY9-KD were done in liver cell lines from male donors, future work to fully understand how rs1967309 and rs158477 interact will focus on additional experiments in cells from both male and female donors in these relevant tissues.”

5. Please consider the imputation quality across the different datasets, and especially how it differs across populations to address the concern that imputation from non-representative reference panel populations could be problematic.

Indeed, we had reported that the INFO score for our two SNPs imputed in LIMAA and NAGD with the Haplotype Reference Consortium (HRC) was lower than our 0.8 initial threshold in Appendix, section “Pre-processing of the LIMAA cohort”. This can be expected in an admixed population, when the reference panel does not include a close enough population, and even more for differentiated SNPs as it is the case here, for which the allele frequencies would be misaligned with the reference panel frequencies. Therefore, our approach was to lower the threshold to 0.7 (which is still considered a high score in most studies).

However, based on this comment and comment R3.3, we performed additional analyses to make sure that the lower INFO score was due to low confidence because of the factors described above, and not to incorrect imputed genotypes. We thus redid the imputation of LIMAA and NAGD with the TOPMED reference panel. For our two SNPs, rs1967309 and rs158477, the r^2^ score for imputation with TOPMED is now higher (r^2^>0.9 for both), meaning that a more diversified reference panel helped the imputation of this admixed population. Furthermore, when we look at the allele frequencies and genotypes, they are highly similar (see Author response table 2, only 5% and 2% of individual allele mismatches in LIMAA for rs1967309 and rs158477, respectively). We recomputed the genotype correlation (LRLD) with these new data in LIMAA, and the value obtained is r^2^ = 0.0047 (*vs*. 0.0046 before).

**Author response table 2. sa2table2:** 

Database	SNP - Variant	TOPMED imputation	Sanger Imputation Server - Haplotype Reference Consortium (in manuscript)
LIMAA	rs1967309 A	77%	76%
	rs158477 G	79%	79%
NAGD-Andean	rs1967309 A	77%	77%
	rs158477 G	74%	73%

Given these small differences, and the time it would require redoing all figures and null distributions with the new imputed data, we decided not to change our analyses in the manuscript and keep imputation with HRC, while adding these considerations to Appendix 1 in the section “Pre-processing of the LIMAA cohort” and “Genotype association between rs1967309 and rs158477 in LIMAA”.

Appendix 1: “To make sure imputation quality did not impact our results because of incorrectly imputed genotypes, we redid the imputation of LIMAA with the TOPMED reference panel (https://imputation.biodatacatalyst.nhlbi.nih.gov/#!). The imputation r^2^ score with TOPMED is higher than 0.9 for both, and only very limited differences in imputed genotypes are seen (only 5% and 2% of individual allele mismatches in LIMAA for rs1967309 and rs158477, respectively for the 3,243 individuals)”

Appendix 1: “Finally, we also considered how the imputation quality in LIMAA could affect the main result of LRLD, because of imputation from non-representative reference panel populations is known to be problematic. We recomputed the genotype correlation (r^2^) in LIMAA with our two SNPs imputed with the TOPMED panel, a more representative panel than the Haplotype Reference Consortium initially used. The value obtained is 0.0047 compared to 0.0046 before, showing that imputation quality is unlikely to have affected our results.”

6. Please discuss in more detail the proposed interpretation for sex-specific selection. First, the type of selection is not stated in the abstract (only "natural selection"). Second, in the discussion two possibilities are proposed (lines 353-358) but it is not clear if any are supported under the presented evidence. Does the lack of correlation with age rule out a mechanism affecting survival? How strong would selection need to be in a single generation to produce this level of LRLD?

We are delighted that the editors and reviewers agree with us about the importance of the sex-specific result we present here. We preferred to be cautious in our first version in our interpretation of this result, in order not to oversell it. But in the light of the newest gene expression results presented in favor of the sex-specific effects, and given the positive feedback received, we are now more comfortable on expanding on this aspect.

We agree that our interpretation of the age-dependant result in the discussion was not clear, as the analysis conducted indeed does not really allowed us to conclude: we did this analysis with what we had at hand in LIMAA, and if we had seen an age-dependent effect, it would have been support for the differential survival hypothesis, but in the case of a non-significant result, it does not necessarily exclude this possibility. Indeed, we may be underpower to detect an age-effect, or simply not looking at the appropriate timeframe, and a pediatric cohort may be needed. To test the in-utero hypothesis, a cohort of Peruvian newborns would allow to evaluate if the genotype correlation is already present at birth, or a cohort of stillborn babies would allow to evaluate if a genotype combination is more frequent in those. We now expanded in the text on how this question could be further resolved in the future.

Finally, we thought a lot about the important question of how strong selection would need to be in a single generation to produce this level of LRLD. Because this is, to the best of our knowledge, the first instance reported of such a phenomenon, there is no theoretical ground on which to base a formal answer to this question and we would need to perform a carefully designed simulation study to answer this question. This is beyond the scope of the present work, but it is an exciting avenue of research. In our opinion however, selection probably needs to be quite strong in order to produce this effect and, despite searching for outstanding causes for infant/children mortality in Peru and looking at male/female ratio statistics for children born in Peru, we did not find any hints on what could be happening. On the other hand, in utero and sperm-specific selection are thought to result from stronger selection, which makes us think that this is the most likely explanation. But why would it differentially impact boys and girls in this case (sex hormones?), and why specifically in Peru (pregnancy in altitude?), remain mysteries.

The changes made in the text are as follow:

– In the abstract:

“rs1967309 region exhibits signatures of positive selection in several human populations”

“We propose that ADCY9 and CETP coevolved during recent human evolution due to sex-specific selection.”

– In the discussion:

“Such two-gene selection signature, where only males show strong LRLD, can happen if a specific genotype combination is beneficial in creating males (through differential gamete fitness or in utero survival, for example) or if survival during adulthood is favored with a specific genotype combination compared to other genotypes. In the case of age-dependent differential survival, the genotypic association is expected to be weaker at younger ages, however the LRLD signal between rs1967309 and rs158477 in the LIMAA cohort did not depend on age neither in males nor in females (Appendix 1). Since very few individuals were younger than 20 years old, it is likely that the age range in this cohort is not appropriate to distinguish between the two possibilities. This age-dependent survival therefore remains to be tested in comparison with pediatric cohorts of Peruvians: if the LRLD signal is absent in newborns for example, it will suggest a strong selective pressure acts early in life on boys. To specifically test the in-utero hypothesis, a cohort of stillborn babies with genetic information could allow to evaluate if the genotype combination is more frequent in these. Lastly, it may be that the evolutionary pressure is linked to the sex chromosomes (69,70), and a three-way interaction between ADCY9, CETP and Y chromosome haplotypes or mitochondrial haplogroups remains to be explored.”

“Furthermore, not much is known about the strength of this type of selection, and simulations would help evaluate how strong selection would need to be in a single generation to produce this level of LRLD.”

7. Please improve figure visibility and naming as suggested in the specific recommendations below. Also, where possible indicate on the figures where the annotated genes are located, as is only currently shown in Figure 1B. Use consistent naming, e.g., for populations in Figure 1 and 3 (are 3a and b only PEL although the figure title says 1000G?).

We improved the quality of all main figures, added gene annotation plots for the several figures (iHS, PBS and LRLD) and used consistent naming throughout. We also switch the wording men/women to males/females to be more consistent in figures and text. We changed all the naming of “Figure” to “Figure” for main and Appendix 1-figures. We also moved several Appendix figures to “child” figures (“figure supplement”).

We also followed the suggestion from reviewers below, and made the following changes:

Figures

– Added a new Figure 1 (see response to E.1 above)

– Figure 1 (now Figure 2)

– Figure 1a (now Figure 2a): We changed the name of the Native Amerind to NAGD to be more consistent in our naming;

– Figure 1b (now Figure 2b) : We kept only the recombination rate of the AMR population and removed the others, since that was confusing (and didn’t give much information since they are all very similar). This rate was plotted mainly to show the LD block around rs1967309;

We added a sentence to specified that the LD block is consistent:

“The genetic variant rs1967309 is located in intron 2 of ADCY9, in a region of high linkage disequilibrium (LD), in all subpopulations in the 1000 Genomes Project (1000G), and harbors heterogeneous genotype frequencies across human populations (Figure 2a).”

– Figure 1c (now Figure 2c): In an attempt to be clearer, we change the representation with lines to a scatter plot (points) and separated by subpopulation. We emphasized rs1967309 values and changed the color, since we added PBS in another figure that used those colors for another meaning. We also removed the different shape of the point for populations other than the Peruvian, and, for the Peruvian, we also removed the special shape to put a black circle to emphases it. (in response to the figure suggestions of R3)

– We also updated the information figure legend (see track changes)

– Figure 2 (now Figure 3):

Figure 2c (now Figure 3c): We changed the name of the Native Amerind to NAGD;

Added the meaning of the black symbols and the blue lines/box under the figures.

We added 2 child figures

– Figure 3 (now Figure 4):

We added the number of individuals in the title

Figure 3a-b (now Figure 4a,b):

We changed the identification 1000G to PEL. We also updated the information in the figure legend of the figure (see track changes)

– Added the meaning of the black symbols and the blue lines/box under the figures.

– We added 3 child figures

– We added the figures of genotype frequency distribution by sex for rs1967309 and rs158477 as a child Figure 4—figure supplement 1 (in response to the figure suggestions of R3)

– Figure 4 (now Figure 5)

We added two plots to this figure (see response to E.4 above).

We added 2 child figures

We added new figures as a child Figure5—figure supplement 2 of a 2-way and 3-way interaction between our SNPs (and sex) for the two tissues that we added in the parent Figure 5.

– Figure 5 (now Figure 6)

Figure 5a (now Figure 6a): We enlarged the size of the plot to make them easier to read.

We added 2 child figures

Tables:

– Added the ‘Key Resources Table’ at the beginning of the Material and Methods section

– Added a table containing information for each cohort as a Table 1 in the main manuscript (in response to E1 and R3.2)

Supplementary figures:

– We removed/modified/added Supplementary Figures:

– To the Appendix 1-figure 2, we added the PCA and UMAP figures to the RFMix result, previously Supplementary Figure 7. For the PCA/UMAP plots, we changed the color to be colorblind friendly.

– From the Appendix 1-figure 3, we moved the subfigure b,d to be a child Figure 4—figure supplement 2 (in response to the figure suggestions of R2). For each PBS plot, we also changed the color to avoid confusion since the initial color were used to identified SNPs in the parent Figure 4.

– To Appendix 1-figure 4, we added gene under LRLD plot and changed labels of the legend (sex True sex labels; Shuffled sex Shuffled sex labels), previously (Supplementary Figure 8)

– From the Supplementary Figure 4 and 6 (LRLD in Andean of NAGD and LIMAA respectively), we moved the subfigure a,b (sex-combined) to become a child Figure 3—figure supplement 1. We also moved d,f of Supplementary Figure 4 (sex-stratified of ADCY9 in Andean of NAGD) to become a child Figure 4—figure supplement 3. We merged the other subfigures to become the new Appendix 1-figure 5.

– We moved the previous Supplementary Figure 9 to a child Figure 5—figure supplement 1. To this figure, we also added the CETP-KD result and CETP and ADCY9 overexpression in response to (E3 and R2.4)

– Appendix 1-figure 6, we added a new figure showing the correlation between ADCY9-CETP across tissues in GTEx (see response to R3.4)

– To the Supplementary Figure 14 (now Appendix 1-figure 7), we added the number of individuals in the title (which represents the number after filtering out individuals missing the genotype of rs1967309 and/or rs158477). We also added results of the analyses of the interaction between SNPs when adding sPEER factors for the stratifying by sex analysis.

– Appendix 1-figure 8: Previously Supplementary Figure 15

– We added a new Appendix 1-figure 9 which are the results from the analysis of the interaction on CETP stratified by sex depending on the number of sPEER added in the regression (to answer E4 and R3.4)

– We moved the previously Supplementary Figure 16 and 17 as child figures Figure 6—figure supplement 1 and 2 respectively.

Appendix – tables:

– We added a new Appendix – Table 1 (see response to E.2 above)

– We moved the previous Supplementary Table 1 to Appendix – table 2, and Supplementary Table 2 to Appendix – Table 3, in concordance with the order mentioned in the main text.

*Reviewer #1 (Recommendations for the authors):Because the experimental design of the authors is complex, I suggest that they provide a Figure representing and the different steps of the rationales as well as their experiments. It may also help if they number the different steps of the experimental design both in the Figure and the Main Text.*

We thank the reviewer for highlighting this necessity for clarification. We agree with the reviewer that it may be difficult to follow all our steps and so, we added main Figure 1 showing for each step (1 to 4), the analyses performed, the datasets used, and the key results obtained (needed to move forward to next steps). We also identified the steps where we did consider sex in the analysis (green coloring). We also modified the main text to add references to the different steps throughout. This is also in response to essential comment E.1 above.

Specific comments:– the authors should explicitly state wich array was used for the genotyping of the LIMAA replication cohort.

Indeed, we did not mention this information in our methods. We have now added this text in the Methods section:

– “This cohort was genotyped with a customized Affymetric LIMAAray containing marker optimized for Peruvian-specific rare and coding variants.”

Reviewer #2 (Recommendations for the authors):a) I think having a summary figure that goes through and connects the different analyses could potentially be very helpful for the flow of this manuscript. I think the collection of analyses are very impressive, but I find myself continually taking a step back to keep track of the different levels presented. You move from selection analyses, to LD analyses, to gene expression analyses, and finally to phenotype analyses. In addition, at multiple points, there is a further stratification by sex. It may not be immediately intuitive on what a summary figure would be for this, but having something to help readers keep track of how or why these different analyses are connected would be very useful.

We thank the reviewer for highlighting this necessity for clarification. We agree with the reviewer that it may be difficult to follow all our steps and so, we added main Figure 1 showing for each step (1 to 4), the analyses performed, the datasets used, and the key results obtained (needed to move forward to next steps). We also identified the steps where we did consider sex in the analysis (green coloring). We also modified the main text to add references to the different steps throughout. This is also in response to essential comment E.1 above.

b) In the introduction the authors highlight that one of the open questions to be addressed is: "However, the underlying mechanisms linking CETP and ADCY9, located 50 Mb apart on chromosome 16, as well as the relevance of the rs1967309 non-coding genetic variant are still unclear." However, the authors do not return to this specific point regarding rs1967309 later in the discussion. I think, since the authors do indeed identify and explore some of the relevance of rs1967309, it would benefit the manuscript to reconnect to this point later at the end of the paper.

We thank the reviewer for this comment. Despite our study demonstrating that rs1967309 is likely involved in a functional mechanism related to *CETP* expression, and that this mechanism implicates sex as a modulator, the exact mechanism remains unknown. Therefore, we avoided to speculate too much in our first version of the manuscript, but we agree with the reviewer that we should return to this point to wrap up the article. We have now added a paragraph that goes back to the initial discovery of rs1967309 and discusses what our results bring to this important ongoing research.

In Discussion:

“Despite these limitations, our results support a functional role for the intronic SNP rs1967309, likely involved in a molecular mechanism related to CETP expression, but this mechanism seems to implicate sex as a modulator in a tissue-specific way, which complicates greatly its understanding. In the dal-OUTCOMES clinical trial, the partial inhibitor of CETP, dalcetrapib, did not decrease the risk of cardiovascular outcomes in the overall population, but rs1967309 in the ADCY9 gene was associated to the response to the drug, which benefitted AA individuals (4). Interestingly, rs1967309 AA is found in both the male and female beneficial combinations of genotypes for CAD, the same that are enriched in Peruvians, but without taking rs158477 and sex into account, this association was masked. The modulation of CETP expression by rs1967309 could impact CETP’s functions that are essential for successfully reducing cardiovascular events. The rs158477 locus could be a key player for these functions, and dalcetrapib may be mimicking its impact, hence explaining the pharmacogenomics association. Furthermore, in the light of our results, some of these effects could differ between men and women (73), which may need to be taken into consideration in the future precision medicine interventions potentially implemented for dalcetrapib.”

c) At the start of the 'sex-specific long-range linkage disequilibrium' section, the authors state the following: "We explored the effect of sex on the LRLD association found between rs1967309 and rs158477. The allele frequencies at rs1967309 were suggestively different between males and females and sex-stratified PBS analyses suggest that the LD block around rs1967309 is differentiated between sexes in the Peruvians…" It is unclear to me if the second sentence is a result of exploring the effect of sex on LRLD associations, or if it is meant to be the justification for exploring the effect of sex. I think clarifying this would be important since it is not otherwise clear why you would move to doing sex-specific analyses, though it does produce an interesting result.

Thanks for asking to clarify this aspect, we now see how exploring the effect of sex was not well motivated in the text. Yes, the reviewer is right that the reason why we did these analyses was because we observed suggestive differences in allele frequencies at rs1967309 in Peru. We had computed F_ST_ between males and females, which was high in the rs1967309 region compared to the rest of the gene (it was in the top 1% of values for PEL, see Author response image 1, but not genome-wide significant after accounting for allele frequency, p=0.11), but the sex-stratified PBS result was striking: the differentiation signal completely disappeared in females. This is what motivated the LRLD sex-stratified analyses that followed. We have now clarified this in the text.

**Author response image 1. sa2fig1:** Weir and Cockerham F_ST_ between males and females in PEL. Horizontal line represents the 99th percentile value for this population (for chromosome 16).

In “Sex-specific long-range linkage disequilibrium signal” section in Results:

“Because the allele frequencies at rs1967309 were suggestively different between males and females (Figure 4—figure supplement 1), we performed sex-stratified PBS analyses, which suggested that the LD block around rs1967309 is differentiated between sexes in the Peruvians (Figure 4—figure supplement 2, Appendix 1). We therefore explored further the effect of sex on the LRLD association found between rs1967309 and rs158477 and performed sex-stratified LRLD analyses.”

d) I'm wondering why the data regarding the lack of impact CETP knockdowns have on ADCY9 expression is not shown? I think this is an interesting and important result. It helps demonstrate a directionality to the interaction that continually shows up elsewhere. If possible, showing this data would be helpful.

The reviewer is right that this is an important result that led us to the hypothesis of modulation of *CETP* expression through ADCY9 SNP. Our initial knockdowns (KD) experiments included only one replicate, and in that first step, significant results were only detected for ADCY9 KD, which led us to validate this result further (by adding N=5 replicates, performing western blot, and ultimately RNAseq). In the CETP-KD, we did not see any significant ADCY9 expression changes, but as we did not pursue the negative results further, we initially chose not to include a figure with N=1, hence the “data not shown” mention.

We have now increased the number of replicates to N=4 for CETP-KD and to N=4 for CETP overexpression, to report these results more appropriately. The results remain unchanged, the differences in *ADCY9* expression between CETP-KD/overexpressed and controls are still not significant and are now reported in Figure 5—figure supplement 1. We also repeated with N=4 an experiment for ADCY9 overexpression, which is also not significant (Figure 5—figure supplement 1). We modified the text accordingly, as reported in essential comment E.3.

e) Now that you have identified some more relevant tissue types and phenotypes from your epistasis analyses, I am wondering if there are any links that can be made back to your mouse knockouts or any other ADCY9 or CETP knockouts? It seems like with a new spectrum of traits to consider, previous knockout results may make more sense now. Are there any OMIM conditions that now may have some connection to these results?

Among tissues showing an interaction between *ADCY9* and *CETP* genes, there is the hypothalamus which regulates the central nervous system, including respiratory and pulse rates. In the mice experiment in Rautureau et al., 2018, we previously saw a modulation of both these phenotypes, suggesting a role of both genes in regulation of the autonomic nervous system, which we will prioritize in future work. We did not find any condition in OMIM linked to tissues where the interaction is present for CETP expression. The only condition linked with CETP is the hypercholesterolemia (mainly HDL), which can be associated with cardiovascular disease, but, in our phenotype association, the interaction is not significant for HDL (Figure 6).

– We added a few sentence in the discussion to link with our previous mice KD:

“In a previous study, we showed that inhibition of both Adcy9 and CETP impacted many phenotypes linked to the ANS in male mice (6), but in the light of our results, these experiments should be repeated in female mice. The function of ANS is important in a number of pathophysiological states involving the cardiovascular system, like myocardial ischemia and cardiac arrhythmias, with significant sex differences reported (62-64).”

f) In regards to the CAD outcome findings, I am wondering if you could make use of some of the CAD polygenic risk scores that exist now to further show a link. Specifically, I am wondering whether you could stratify individuals into high and low risk for CAD, and see whether the protective genotype combinations are overrepresented among the low risk individuals vs. high risk (and between the sexes as well). Some example CAD PRSs can be found in the PGS Catalogue: https://www.pgscatalog.org/trait/EFO_0001645/

Thank you for sharing this idea, this is a very interesting comment that we will consider in our follow-up work. We however feel this is beyond the scope of the present paper.

Reviewer #3 (Recommendations for the authors):I have a few comments and questions below on the approach and results.1. Population/cohort specific effects. Is there any known ascertainment bias of the cohorts? Skimming the source papers, it looks like for one cohort, they all presented to a clinic for tuberculosis?

The LIMAA cohort is indeed from a tuberculosis study but includes many individuals who did not have tuberculosis. To make sure there was no ascertainment bias in terms of genetic ancestry, we performed PCA and UMAP analyses as QC and saw no separation according to disease state. If there was signal for genetic predisposition to tuberculosis here, this would affect a restricted number of loci, and as there’s no evidence that our loci contribute to tuberculosis, we do not think this is a problem.

We added the following text to supplementary text, section “Comparison between Peruvian cohorts”

– Appendix 1: “Also, the LIMAA cohort is initially from a tuberculosis study (32), but our PCA and UMAP analysis showed no separation according to disease state.”

I'm not sure if I missed it – but is the sex-effect replicated in the UK biobank data? And the linkage missing or apparent in the UK biobank data? One result that is glossed over (again apologies if I missed it in the supplement), but the effect looks reversed/stronger in females in the Andean population (sup Figure 4 d/f)? Is this true or just not significant/too noisy? Also, if this is testable with the other datasets available (since it is a much larger cohort), they could be used as negative controls of sorts. Ie the UK biobank and/or the other native cohorts.

We did test the LRLD in the UKB (Author response image 2), but no significant signal was seen, similarly to the GBR population (or any European population) in the 1000G. The AA genotype frequency at rs1967309 is in line with GBR in UKB White British participants (see details in response to R3.5). We added an Appendix – table 1 containing the r^2^ values and their p-values mentioned in the main text, and we also added a few other populations from 1000G and NAGD as negative controls.

**Author response image 2. sa2fig2:** LRLD of rs1967309 with CETP gene and rs158477 in ADCY9 gene in UKB for SNPs with a MAF>5%. The horizontal line is the 99^th^ percentile of all pairs of SNPs between ADCY9 and CETP genes.

2. Speculation/comments on why there would be a male/female difference in these population (or any). This circles back to the UK biobank question I had earlier. Are frequencies linked to Mt or Y chromosome haplotypes? Is there a way to measure or view this? I understand the RFMix analysis was meant to test for admixture, but would it also show something like this?

Yes, the reviewer is right that Y chromosome and Mt haplotypes are potentially at play here, and this would be presumably testable in the LIMAA cohort, but it is beyond the scope of the present study to evaluate this. It is however something we plan to investigate in follow-up analyses and we added a mention to the discussion:

“Lastly, it may be that the evolutionary pressure is linked to the sex chromosomes (69,70), and a three-way interaction between ADCY9, CETP and Y chromosome haplotypes or mitochondrial haplogroups remains to be explored.”

As for the male/female differences, we now discuss in more detail the proposed interpretation for sex-specific selection in this new version (see response to essential comment E.6). About the sex-effects on UKB phenotypes, we prefer not to speculate in the manuscript since more investigation is needed, but cardiovascular disease is known to be sexually dimorphic, and some of these differences are driven by genetics and interaction with sex hormones, so this constitute a possible hypothesis to be tested.

As it is hard to say definitively or exclude all potential counterfactuals for these results, without including additional datasets from the same region, along with geographic and historical accounts. I wouldn't want to speculate on this of course, but perhaps characterizing the samples a little further in the supplement might help (like a table of sorts, with totals, split by male/female, and any other known demographics). I found it a little difficult to follow all the different populations tested; and which is the "discovery" and which is the "replication" cohort. There are also some discrepancies in the numbers in the text, supplement and captions.PEL (1000G) LIMAA (?) Andean (NAGD) UK Biobank?Males 41 2076 ? 2078 54 ?Females 44 1433 ? 1434 34 ?Total 85 3509 ? 88 413083

We thank you for pointing out these discrepancies in the numbers in the text. This was due to sometimes reporting sample size before or after QC, and to problems when merging different versions of the initial manuscript. We have clarified whether the numbers are reported pre- or post-QC and verified them multiple times. We hope no more discrepancies remain.

Concerning better characterization of the samples, we added two tables to our manuscript:

1. Table 1 contains the name of each cohort and its abbreviation, the number of samples with percentage of females, the ethnicity, and the age (if this information was available). This is also in response to essential comment E.1 above.

2. Appendix – Table 1 displays information about the number of samples for each LRLD analysis, with their value of r2 and p-values. This is also in response to essential comment E.2 above.

3. Data quality and artefacts. Although this is probably unlikely (but a potential concern/caveat) – wouldn't imputing data from non-represented reference genomes lead to false positives or incorrect phasing? Just curious if there was any check using a different reference panel? I understand this might be out of the scope of this particular analysis to delve into more details about it, but maybe as a sanity check? I'm coming to this from looking through this paper that evaluated potential influences/causes for LRLD:

https://journals.plos.org/plosone/article?id=10.1371/journal.pone.0080754

Indeed, we had reported that the INFO score for our two SNPs imputed in LIMAA and NAGD with the Haplotype Reference Consortium (HRC) was lower than our 0.8 initial threshold in Appendix, section “Pre-processing of the LIMAA cohort”. Based on this comment and essential comment E.5, we performed additional analyses to make sure that the lower INFO score was due to low confidence because of the factors described above, and not to incorrect imputed genotypes. We thus redid the imputation of LIMAA and NAGD with the TOPMED reference panel and we recomputed the genotype correlation (LRLD) with these new data in LIMAA, and the value obtained is r^2^ = 0.0047 (*vs*. 0.0046 before). Given these very tiny difference, and the time it would require redoing all figures and null distributions with the new imputed data, we decided not to change our analyses in the manuscript and keep imputation with HRC, while adding these considerations to Appendix 1 in the section “Pre-processing of the LIMAA cohort” and “Genotype association between rs1967309 and rs158477 in LIMAA”.

See response to essential comment E.5 above for details on changes made.

4. Sex-biased gene expression and eQTLs. Was there any sex-biased expression in the assessed data? Noted sample sizes are still limited here, so I agree this might not be powered. But for some tissues, if you repeat the PEER analysis not only by tissue, but by sex? Correct me if I am wrong, but including sex as a covariate would regress out the sex effects? I'd be interested in seeing the split analysis, if there was a modulating effect that was also sex specific.

We thank the reviewer for this comment, as this resulted in the most significant addition to the new version of the manuscript. Initially, we performed a preliminary sex-stratified analysis in our RNAseq discovery cohort (GEUVADIS) which were not very conclusive due to low sample size and did not pursue this in other cohorts (GTEx and CaG). But thanks to this comment, we calculated PEER factors by stratifying the datasets by sex (sPEER factors) as suggested and we performed further analyses in GTEx and CaG that produced very interesting new results, in line with the sex-specific nature of our other results, that we are happy to report.

See response to essential comment E.4 above for summary of results and details on changes made.

Is the expectation that ADCY9 and CETP are negatively correlated with a particular genotype but only in males? Or is there no expectation for sex-specific functional differences too (ie it is either genetic or phenotypic/functional). One way to check could be to look to see if these genes correlated in the GTEx/GEUVADIS/CARTaGENE data.

To answer your question, we looked at the correlation between expression of *ADCY9* and *CETP* across tissues from GTEx. In the Author response image 3 (left panel) we show the p-value of the correlation *ADCY9* and *CETP* expression (correcting for sex, age, PCs, collection site, sequencing platform and total ischemic time) combining males and females. The genes indeed correlate negatively in most of the tissues of GTEx, which concords with the results in ADCY9-KD experiment. We added Appendix 1-figure 6 to report this.

**Author response image 3. sa2fig3:** Correlation between ADCY9 and CETP genes across tissues of GTEx. (left) sex-combined, the shape represents the direction of the correlation. (right) Comparison of β of the correlation between both genes between male and female. Bars represent the standard error obtained from the summary of the lm() in R.

We then stratified the dataset by sex and present (Author response image 3, right panel) the comparison of the estimated betas in males and females. The direction of effect is generally consistent and most of the tissues show that ADCY9 and CETP expressions are negatively correlated in both sexes. The tissue which is an outlier is the breast mammary tissue. We do not include this sex-stratified comparison of betas in the manuscript as we feel that it will need to be better characterized in the context of how *CETP* expression correlates with other genes of the transcriptome. Because it is independent of our SNPs, this is beyond the scope of this study.

“When looking across tissues in GTEx, ADCY9 and CETP expressions negatively correlate in almost all the tissues (Appendix 1-figure 6, Appendix), which is consistent with the effect observed during the ADCY9-KD experiment, showing increased expression of CETP expression when ADCY9 is lowly expressed (Figure 5a, Figure 5—figure supplement 1a,b).”

Appendix 1: “To test if ADCY9 and CETP expression is correlated across tissues in humans, we used data GTEx and performed a linear regression correcting for the first 5 PCs, age, sex, the collection site (SMCENTER), the sequencing platform (SMGEBTCHT) and total ischemic time (TRISCHD). We find that ADCY9 and CETP gene expression levels are negatively correlated (and significantly(p<0.05) so for Adipose-Subcutaneous, Adrenal Gland, Artery Coronary, Artery Tibial, Brain-Cerebellar Hemisphere/Cerebellum/Cortex/Frontal Cortex/Putamen (basal ganglia), Breast-Mammary Tissue, Esophagus-Gastro esophageal Junction/Muscularis, Heart-Left Ventricle, Lung, Prostate, Small Intestine-Terminal, Spleen, Stomach, Uterus, Whole blood), except in skin tissues and cells cultured fibroblast, for which a significant positive correlation is found (Appendix 1-figure 6).”

5. Epistatic/phenotypic data associations. The difference in genotype and MI association is a very interesting effect, but because the data is from a different population, I believe the message/point is kind of lost in the narrative. What are the distributions of the genotypes in the UK biobank? And how do these differ by sex? And what are the distributions of the two genotype combinations? Furthermore, a lot of these traits are correlated, I'm curious to see what those correlations are for these selected traits? Those that aren't associated with the traits but are associated when split by genotype, is that also a numbers effect? I'm not sure I caught that in the tests.

The data being from a different population is a clear limitation of our results here, as stated in the Discussion. We looked at the distribution of the genotypes in the UKB for both of our SNPs, for sex-combined and stratified by sex and we present them in the Author response table 3 and report them in the Appendix 1. We also present the number of individuals by genotypes combination (Author response figure 4), showing that the frequencies and distribution of the combination between our SNPs are higher similar in sex-combined and sex-stratified groups.

**Author response table 3. sa2table3:** 

	All	Male	Female
rs1967309-A	0.392818	0.392935	0.392719
rs158477-G	0.471628	0.471797	0.471484

– Appendix 1: “Individually, SNP rs1967309 (fA=39%) is nominally associated with heart rate, and rs158477 (fG=47%) with the systolic blood pressure […]”

Author response table 3. Allelic frequencies by sex and sex combined of rs1967309 and rs158477 in the UKb.

**Author response image 4. sa2fig4:** Number of samples by genotype combination, in sex-combined and stratified by sex.

About the correlation between phenotypes, we used a simple cor.test() in R between our selected phenotypes (without adjusting for any covariates, Author response image 5). As expected, many phenotypes are highly correlated, and when we stratified by sex, the correlation is similar for both sexes. We note that the correlation between FEV and FVC and CAD (ischaemic heart disease) is very low (r2 = -0.03 and -0.05, respectively for the sex combined).

**Author response image 5. sa2fig5:** Correlation between phenotypes in the UKB in sex-combined and stratified by sex.

Figure suggestions/comments:Figure 1 and consistency with labels/IDs. Would it be possible to show the LIMAA cohort here? It is not the "First nation" set -which I believe is the NAGD? The AND subpopulation (Andean) is the one carried on in the analyses, correct? It would be good to highlight these/specify in the caption and/or the figure. Is there a way to also show these frequencies by sex in those subpopulations you later work with? Ie for PEL, AND and LIMAA?In Figure 1b – Is the recombination rate the lines? Would it be possible to only plot the AMR? It is hard to see/interpret the results here. The others could be in a supplementary FigureIn Figure 1 c – the PBS is also hard to see clearly here. One option would be to smooth the lines out or again, only show the PEL population.The sex specific results are also super interesting (sup Figure 3 b/d) -> I think moving these also to this main figure shows a clearer point on sex effects.iHS and recombination rates. I'm not very aware of this measure, but I was curious as to whether the other SNPs that have highly significant iHS values are of relevance, and what the interpretation would be here? Ie the one near the start of the gene seems to have the highest iHS value for the AMR pop. Is that also in high LD with the main SNP of interest in the ADCY9 gene?Figure 2:Figure 2c, also show the LIMAA cohort as suggested for figure 1?Figure 4: Could you repeat panel b with the GTEx data at all in the well-powered tissues?Figure 5:Points in panel a are a little small, perhaps increasing the point size as in panel b/c?

We thank the reviewer for these suggestions, we integrated them in the revised version and describe the changes to figures in response to essential comment E.7 above. About the question on iHS and recombination rates, we have the following sentence in Appendix 1:

“Other SNPs in ADCY9 are found to have absolute iHS values higher than 2, especially in the long intron 1 and around the last exon, but characterizing these signals is beyond the scope of this study.”